# Pathogenic *BRCA1* variants disrupt PLK1-regulation of mitotic spindle orientation

Zhengcheng He[1], Ryan Ghorayeb[1], Susanna Tan[2,3], Ke Chen[1], Amanda C. Lorentzian[1], Jack Bottyan[1],
Syed Mohammed Musheer Aalam [4], Miguel Angel Pujana [5], Philipp F. Lange [6,7], Nagarajan Kannan [4,8],
Connie J. Eaves [2,3,6,9,10] & Christopher A. Maxwell [1,7✉]

Preneoplastic mammary tissues from human female *BRCA1* mutation carriers, or *Brca1*-mutant mice, display unexplained abnormalities in luminal differentiation. We now study the division characteristics of human mammary cells purified from female *BRCA1* mutation carriers or non-carrier donors. We show primary *BRCA1* mutant/+ cells exhibit defective BRCA1 localization, high radiosensitivity and an accelerated entry into cell division, but fail to orient their cell division axis. We also analyse 15 genetically-edited *BRCA1* mutant/+ human mammary cell-lines and find that cells carrying pathogenic *BRCA1* mutations acquire an analogous defect in their division axis accompanied by deficient expression of features of mature luminal cells. Importantly, these alterations are independent of accumulated DNA damage, and specifically dependent on elevated PLK1 activity induced by reduced BRCA1 function. This essential PLK1-mediated role of BRCA1 in controlling the cell division axis provides insight into the phenotypes expressed during *BRCA1* tumorigenesis.

[1] Department of Pediatrics, University of British Columbia, Vancouver, British Columbia, Canada. [2] Terry Fox Laboratory, British Columbia Cancer Research Institute, Vancouver, British Columbia, Canada. [3] Department of Medicine, University of British Columbia, Vancouver, British Columbia, Canada. [4] Division of Experimental Pathology and Laboratory Medicine, Department of Laboratory Medicine and Pathology, Mayo Clinic, Rochester, MN, USA. [5] ProCURE, Catalan Institute of Oncology, Oncobell, Bellvitge Institute for Biomedical Research (IDIBELL), L'Hospitalet del Llobregat, Barcelona, Catalonia, Spain. [6] Department of Pathology and Laboratory Medicine, University of British Columbia, Vancouver, British Columbia, Canada. [7] Michael Cuccione Childhood Cancer Research Program, British Columbia Children's Hospital, Vancouver, British Columbia, Canada. [8] Mayo Clinic Cancer Center, Mayo Clinic, Rochester, MN, USA. [9] Department of Medical Genetics, University of British Columbia, Vancouver, British Columbia, Canada. [10] School of Biomedical Engineering, University of British Columbia, Vancouver, British Columbia, Canada. ✉email: cmaxwell@bcchr.ubc.ca

Germline variants in *BRCA1 DNA repair associated* (*BRCA1*) confer an increased risk of breast cancer that can reach 72% by an age of 80 years[1] subject to modification by common genetic variants[2]. Risk is highest for carriers of protein-truncating variants and elevated for carriers of rare missense variants, specifically those located in the RING domain or the BRCT1 domain[3]. Breast cancer produced in women with pathogenic *BRCA1* variants tend to be estrogen receptor (ER)-negative and triple-negative[4], exhibit genomic heterogeneity[5], and lack phenotypic characteristics of the mature luminal cells of the normal adult female mammary gland[6]. These findings have led to the concept that BRCA1 plays an important role in modulating the proliferative and differentiation activity of primitive types of normal mammary cells[7]. Several findings support this concept[8–11], including the reports of an accumulation of luminal progenitors (LPs)[12] and fewer ER+ luminal cells[13] in pre-neoplastic *BRCA1*-mutant human mammary tissue. In transgenic *BLG-Cre;Brca1f/f;p53+/−* mice, longitudinal single-cell RNA-sequencing analysis of their mammary tissue also indicates their genesis of triple-negative mammary tumors is preceded by aberrant LP differentiation[14]. Loss of BRCA1 function in animal models has been causally associated with defects in cellular proliferation[15] and mitotic integrity[16] and we have previously shown loss of control of mitotic spindle orientation in human mammary cells engineered to express reduced levels of BRCA1[17]. However, the relevance of these findings to human mammary progenitor cells isolated from female *BRCA1* mutation carriers and the pathways involved have remained unresolved.

In epithelial tissues, progenitor cell division oriented relative to the basement membrane is normally highly controlled[18]. The general process of oriented cell division requires the mitotic spindle to be positioned by a polo-like kinase 1 (PLK1)-mediated pathway at spindle poles[19], in collaboration with hyaluronan-mediated motility receptor (HMMR)[20,21] and dynein light chain LC8-type 1 (DYNLL1)[20] to reduce the activity of dynein complexes at the cell cortex[19]. The PLK1-mediated intrinsic positioning pathway is also active in the mammary gland[22], where oriented cell divisions help to determine notch receptor 1 (NOTCH1) signaling in daughter cells[23,24]. The potential for disruption of oriented cell division to occur via pathogenic *BRCA1* mutation is not known but is supported by the location of BRCA1 to the mitotic spindle[25], its regulation of HMMR degradation[16,26], and deficits in oriented divisions observed in human mammary epithelial cells transduced with shRNA that target BRCA1[17]. Further supporting this idea is the finding that genetic variations in *HMMR*[27] modify breast cancer risk in *BRCA1* mutation carriers and DYNLL1 expression correlates with survival in *BRCA1*-mutant patients that develop ovarian cancer[28].

Here, we show that active PLK1 is the shared downstream mediator of two hallmarks of *BRCA1*-mutant mammary cells: an altered orientation of progenitor cell division and an accompanying deficit in the expression of features of mature luminal cells.

## Results

### Mammary progenitor populations from *BRCA1* mutation carriers exhibit multiple alterations in proliferation.
We first asked whether mammary progenitor cells isolated from *BRCA1* mutation carriers show altered responses to DNA damage and/or changes to their spindle orientation by comparison to non-carrier controls. Accordingly, we isolated epithelial cell fractions with proliferative activity[29] at high purity by fluorescence-activated cell sorting (FACS). Epithelial cell adhesion molecule (EpCAM)$^{high}$ integrin subunit alpha 6 (CD49f)$^+$ luminal progenitors (LPs) and EpCAM$^{low/−}$CD49f$^+$ basal cells (BCs) were isolated (Figure S1a)

from reduction mammoplasty samples from 3 cancer-free non-carriers or prophylactic mastectomies obtained from 3 *BRCA1*-mutant donors (Fig. 1a, Table S1). Endpoints of DNA damage, proliferation and oriented cell division were assessed in progeny generated within 8 days in vitro on collagen-coated dishes (Figure S1b).

DNA damage was induced by exposure of these LP- and BC-derived cells to 1 Gy of X-radiation. Since BRCA1 is known to form distinctive nuclear foci at γH2Ax-positive sites of DNA damage[30,31] in a cell cycle-dependent manner[32], we measured BRCA1-positivity and γH2Ax-positive nuclear foci in Cyclin B1 (CCNB1)-negative (cell cycle-matched) cells both 30 min and 24 h after X-radiation (Fig. 1b, Figure S1c,d). BRCA1-positive foci were statistically significantly reduced among the progeny of both the LPs and BCs isolated from the *BRCA1* mutation carriers as compared to non-carriers (Fig. 1c), consistent with results reporting pathogenic variants associate with the bottom quartile of expression[33]. LP or BC sources of non-carrier and *BRCA1*-mutant cells did not differ either in the basal levels of γH2Ax-positive foci, nor their levels measured 30 min post-irradiation (Fig. 1d). One day post-irradiation, however, γH2Ax-foci were significantly elevated in the *BRCA1*-mutant LP progeny and reduced in the *BRCA1*-mutant BC progeny compared to their non-carrier counterparts (Fig. 1d). Simultaneous assessment of markers of mitotic instability, including micronuclei and nuclear budding[34,35] (Figure S2a), also showed a substantial concomitant induction of these in the irradiated *BRCA1*-mutant samples (Fig. 1e, Figure S2b).

Consistent with the expression of a mitotic instability phenotype, *BRCA1*-mutant samples displayed a corresponding ~50% reduction in their content of cells with colony-forming activity[29] at day 8 post 1 Gy X-radiation (Fig. 2a). To determine whether the reduced colony-forming activity was related to a potential loss of proliferative capacity or mitotic spindle control[17], individual cells were seeded on collagen-coated L-shaped micro-patterned plates. L-shaped patterns define the adhesion cues provided to a round mitotic cell and educate the intrinsic spindle positioning mechanism to orient cell division along the hypotenuse of the pattern[36]. We then measured the fraction of individual adhered cells that underwent at least one mitosis over the next 24 h. This showed the initial rate of entry into mitosis of the *BRCA1*-mutant LP- and BC-derived cells was significantly faster than that of their non-carrier counterparts (Fig. 2b, c, Figure S3a). However, the same *BRCA1*-mutant LP- and BC-derived cells showed reduced levels of cell death (Figure S3b) and a selective loss of the normal ability to orient their divisions in response to collagen-mediated adhesion cues (Fig. 2b, d, Figure S3c). Taken together, these analyses indicate primitive *BRCA1*-mutant cells display a higher rate of cell division despite their characteristically low levels of BRCA1-positive nuclear foci, an enhanced radiosensitivity, and a loss of cell division orientation control.

### The intrinsic control of cell division orientation is selectively altered in mammary cells with pathogenic *BRCA1* mutations.
To determine whether pathogenic or benign variants that span the entire *BRCA1* gene show different effects on their control of cell division orientation, we acquired 11 MCF10A cell lines with different heterozygous *BRCA1* mutations proximal to the RING domain. These included a negative control variant (*Ex 10/+*), a positive control variant (*ΔEx2-3/+*), four pathogenic variants (*185delAG/+;C61G/+;C64R/+;R71G/+*), four benign variants (*D67Y/+;L246V/+; Q356R/+;I379M/+*), and one variant of uncertain significance (VUS) (*S316G/+*)[37]. We also generated 4 additional MCF10A cell lines encoding variants in the C-terminal BRCT

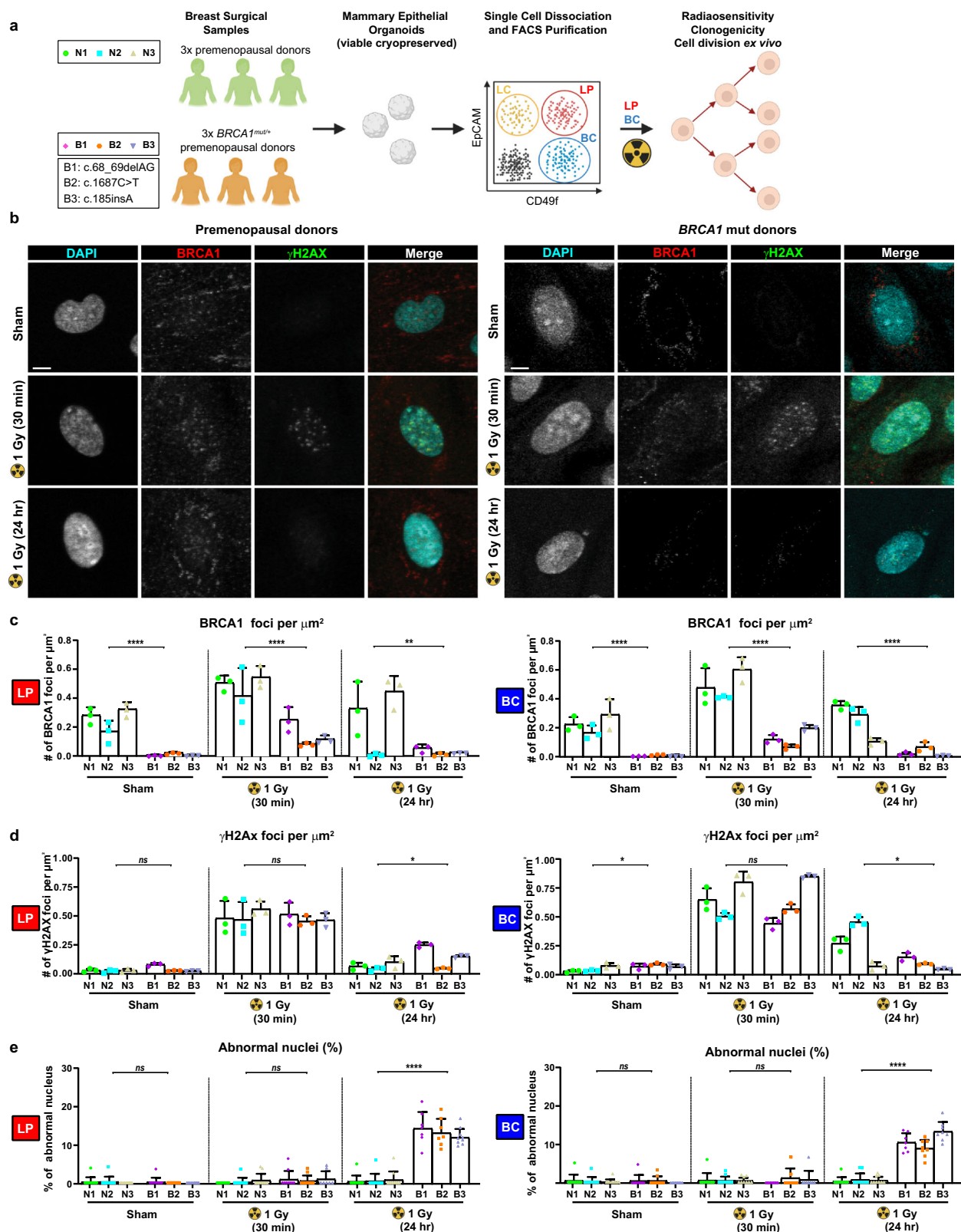

domains of BRCA1 (*A1780E/+;5832insG;R1835X/+;W1837R/+*) (Figure S4a,b). These 15 unique *BRCA1* mutant/+ MCF10A cell lines were then sham-treated or X-radiated with 1 Gy and BRCA1 expression levels were assessed 30 min later (Fig. 3a, Figure S4c). Cell lines encoding benign *BRCA1* variants, a negative control variant, and the parental line expressed similar levels of BRCA1 foci (Fig. 3b). In contrast, the cell lines encoding five pathogenic *BRCA1*

variants (*185delAG/+;C61G/+;C64R/+;A1780E/+;5832insG*) and the VUS expressed significantly reduced levels of BRCA1 nuclear foci (Fig. 3b), similar to our results for the cells derived from the LPs and BCs isolated from the human *BRCA1*-mutant mammary tissues (Fig. 1b). However, three MCF10A cell lines that encode *BRCA1* variants classified as pathogenic or likely pathogenic (*R71G/+;R1835X/+;W1837R/+*) expressed normal levels of BRCA1

**Fig. 1 Mammary progenitor cells from _BRCA1_ mutation carriers express low BRCA1 and enhanced radiosensitivity. a** Schematic overview of the experimental strategy. Non-cancer breast tissue samples from three premenopausal non-carrier samples (N1, N2, and N3) and three premenopausal _BRCA1_ mutation carriers (B1, B2, and B3) were preserved as mammary epithelial organoids and dissociated. Luminal progenitor (LPs) and basal cell (BC) subpopulations were FACS-purified based on surface marker expression, and cultured to measure proliferation and cell division orientation. **b** Immunofluorescence analysis of BRCA1 and phospho-Histone H2AX (Ser139) (γH2Ax) in Cyclin B1-negative (not shown) LPs isolated from a premenopausal non-carrier donor (N2) or a _BRCA1_ mutation (mut) carrier (B1). LPs were sham-treated, or irradiated with 1 Gy X-radiation, and fixed 30 min or 24 h later. Scale bars = 4 μm. **c** BRCA1-positive foci per μm$^2$ nucleus area measured in LPs or BCs isolated from premenopausal non-carrier donors or _BRCA1_ mutation carriers. BRCA1-positive foci are reduced 3-fold (30 min post-IR) to >20-fold (sham) in _BRCA1_ mutation carriers (mean ± SD, triplicate experiment values shown, $n = 20$ cells per experiment). ****$P = 2.48E{-}07$, ****$P = 4.81E{-}06$, **$P = 0.0069$, ****$P = 6.06E{-}07$, ****$P = 2.76E{-}07$, ****$P = 5.72E{-}05$; two-tailed unpaired $t$-test. **d** γH2Ax-positive foci per μm$^2$ nucleus area measured in LPs and BCs isolated from premenopausal non-carrier donors and _BRCA1_ mutation carriers (mean ± SD, triplicate experiment values shown, $n = 20$ cells per experiment). ns $P = 0.1214$, ns $P = 0.595$, *$P = 0.0273$, *$P = 0.0216$, ns $P = 0.6898$, *$P = 0.0114$; two-tailed unpaired $t$-test. **e** Percentage of abnormal nuclei measured in BCs and LPs isolated from premenopausal non-carrier donors or _BRCA1_ mutation carriers (mean ± SD, field of view values shown, triplicate experiments, 3 fields of view per experiment). ns $P = 0.4906$, ns $P = 0.2888$, ****$P < 1E{-}15$, ns $P = 0.9006$, ns $P = 0.7111$, ****$P = 1E{-}15$; two-tailed Mann–Whitney test. Representative images of micronucleus and nuclear budding are provided in Supplementary Figure S2a. Abnormal nuclei are subdivided as micronucleus or nuclear budding in Supplementary Figure S2b.

foci (Fig. 3b). While mitotic cell lysates prepared from individual _BRCA1_ mutant/+ MCF10A cell lines showed alterations in total levels of BRCA1 relative to parental cells, a consistent change in the total levels of BRCA1 was not observed across the 15 unique BRCA1 mutant/+ MCF10A cell lines (Fig. 3c).

We next cultured the 15 _BRCA1_ mutant/+ cell lines as single cells on fibronectin-coated L-shaped micro-patterned plates (Fig. 3d) and determined the cell division angles displayed (Figure S4d). These showed the MCF10A cells encoding benign _BRCA1_ mutant/+ genotypes correctly oriented cell divisions in contrast to all of the cells encoding pathogenic _BRCA1_ variants with the same 3 exceptions (i.e., R71G/+;R1835X/+;W1837R/+) (Fig. 3e); for illustrative _BRCA1_ mutations, cell division angles were confirmed in two distinct edited clones (Figure S4e). Notably, the ability to orient cell division was significantly correlated with the expression of BRCA1 nuclear foci ($R^2 = 0.754$; $P < 0.0001$) (Fig. 3f). This remarkable consistency between the behavior of cells obtained from female carriers of _BRCA1_ mutations and 15 cell lines engineered to mimic a spectrum of implicated _BRCA1_ variants suggests a mechanistic association of reduced BRCA1 foci and a deficient control of cell division orientation in mammary cells with clinically described pathogenic variants of _BRCA1_.

**DNA damage is not sufficient to impair the control of the cell division axis.** To further evaluate the impact of _BRCA1_ mutations on protein function, we measured other characteristics, including the levels of DNA damage and sensitivity to PARP inhibitors[38], in these _BRCA1_ mutant/+ cell lines. As found from our analyses of the mammary cells obtained from _BRCA1_ mutation carriers, the cells from the _BRCA1_ mutant/+ MCF10A models were also not distinguished by basal levels of γH2Ax foci (Figure S5a). However, low levels of BRCA1 foci measured in the MCF10A models did correlate with elevated levels of γH2Ax foci measured 24 h after exposure to 1 Gy X-radiation (Fig. 4a, Figure S5b), or 0.25 μg/ml mitomycin C (Figure S5c). Similarly, levels of BRCA1 foci positively correlated with resistance to 0.25 Gy radiation specified by progenitor numbers measured after 5 days ($R^2 = 0.679$; $P < 0.0001$) (Fig. 4b). Interestingly, however, sensitivity to the PARP inhibitor, talazoparib, as indicated by IC50 values, did not correlate with the numbers of BRCA1 foci (Fig. 4c). Across the 15 _BRCA1_ mutant/+ cell lines, an inability to orient the cell division axis correlated with elevated levels of persistent DNA damage (Figure S5d,e), suggesting a relationship between elevated DNA damage and misoriented cell division in _BRCA1_-mutant cells.

DNA lesions can induce chromosome bridges and changes in ploidy[39,40], which have the potential to delay metaphase and disturb the intrinsic positioning pathway. Thus, we tested whether graded doses of X-radiation might be sufficient to induce a misorientation of cell division in the parental MCF10A cells. Exposure to graded doses of up to 1 Gy induced numerous dose-dependent phenotypic changes in these cells apparent 24 h later, including increased numbers of γH2Ax foci (Fig. 4d) and abnormal anaphase cells (Figure S6a), reduced cell viability (Figure S6b), increased micronuclei and abnormal nuclei (Figure S6c), and decreased 4-day clonogenic activity (Fig. 4e). However, the intrinsic positioning pathway in these cells was not altered (Fig. 4f). Despite dose- and time-dependent changes to cell viability and abnormal nuclei up to 72 h post X-radiation (Figure S6b,c), parental MCF10A cell division angles were not altered at these later timepoints (Figure S6d) and X-radiation (1 Gy) did not exacerbate the division axis in _BRCA1_ mutant cells (Figure S6e). To test this relationship more directly, we transfected parental MCF10A cells to express mCherry-53BP1. One day later, sham-treated or X-radiated (1 Gy) cells were imaged on L-patterns to track 53BP1-positive DNA damage foci and cell division angles in the same individual cells (Fig. 4g). This analysis revealed an increase in mean 53BP1-positive foci in the irradiated cells (sham: 2.1 ± 0.3 foci; irradiated: 9.1 ± 0.8 foci) without corresponding changes to the cell division axis (sham: 10.2° ± 1.7°; irradiated: 8.3° ± 1.7°) in the same cells (Fig. 4h). Thus, demonstrable levels of induced DNA damage were not sufficient to disturb the intrinsic spindle positioning pathway.

**Hyperactive PLK1 deregulates the intrinsic spindle position pathway in mammary cells encoding heterozygous pathogenic _BRCA1_ variants.** To gain insight into the mechanism by which loss of BRCA1 disturbs control of the cell division axis, we analyzed the proteomic differences between MCF10A cells transduced with lentiviral vectors expressing shRNA targeting BRCA1 (shBRCA1, hereafter referred to as BRCA1-silenced), or a non-hairpin shRNA (shNHP) as a negative control. Western blot analysis showed a significant and prolonged reduction in BRCA1 expression was achieved in the BRCA1-silenced cells (Figure S7a) and a corresponding reduction in BRCA1 nuclear foci similar to that seen in the MCF10A cells encoding heterozygous pathogenic _BRCA1_ mutations (Figure S7b). Proteome analysis performed on lysates isolated from the control-transduced and BRCA1-silenced sub-confluent MCF10A cultures showed differential over-representation of gene ontology (GO) terms related to G1/S transition, DNA replication, and activation of ataxia telangiectasia and rad3 related (ATR) in response to replication stress

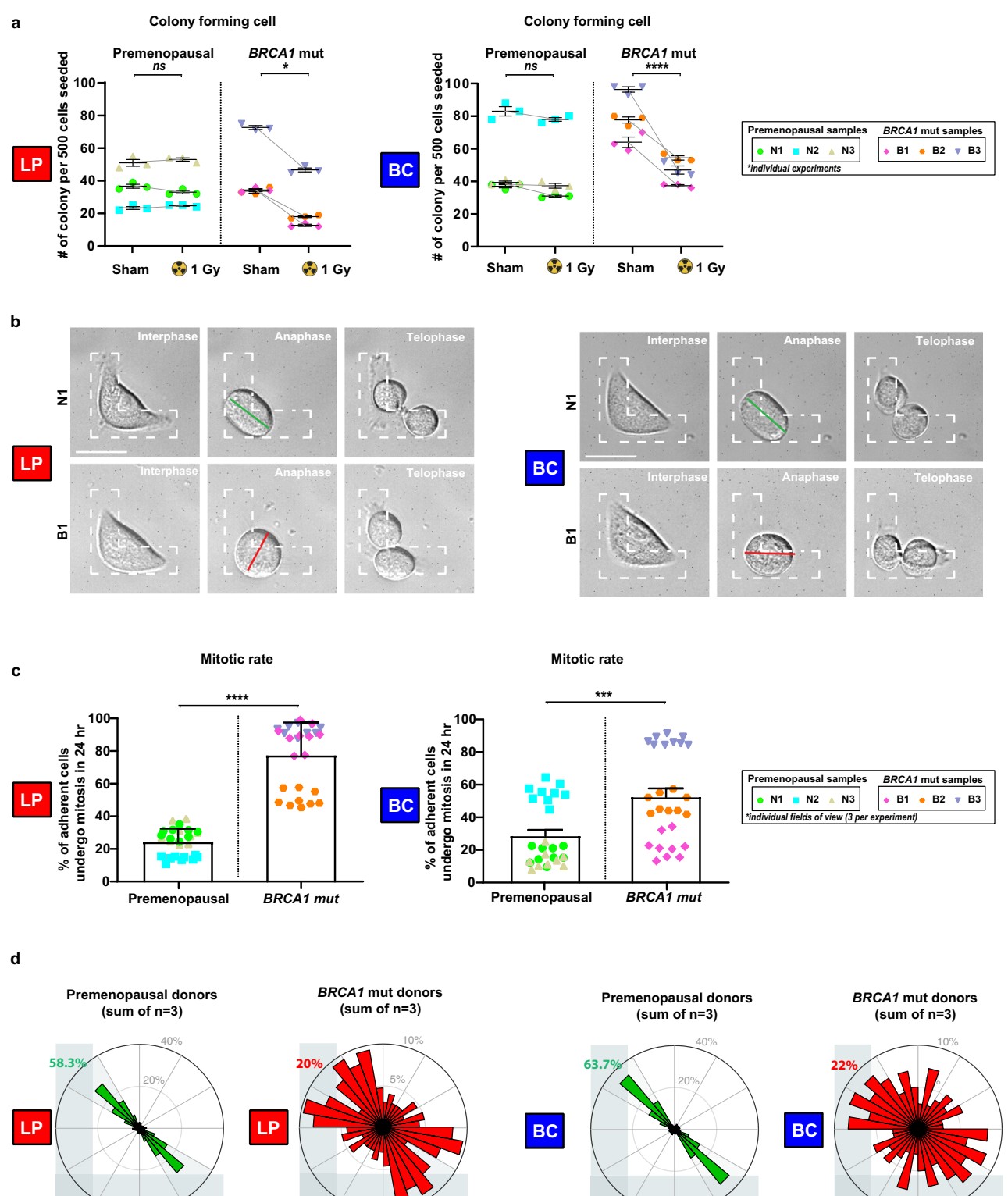

(Fig. 5a); conversely, GO terms related to metabolic processes were underrepresented (Figure S7c,d). The most over-represented terms in the BRCA1-silenced cells were related to cell cycle and mitosis (Fig. 5a), including overexpression of critical mitotic kinases: aurora kinase B (AURKB); PLK1; and, three PLK1-substrates: mitotic interactor and substrate of PLK1 (MISP), rio kinase (RIOK), and monopolar spindle 1 kinase (TTK/MPS1) (Fig. 5b). BRCA1-silenced cells also showed a significant increased time spent with aligned chromosomes during

metaphase, which was restored through incubation with a MPS1 inhibitor (100 nM NMS-P715) (Figure S8a-c). However, MPS1 inhibition was not sufficient to normalize control of the cell division axis for BRCA1-silenced cells (Figure S8d). Thus, the restoration of normal metaphase kinetics through MPS1 inhibition was not sufficient to correctly orient the cell division axis in BRCA1-silenced cells.

PLK1 activity at mitotic spindle poles co-ordinates the spindle positioning pathway[21,41]. Total levels of PLK1 and active PLK1

**Fig. 2 Mammary progenitor cells from *BRCA1* mutation carriers display an elevated mitotic rate, but do not orient cell divisions. a** Colonies formed after 8 days per 500 cells seeded for BCs and LPs isolated from premenopausal non-carrier donors (N1, N2, and N3) or *BRCA1* mutation carriers (B1, B2, and B3). Prior to seeding, cells were sham-treated or X-radiated with 1 Gy (mean ± SD, triplicate experiment values shown). ns $P = 0.9851$, *$P = 0.0225$, ns $P = 0.6867$, ****$P = 1.73E−05$; two-tailed unpaired *t*-test. **b** Individual LPs or BCs from a premenopausal non-carrier donor (N1) or a *BRCA1* mutation carrier (B1) seeded on fibronectin-coated, L-shaped micropatterns and imaged during cell division to determine the orientation of cell division at anaphase indicated by a line connecting the spindle poles. Scale bars = 20 µm. **c** Percentage of adherent cells that underwent mitosis in a 24-h period following seeding on L-shaped micropatterns (mean ± SD, field of view values shown, three fields per experiment, triplicate experiments). Data is presented for BCs and LPs isolated from premenopausal non-carrier donors (N1, N2, and N3) or *BRCA1* mutation carriers (B1, B2, and B3). Representative images are provided in Supplementary Figure 3a. ****$P < 1E−15$, ***$P = 0.0006$; two-tailed unpaired *t*-test. **d** Distribution of cell division angles measured at anaphase in 10°-wide sectors for BCs and LPs isolated from premenopausal non-carrier donors and *BRCA1* mutation carriers. Data is pooled for three patient samples per genotype (each patient has $n = 50$ cells per experiment, duplicate experiments). Distribution of cell division angles for individual samples is provided in Supplementary Figure S3c. Gray percentages indicate the percent of total mitotic cells examined.

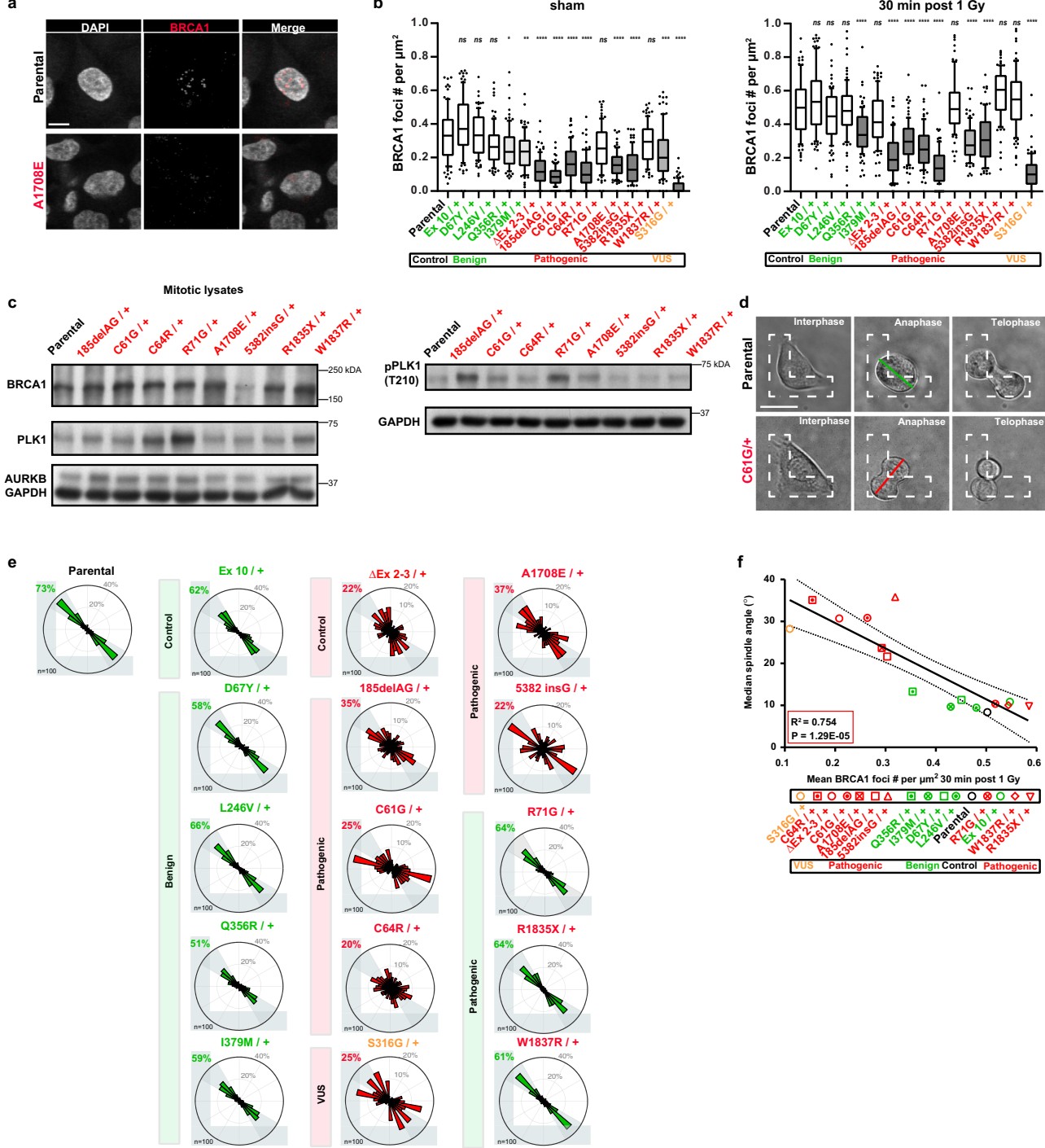

**Fig. 3 Single cell division angles distinguish *BRCA1* mutant MCF10A cells. a** Immunofluorescence analysis of BRCA1-positive nuclear foci in parental or *BRCA1^{A1708E/+}* MCF10A cells. Scale bar = 10 μm. **b** BRCA1-positive foci per μm² nucleus area in *BRCA1* mutant/+ MCF10A cells. In all panels, parental cells are in black text and *BRCA1*^{mutant/+} MCF10A cells edited to encode a benign variant (n = 5) is in green text, a pathogenic variant (n = 9) is in red text, or a VUS (n = 1) is in orange text. Box and whisker plot with median and 10–90 percentiles, n = 30 cells per experiment, triplicate experiments. Bars are filled according to P-value (white bars, ns; light gray bars, *P < 0.05 or **P < 0.01; gray bars, ***P < 0.001; dark gray bars, ****P < 0.0001). ns P = 1.00E + 00, ns P = 1.00E + 00, ns P = 0.558, *P = 0.0278, **P = 0.0055, ****P < 1E−15, ****P = 1E−15, ****P = 4.10E−11, ****P < 1E−15, ns P = 0.1328, ****P = 2.57E−12, ****P < 1E−15, ns P = 1.00E + 00, ***P = 1.18E−04, ****P < 1E−15; ns P = 1.00E + 00, ns P = 1.00E + 00, ns P = 1.00E + 00, ****P = 5.84E−05, ns P = 0.739, ****P < 1E−15, ****P = 1.05E−09, ****P = 1.40E−14, ****P < 1E−15, ns P = 1.00E + 00, ****P = 4.83E−11, ****P = 3.78E−08, ns P = 0.1249, ns P = 1.00E + 00, ****P < 1E−15; Kruskal–Wallis test; Dunn's multiple comparisons test. **c** Levels of BRCA1, PLK1, AURKB, and pPLK1 (T210) measured by immunoblot in nocodazole-synchronized mitotic lysates from MCF10A (parental) or MCF10A cells encoding causal *BRCA1* mutations (red text). GAPDH was used as a loading control. **d** Individual parental and *BRCA1^{C61G/+}* MCF10A cells seeded and imaged on fibronectin-coated, L-shaped micropatterns to determine the orientation of cell division at anaphase indicated by a line connecting the spindle poles. Scale bars=20 μm. **e** Circular graphs show the distribution of cell division angles measured in 10°-wide sectors during anaphase for parental MCF10A or *BRCA1*^{mutant/+} MCF10A cells. Variant site and variant classification are indicated (n = 50 cells per experiment, duplicate experiments). Gray percentages indicate the percent of total mitotic cells examined. **f** Correlation of median spindle angle and mean BRCA1 foci per μm² nucleus area measured 30 min post X-radiation (1 Gy) for parental MCF10A cells and *BRCA1*^{mutant/+} MCF10A cells. Values are plotted for including a simple linear regression (goodness of fit, $R^2 = 0.754$, $P = 1.29E-05$) and dashed lines indicate the 95% confidence band.

(phosphorylated T210, hereafter pPLK1) measured in whole-cell lysates were not consistently altered in the MCF10A cells encoding pathogenic *BRCA1* mutations (Fig. 3c). Therefore, we were prompted to measure the intensity of active pPLK1 in control and BRCA1-silenced mitotic cells. We first immunolocalized BRCA1 through mitosis and found its localization to be spatiotemporally-restricted at prometaphase and metaphase spindle poles (Figure S9a,b). Spindle-localized BRCA1 intensity was significantly reduced (Figure S9c), but total PLK1 was relatively unaffected (Figure S9d), in *BRCA1* mutant/+ MCF10A cells. The intensity of pPLK1 was increased 2-fold in the BRCA1-silenced cells (Fig. 5c) and was also significantly increased in *BRCA1* mutant/+ MCF10A cells, with the exceptions of R71G/+, R1835X/+, and W1837R/+ cells (Figure S9e) indicating strong correlation with an inability to orient the cell division axis (Figure S9f). We then asked whether inhibition of active PLK1 was sufficient to rescue the spindle positioning pathway. Treatment of the parental MCF10A cells with graded doses of BI2536, a PLK1 inhibitor, efficiently reduced pPLK1 intensity at spindle poles (Fig. 5d). However, because BI2536 can also severely inhibit mitotic progression and cell viability[42], we undertook serial dilution experiments. These showed that 0.1 nM BI2536 (1% of the IC50 dose, 10.8 nM) produced a significant reduction in pPLK1 at spindle poles with no discernible effect on viability (Fig. 5d, e) and was also sufficient to normalize the cell division orientation of the BRCA1-silenced cells (Fig. 5f) and *BRCA1* mutant/+ MFC10A cells (Figure S9g) without affecting the control-transduced cells (Fig. 5g). Similar rescue experiments performed on primary human *BRCA1*-mutant LP- and BC-derived cells also showed elevated pPLK1 (T210) intensity at the spindle poles (Fig. 5h, Figure S10a) and treatment of these cells with 0.1 nM BI2536 likewise reduced their hyperactive PLK1 (Figure S10b) and normalized their cell division orientation (Fig. 5i). These findings indicate the engineered suppression of BRCA1 activity in human mammary cells replicates the elevated pPLK1 intensity and defective cell division orientation exhibited by primary human *BRCA1*-mutant cells and both can be reversed by a non-toxic exposure to a small-molecule inhibitor of PLK1.

We then asked whether inhibition of related mitotic kinases can also rescue the defective cell division orientation exhibited by BRCA1-silenced cells. In contrast with pPLK1, the normalized intensity of active aurora kinase A (AURKA) (phosphorylated Thr288) and AURKB were not changed in BRCA1-silenced cells (Figure S11a, Figure S12a,b). We then tested serial dilutions for small-molecule inhibitors targeting AURKA (MLN8237) and AURKB (AZD1152) in parental cells to determine IC50 values

and the lowest dose able to significantly reduce kinase intensities (Figure S11b,c, Figure S12c,d). However, neither of these treatments was able to restore the altered cell division orientation of the BRCA1-silenced cells, and the low-dose treatments with either inhibitor also appeared to damage the process in the control cells (Figure S11d, Figure S12e). Finally, we determined whether hyperactivation of PLK1 (independent of *BRCA1* mutation) would be sufficient to misorient cell division. To do so, we generated three MCF10A clones that overexpress GFP-PLK1, and one clone that overexpresses GFP alone (Figure S13a) and found the overexpression of GFP-PLK1 to be sufficient to induce the loss of control of the cell division axis (Figure S13b). These findings indicate the specificity of the effect of deficient BRCA1 in inducing a PLK1 hyperactivity sufficient to disrupt the normal cell division axis control that is, nevertheless, reversible by small-molecule inhibition of PLK1.

**Colonies of BRCA1-silenced cells lose luminal features in a PLK1-dependent manner.** To investigate the possibility that the altered control of the cell division axis in BRCA1-deficient cells might also influence their acquisition of luminal differentiation features[43], we designed experiments to compare a number of phenotypic properties of the cells generated in 5-day colonies and whether PLK1 inhibition would rescue any abnormal phenotypes detected. For this, we used a MCF10A subline that expresses red fluorescence protein (RFP) at the endogenous *tubulin alpha 1B* (*TUBA1B*) locus to track mitotic spindle position and facilitate a quantitative analysis of their clonal progeny based on the density of their nuclei and the number of cell–cell contacts in the colonies produced (Figure S14a–c); these quantitative properties were highly correlated to expression of CDH1, ZO-1, and vimentin in colonies generated from primary LPs or BCs (Figure S14d–i) as well as in MCF10A colonies[17]. About 60% of control-transduced MCF10A RFP-TUBA1B cells produced dense colonies after 5 days (Fig. 6a). In contrast, BRCA1-silenced cells were poorly clonogenic and more prone to form sparse colonies (Fig. 6a), composed of vimentin-positive cells as found previously[17]. In fact, elevated expression of EGFP-PLK1 was sufficient to promote sparse colony formation in the absence of *BRCA1* mutation (Figure S13c). Moreover, PLK1 inhibition (0.1 nM BI2536) in BRCA1-silenced cells promoted the formation of more densely packed colonies, composed of CDH1+/ZO-1+ cells as found previously[17]; whereas, similar treatments of control-transduced cells had minimal effect on either colony yields or structure (Fig. 6b). The promotion of luminal features in BRCA1-silenced cells was also specifically dependent on PLK1 inhibition as

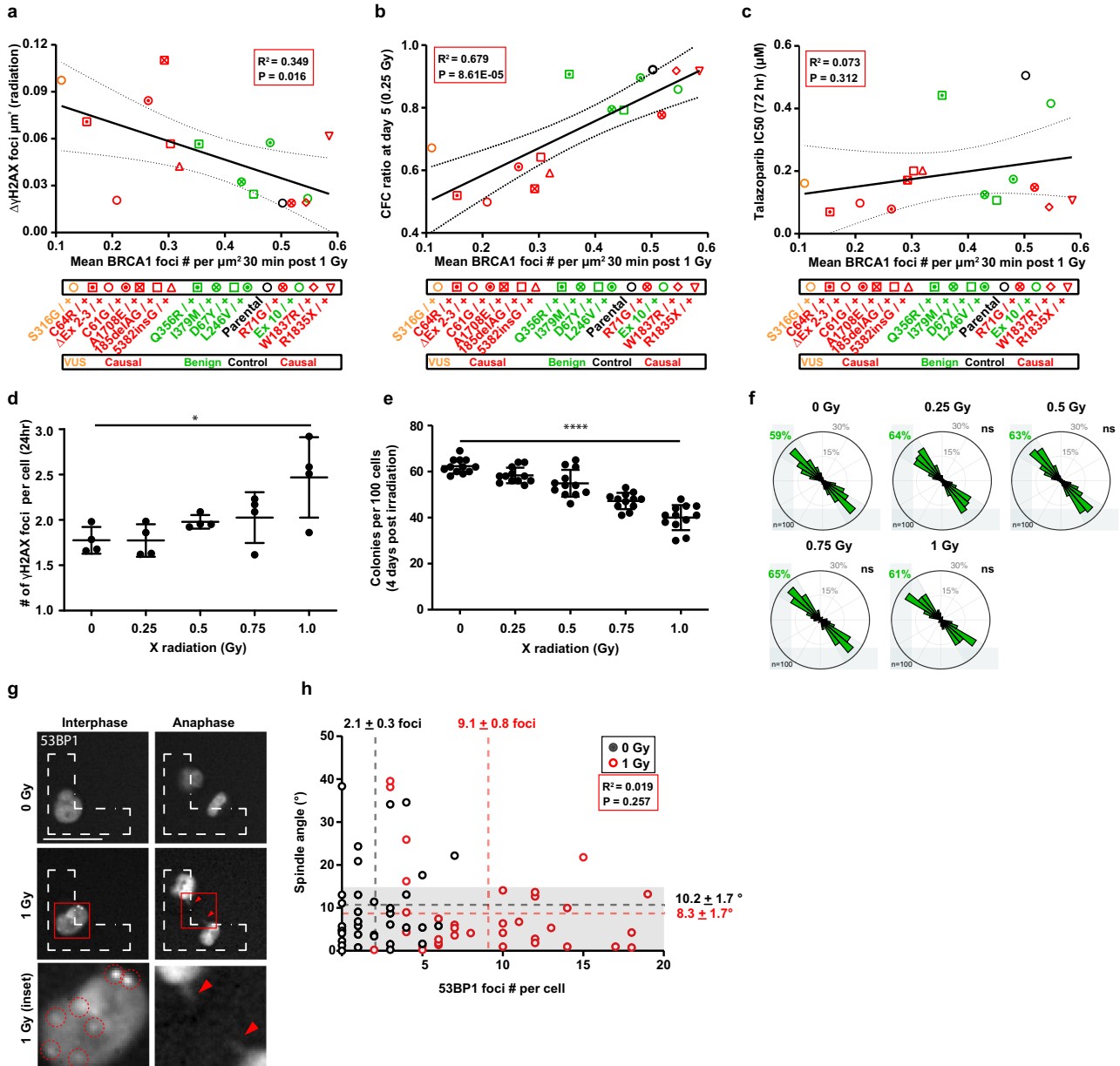

**Fig. 4 DNA damage is not sufficient to misorient cell divisions. a** Correlation plot of mean BRCA1-positive foci per μm² nucleus area and γH2AX-positive foci per μm² nuclear area for parental or *BRCA1*mutant/+ MCF10A cells. BRCA1 foci were measured 30 min after X-radiation (1 Gy). γH2AX-positive were measured 24 h after X-radiation (1 Gy) and normalized to basal levels in sham-treated cells (ΔγH2AX). Plotted values include a simple linear regression (goodness of fit, $R^2 = 0.349$, $P = 0.016$) and dashed lines indicate the 95% confidence band. In all panels, parental cells are in black text and *BRCA1*mutant/+ MCF10A cells edited to encode a benign variant ($n = 5$) is in green text, a pathogenic variant ($n = 9$) is in red text, or a VUS ($n = 1$) is in orange text. **b** Correlation plot of mean BRCA1-positive foci per μm² nucleus area and colony-forming cell (CFC) ratio measured at day 5 after 0.25 Gy X-radiation for parental or *BRCA1*mutant/+ MCF10A cells. Plotted values include a simple linear regression (goodness of fit, $R^2 = 0.679$, $P = 8.61E-05$) and dashed lines indicate the 95% confidence band. **c** Correlation plot of mean BRCA1-positive foci per μm² nucleus area and Talazoparib inhibitory concentration 50 (IC50) measured after 72 h for parental or *BRCA1*mutant/+ MCF10A cells. Plotted values include a simple linear regression (goodness of fit, $R^2 = 0.073$, $P = 0.312$) and dashed lines indicate the 95% confidence band. **d** Number of γH2AX-positive foci in parental MCF10A cells 24 h after X-radiation (mean ± SD, 4 wells from duplicate experiments, $n = 160$ cells per well). *$P = 0.0105$; one-way ANOVA. **e** Colony-forming cells per 100 parental MCF10A cells plated measured 4 days after X-radiation (mean ± SD; 6 wells from duplicate experiments per dose, 100 cells seeded per well). ****$P < 1E-15$; one-way ANOVA. **f** Circular graphs show the distribution of cell division angles measured in 10°-wide sectors at anaphase for parental MCF10A cells. Cells were imaged for 24 h after X-radiation (50 mitotic cells per experiment, duplicate experiments). ns = not significant; Mann–Whitney test. Gray percentages indicate the percent of total mitotic cells examined. **g** Parental MCF10A cell division on fibronectin-coated, L-shaped micropatterns after transfection with mCherry-53BP1 and sham or 1 Gy X-radiation. Representative images indicate mCherry-53BP1-positive foci (inset magnified) and lagging chromosomes in an irradiated MCF10A cell. **h** Mean spindle angle is plotted versus mean mCherry-53BP1 foci per cell ($n = 2$ experiments, 35 mitotic cells total examined) for sham-treated (black) or X-radiated (1 Gy, red) MCF10A cells. Spindle angles <15° (highlighted in gray area) were considered as correct spindle orientation. Scale bar = 20 μm. Values are analyzed with a simple linear regression (goodness of fit, $R^2 = 0.019$, $P = 0.257$). Dashed lines indicate the mean ± SD.

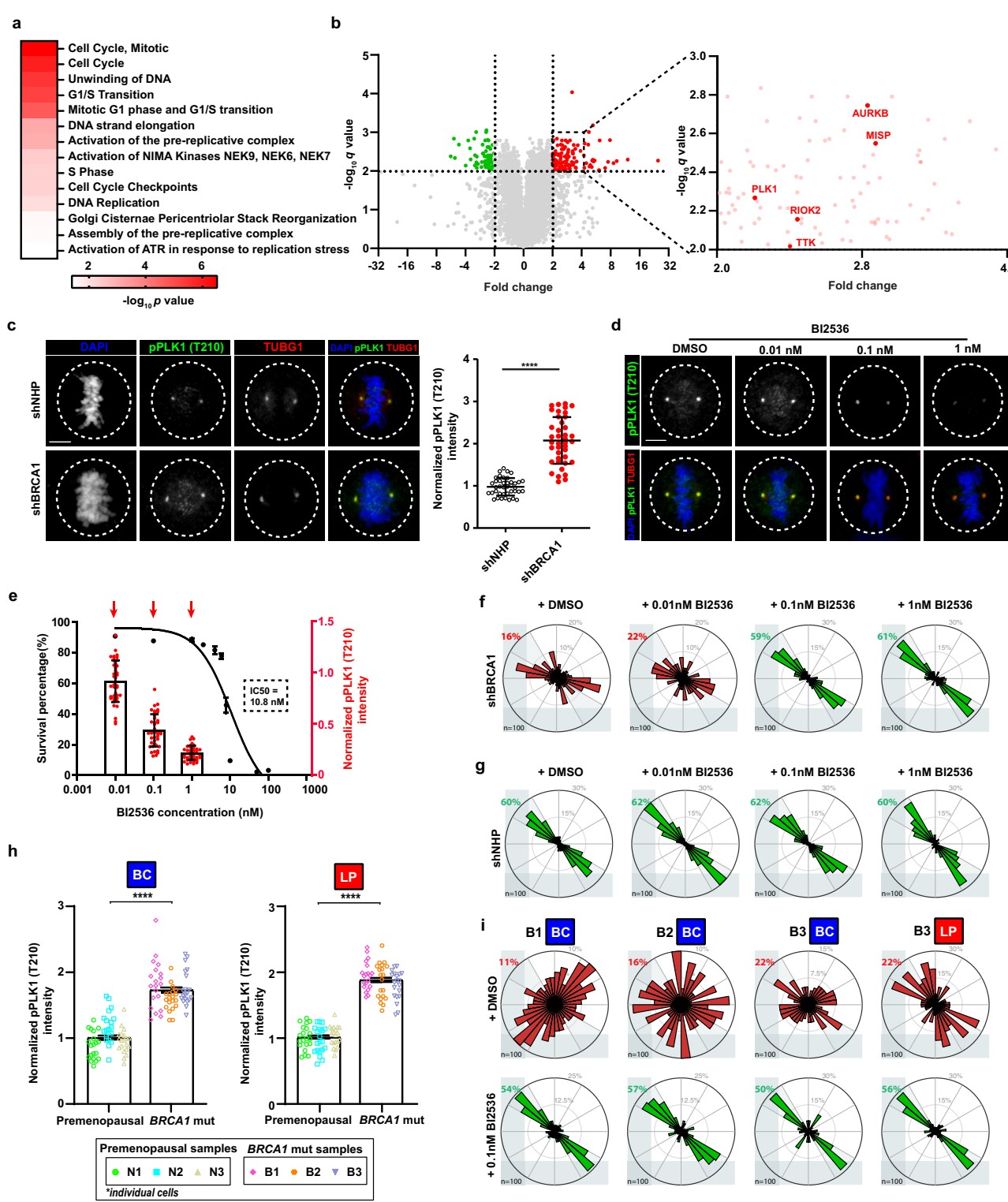

treatments with inhibitors against AURKA (MLN8237, Figure S15a) or AURKB (AZD1152, Figure S15b) did not modify colony structures despite significantly reducing the numbers of cells they contained. These findings indicate that the elevated PLK1 activity induced in BRCA1-deficient clonogenic cells compromises not only their ability to correctly orient their cell division axis, but also inhibits their generation of progeny with the cell–cell contact and expression features of luminal cells.

To gain further insight into this process, we imaged the colony formation process and measured the cell division axis and extent of proliferation and proximity of the progeny produced (Fig. 6c), as well as the number of cell–cell contacts measured by Delaunay triangulation (Figure S16a,b). Measurements undertaken after an initial 1st or 2nd division (after 36 h), showed control-treated cells proliferated with linear kinetics and their progeny maintain proximity (Fig. 6d) leading to more cell–cell contacts (Fig. 6e) and oriented cell divisions (Hertwig rule: <30° with respect to the cell's long axis, Figure S16c). In contrast, proliferation of the BRCA1-silenced cells was reduced (Fig. 6f), the orientation of the cell divisions was skewed (Figure S16d), and the progeny cells

**Fig. 5 PLK1 is hyperactive and its inhibition orients division in *BRCA1* mutant mammary cells. a** Gene Ontology (GO) terms of significantly over-represented proteins (>2-fold; *q* value < 0.01) in lysates from MCF10A cells transduced with shBRCA1 (BRCA1-silenced) versus MCF10A cells transduced with shNHP (control-transduced). *P* values were generated with g:SCS threshold (*P* < 0.05) from g:Profiler. **b** Volcano plot of differentially expressed proteins in BRCA1-silenced versus control-transduced MCF10A cells. Overexpressed (red) and underexpressed (green) proteins are indicated. Zoomed volcano plot represents proteins significantly elevated 2- to 4-fold with *q* value 0.001 to 0.01. Desired FDR (false discovery rate) < 0.01; multiple two-tailed unpaired *t*-test. **c** Immunofluorescence analysis of phosphorylated PLK1 (Thr210) (pPLK1) and gamma-tubulin (TUBG1)-positive mitotic spindle poles in control-transduced or BRCA1-silenced MCF10A metaphase cells. Mean ± SD pPLK1 intensity normalized to mean value for control-transduced cells and plotted for duplicate experiments including *n* = 20 individual mitotic cell values per experiment. ****$P$ < 1E−15; two-tailed unpaired *t*-test. Scale bar = 5 μm. **d** Immunofluorescence analysis of pPLK1 intensity on MCF10A metaphase spindle poles after treatment with a PLK1 small-molecule inhibitor, BI2536. pPLK1 intensity was normalized to mean value for vehicle treated cells. Scale bar = 5 μm. **e** Cell survival after 48 h (black, plotted as mean ± SEM, 3 experiments) and box plots of pPLK1 levels (red, plotted as *n*-20 individual cells per experiment, duplicate experiments) in MCF10A metaphase cells treated with BI2536. Red arrows indicate BI2536 doses shown in panel (**f**). **f** Distribution of cell division angles in 10°-wide sectors measured in BRCA1-silenced MCF10A anaphase cells treated with DMSO or BI2536 (*n* = 50 cells per experiment, duplicate experiments). For panels **f**, **g**, and **i**, gray percentages indicate the percent of total mitotic cells examined. **g** Distribution of cell division angles in 10°-wide sectors measured in control-transduced MCF10A anaphase cells treated with DMSO or BI2536 (*n* = 50 cells per experiment, duplicate experiments). **h** pPLK1 intensity measured at centrosomes in primary BCs or LPs isolated from premenopausal non-carrier donors (N1, N2, and N3) or *BRCA1*-mutation carriers (B1, B2, and B3) (mean ± SEM across three donors; *n* = 20 cells per donor represented as color-coded values). pPLK1 intensity was normalized to mean value for premenopausal non-carrier cells. ****$P$ < 1E−15, ****$P$ < 1E−15; two-tailed unpaired *t*-test. **i** Distribution of cell division angles in 10°-wide sectors measured at anaphase in primary *BRCA1*-mutant BCs or LPs treated with DMSO or 0.1 nM BI2536 (*n* = 50 cells per experiment, duplicate experiments).

failed to maintain proximity and form cell-cell contacts (Fig. 6f, g), resulting in clones consisting of sparse collections of vimentin-positive cells (Fig. 6a). BRCA1-silenced cells did not differ in shape (Figure S16e) but did require longer cycle times between consecutive cell division (Figure S16f) and showed higher rates of apoptosis (Figure S16g) explaining their reduced rate of clonal expansion (Fig. 6f). Thus, BRCA1-mediated oriented cell division appears essential to produce cells with multiple luminal features, and BRCA1-deficiency causes their progeny to sustain a basal phenotype.

**Cre-driven *Brca1* genomic loss alters organoid development in a PLK1-dependent manner.** Single-cell RNA-sequencing data of *Blg-Cre;Brca1^f/f^;p53^+/−^* mouse mammary cells indicated a cell-autonomous aberrant differentiation of the LP compartment[14]. To determine if this would likewise be related to an initial loss of cell division axis control and abnormal growth, we isolated mammary epithelial cells from 6-week old virgin *Brca1^+/+^;p53^+/+^* (WT) and *Blg-Cre;Brca1^f/f^;p53^+/−^* mice, transduced them with a lentivirus encoding EGFP-Cre or GFP-alone, isolated the GFP-positive cells by FACS (Figure S17a) and confirmed the Cre-driven loss of BRCA1 foci in EGFP-Cre-transduced *Blg-Cre;Brca1^f/f^;p53^+/−^* cells (Figure S17b). Examination of their proliferation in vitro showed they also failed to orient their divisions when grown as single cells on L-shaped micro-patterned plates (Fig. 7a), indicating this role of BRCA1 is also preserved in mice. Cell division angles were then determined relative to the basement membrane after 5 days of organoid culture for GFP-transduced WT cells, GFP-transduced *Blg-Cre;Brca1^f/f^;p53^+/−^* cells, and EGFP-Cre-transduced *Blg-Cre;Brca1^f/f^;p53^+/−^* cells (Figure S17c). The results showed division angles were heterogeneous (both parallel and perpendicular to the membrane) for control-transduced cells while they were generally restricted to a plane parallel (<30%) to the basement membrane for the EGFP-Cre-transduced *Blg-Cre;Brca1^f/f^;p53^+/−^* cells (Fig. 7b, Figure S17c). After 10 days in these cultures, the control-transduced cells could be seen to be generating robust organoid structures with budding protrusions. In contrast, the cells with a Cre-driven genetic deletion of *Brca1* showed a severely impaired production of organoid structures (Fig. 7c, d), which were often characterized by a round, spheroid-like appearance (Fig. 7e–g). Phenotypic analysis of the organoid structures generated from the control-transduced cells showed the presence of cells expressing the K14+ phenotype of BCs on the organoid surface and cells expressing the K8/K18+ phenotype of luminal

cells in the interior (Fig. 7h). In contrast, the spheroid structures produced from the EGFP-Cre-transduced primary *Blg-Cre;Brca1^f/f^;p53^+/−^* cells contained very few K8/K18+ cells (Fig. 7h). Importantly, addition of the PLK1 inhibitor to these cultures caused them to generate organoid structures grossly and phenotypically indistinguishable from those generated by the control cells (Fig. 7e–h), although their yields remained significantly lower (Fig. 7d). Taken together, these results provide further evidence that wildtype BRCA1 plays an essential role in controlling PLK1 activity which, in turn, is required to correctly orient the cell division axis and promote the acquisition of luminal features in primitive human and mouse mammary epithelial cells.

Aberrant luminal progenitor differentiation precedes triple-negative breast cancer in mammary tissues from *Blg-Cre;Brca1^f/f^;p53^+/−^* mice[14]. To access a potential association between PLK1 and breast cancer risk in human populations of *BRCA1* mutation carriers, we examined the summary results of the genome-wide association studies of the Consortium of investigators of modifiers of BRCA1/2 (CIMBA)[44]. We identified a rare variant in the first intron of *PLK1* (rs138974428; imputation $r^2 = 0.95$; Europeans minor allele frequency = 0.006) as potentially associated with reduced risk of breast cancer in carriers of pathogenic *BRCA1* variants (hazard ratio (HR) = 0.66, $P = 0.0045$) but not *BRCA2* variants (HR = 0.78, $P = 0.096$) (Figure S18a). This variant overlaps with a predicted 5'-*PLK1* enhancer in different cell types[45] (Figure S18b), which is found to be active in FACS-enriched LPs and mature luminal cells and primed in BCs and stromal cells[46] (Figure S18c). Moreover, rs138974428 was found to be potentially associated in the same direction for ER-negative breast cancer risk in the CIMBA and Breast Cancer Association Consortium (BCAC) meta-analysis results (HR = 0.74; $P = 0.00037$) without significant effect on ovarian cancer risk (Figure S18a). Thus, in the human population, variation in the enhancer region of *PLK1* appears to specifically modify breast cancer risk for *BRCA1* mutation carriers.

**Discussion**

BRCA1 has been proposed as exerting control over normal primitive mammary cell proliferation[7]. Here, we provide a direct mechanistic connection between BRCA1 function and oriented cell division in human mammary LPs and BCs. Specific modification of cancer risk by common genetic variation in *HMMR*[27] and DYNLL1 expression[28] indirectly implicates the intrinsic

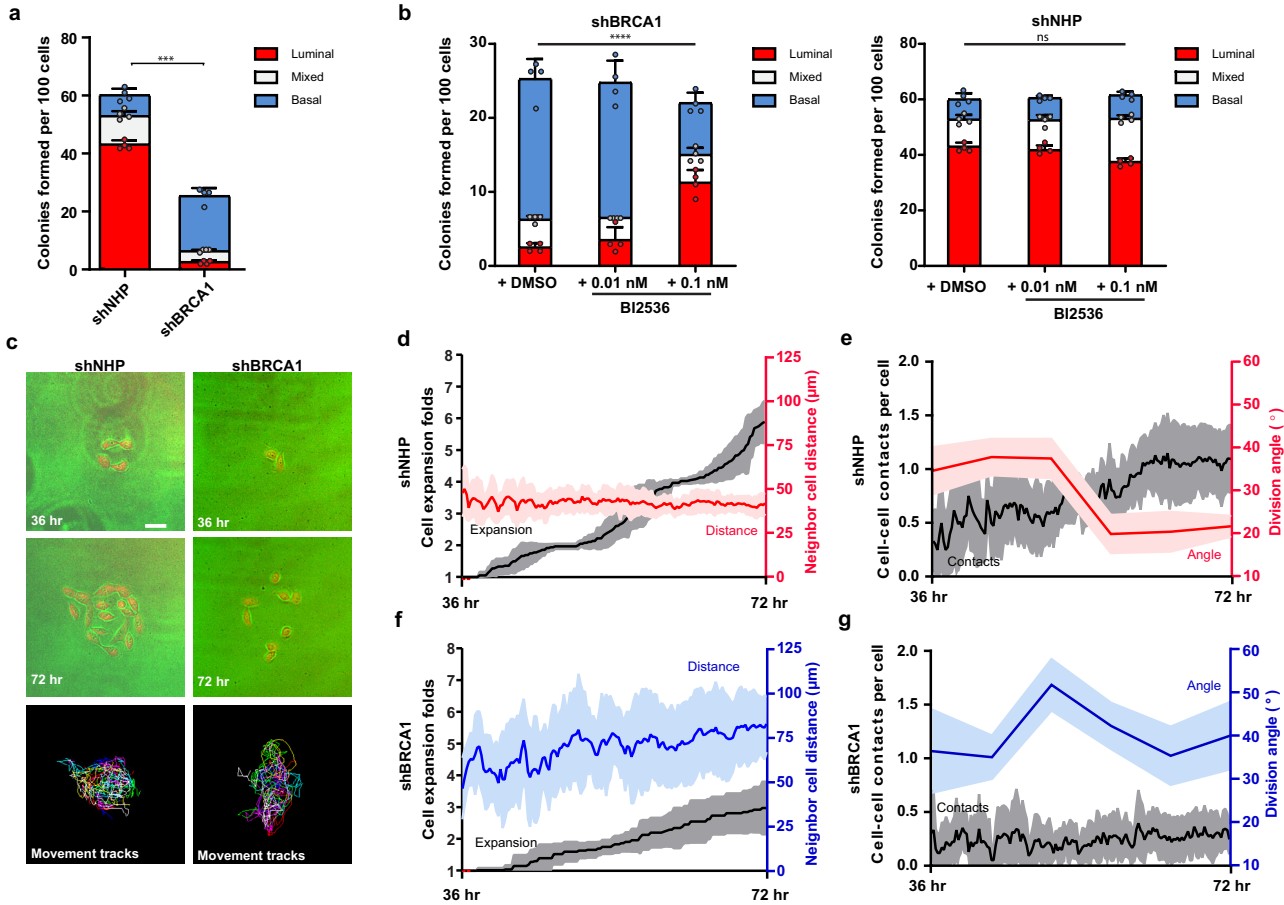

**Fig. 6 BRCA1 silencing reduces luminal properties in a PLK1-dependent manner. a** Colony structures derived from 100 MCF10A RFP-TUBA1B cells that were transduced either with non-hairpin shRNA (shNHP, hereafter control-transduced) or with BRCA1-targeting shRNA (shBRCA1, hereafter BRCA1-silenced) and classified at day 5 by the density of nuclei in the colony, as outlined in Supplementary Figure S14 (mean ± SD from four wells across duplicate experiments). ***$P = 0.0005$; two-tailed unpaired $t$-test. **b** Day 5 colony structures derived from 100 BRCA1-silenced (shBRCA1) or 100 control-transduced (shNHP) MCF10A RFP-TUBA1B cells grown in the presence of DMSO or BI2536 (mean ± SD from four wells across duplicate experiments). ****$P = 1.09E$ $-04$, ns $P = 0.5425$; one-way ANOVA. **c** Colony formation assays for control-transduced (shNHP) or BRCA1-silenced (shBRCA1) MCF10A RFP-TUBA1B cells imaged every 15 min for 36 h. 2–4 cell clusters were chosen for imaging at 36 h after seeding. Motion tracking through colony formation is indicated. Scale bar = 50 μm. **d** Cell number expansion (black y-axis, LHS) and cell distances measured by Delaunay triangulation (red y-axis, RHS) plotted through the colony formation process for control-transduced (shNHP) MCF10A RFP-TUBA1B cells. Panels **d–g** plot: mean ± SD band plotted except for division angle mean ± SEM plotted, $n = 3$ wells imaged per experiment, triplicate experiments. **e** Cell contacts per cell (black y-axis, LHS) and cell division angles relative to the cell's long axis (red y-axis, RHS) plotted through the colony formation process for control-transduced (shNHP) MCF10A RFP-TUBA1B cells. **f** Cell number expansion (black y-axis, LHS) and cell distances measured by delaunay triangulation (blue y-axis, RHS) plotted through the colony formation process for BRCA1-silenced (shBRCA1) MCF10A RFP-TUBA1B cells. **g** Cell contacts per cell (black y-axis, LHS) and cell division angles relative to the cell's long axis (blue y-axis, RHS) plotted through the colony formation process for BRCA1-silenced (shBRCA1) MCF10A RFP-TUBA1B cells.

spindle positioning pathway in *BRCA1*-mutant tumorigenesis. PLK1 is now revealed to be aberrantly activated when BRCA1 function is compromised which associates with loss of control of spindle positioning and deficient acquisition of luminal features by the progeny generated (Figure S19). Our data thus provides insight into the mechanism that links an abnormal control of the division axis of mammary cells with the phenotypic changes identified in female *BRCA1* mutation carriers and that may impact the type of tumors later produced.

The contribution of the cell division axis to the symmetric or asymmetric division of mammary progenitor cells remains poorly understood. The occurrence of asymmetric divisions in the mammary gland is supported by multiple lines of evidence[47–49], including in situ analyses of bipotent mouse mammary stem cells expressing either the K5 or the K14 promoter and found to be capable of generating both basal and luminal progeny[50]. Indeed, in the cap cells of terminal end buds, cell divisions oriented parallel to the basement membrane were found to be prone to generate myoepithelial cells, regulated by the activity of AURKA and reliant on NOTCH signaling to promote a luminal lineage[24]. Here, we find a similar phenotype in mammary epithelial cells derived from *Blg-Cre;Brca1^f/f^;p53^+/−^* mice and ascribe mutant *BRCA1*-dependent misorientation of the division axis to PLK1 hyperactivity, which is known to enhance NUMB degradation[51]. Although we do not find hyperactive AURKA in BRCA1-mutant human cells, we observe a similar correlation between asymmetric division perpendicular to organoid basement membrane and the acquisition of luminal lineage (K8/K18-positivity) in mouse MECs. Putative dysregulation of NOTCH signaling downstream[52,53] warrants further study. We also do not currently know how BRCA1 loss induces an increase in pPLK1 at the spindle pole, and future research into mechanisms that may modulate PLK1 phosphorylation or recruitment to the poles is warranted.

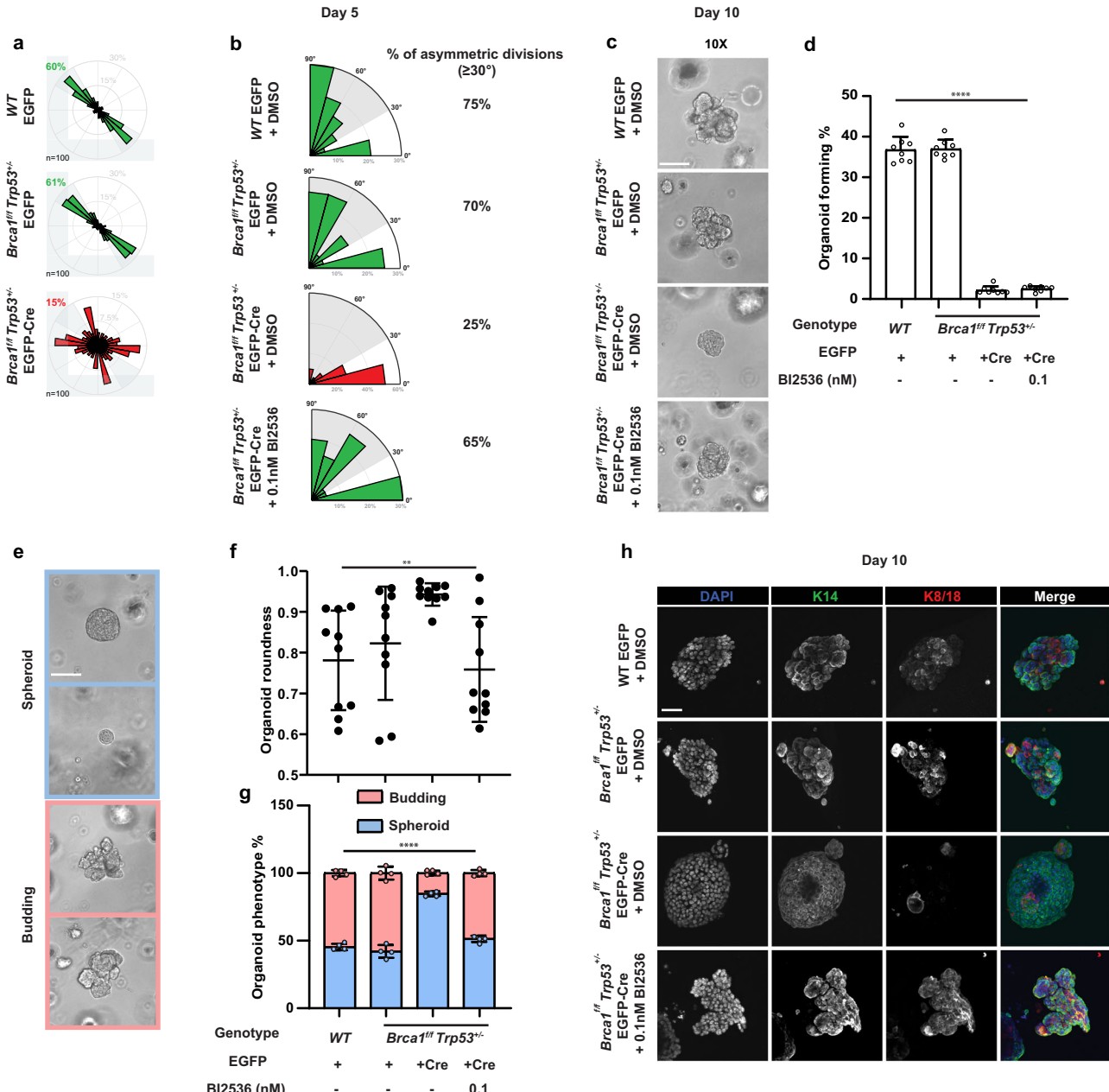

**Fig. 7 Cre-driven *Brca1* mutation alters organoid structures in a PLK1-dependent manner. a** Distribution of cell division angles in 10°-wide sectors measured at anaphase for *Brca1⁺/⁺;Trp53⁺/⁺* or *Blg-Cre;Brca1^f/f^;Trp53⁺/⁻* murine mammary epithelial cells (MECs) transduced with EGFP or EGFP-Cre lentivirus (*n* = 50 cell divisions per experiment, duplicate experiments). **b** Distribution of cell division angles in 30°-wide sectors measured in day 5 organoids generated from *Brca1⁺/⁺;Trp53⁺/⁺* or *Blg-Cre;Brca1^f/f^;Trp53⁺/⁻* murine MECs transduced with EGFP or EGFP-Cre and treated with DMSO or 0.1 nM BI2536. Angles <30° considered as parallel, symmetric divisions while angles ≥30° (highlighted in gray) considered as asymmetric divisions (*n* = 10 cell divisions per experiment, duplicate experiments). **c** Brightfield images of day 10 organoids generated from *Brca1⁺/⁺;Trp53⁺/⁺* or *Blg-Cre;Brca1^f/f^;Trp53⁺/⁻* murine MECs transduced with EGFP or EGFP-Cre and treated with DMSO or 0.1 nM BI2536. Scale bar = 100 μm. **d** Organoid forming percentage measured at day 10 and generated from *Brca1⁺/⁺;Trp53⁺/⁺* or *Blg-Cre;Brca1^f/f^;Trp53⁺/⁻* murine MECs transduced with EGFP or EGFP-Cre and treated with DMSO or 0.1 nM BI2536 (mean ± SD, *n* = 4 fields imaged per experiment, duplicate experiments). ****$P$ < 1E−15; one-way ANOVA. **e** Brightfield images of day 10 organoids classified as spheroid or budding structures. Scale bars = 100 μm. **f** Organoid roundness measured at day 10 and generated from *Brca1⁺/⁺;Trp53⁺/⁺* or *Blg-Cre;Brca1^f/f^;Trp53⁺/⁻* murine MECs transduced with EGFP or EGFP-Cre and treated with DMSO or 0.1 nM BI2536 (Violin plot, *n* = 5 organoids imaged per experiment, duplicate experiments). **$P$ = 0.0042; one-way ANOVA. **g** Organoid phenotype percentage measured at day 10 and generated from *Brca1⁺/⁺;Trp53⁺/⁺* or *Blg-Cre;Brca1^f/f^;Trp53⁺/⁻* MECs transduced with EGFP or EGFP-Cre and treated with DMSO or 0.1 nM BI2536 (mean ± SD, *n* = 2 fields per experiment, duplicate experiments). ****$P$ = 4.37E−10; one-way ANOVA. **h** Immunofluorescence analysis of K14 and K8/18 in day 10 organoids generated from *Brca1⁺/⁺;Trp53⁺/⁺* or *Blg-Cre;Brca1^f/f^;Trp53⁺/⁻* murine MECs transduced with EGFP or EGFP-Cre and treated with DMSO or 0.1 nM BI2536. Scale bar = 40 μm.

*Brca1* deletion in MECs impaired oriented cell divisions and, via symmetric cell divisions parallel to the basement membrane, generated spheroids that appeared myoepithelial lineage-restricted (K14+). This restricted output of K14+ cells is similar to the alpha-smooth muscle actin-positive single layer structures generated by BRCA1-silenced human mammary cells grown in humanized immunodeficient mice[8]. Thus, our findings that BRCA1 function is critical to the intrinsic spindle positioning pathway and asymmetric cell divisions suggests this deficit may also explain the deficient or abnormal luminal differentiation[12,14,54,55] that appear to characterize mammary tissues in *Brca1*-mutant mouse models and female carriers of *BRCA1* mutations.

We find single causal mutations in *BRCA1* ($n = 7$) distributed between exon 2 (185delAG) and exon 20 (5283insG) in MCF10A cells, or exon 2 ($n = 2$, 185delAG or 185insA) and exon 11 (truncating c.1687C>T) in human carriers, cause the loss of control of the cell division axis. But, single causal mutations ($n = 2$) in exon 24 permit cell division axis control in MCF10A cells and do not alter the intensity of phosphorylated PLK1 at metaphase spindle poles. This pattern for loss of cell division axis control aligns with recent domain-specific breast cancer risk estimates for missense variants in *BRCA1*; there is an increased risk associated with protein-truncating variants and variants located in the regions encoding the RING domain and the BRCT1 domain[3]. In our study, radiation-induced DNA damage was permissive to cell division axis control but causal mutations in the RING domain were disruptive, implicating E3 ubiquitin ligase activity as a critical process. BRCA1-BARD1 activity controls HMMR abundance[26,27] that lies upstream of PLK1 activation[21]. Indeed, overexpression of HMMR is sufficient to misorient cell division on L-patterns[17]. However, R71G lies in the RING domain and is predicted to cause aberrant splicing of the transcript and result in a truncated protein[56]. But, MCF10A R71G/+ cells did not differ from parental cells in their control of the cell division axis or in the intensity of pPLK1 at spindle poles. Thus, our R71G/+ engineered cells may not reconstitute the pathogenic attributes of this mutation. But, this is a common outcome for R71G/+ engineered cells[57–59].

This study compares the clonal growth and cell division kinetics of mammary cells isolated from non-involved tissues from female *BRCA1* mutation carriers or non-carriers. Interestingly, FACS-enriched LP and BC populations isolated from non-carrier tissues showed significant ($p < 0.01$) differences in γH2Ax foci that persisted 24 h after X-irradiation, indicating these sub-populations may possess differential DNA damage responses[60,61]. In the mutation carriers, BRCA1 foci formation was significantly reduced, but without evidence of changes to basal levels of DNA damage. Nevertheless, the mutant cells were deficient in their responses to X-radiation possessing a reduced clonogenic activity, implicating BRCA1 in the supervision of genotoxic stress. While small-molecule inhibition of hyperactive PLK1 rescued the cell division axis and promoted the acquisition of luminal features in all of these examples of BRCA1 deficiency, PLK1 inhibition was not able to restore cell outputs. Therefore, supervision of genotoxic stress by BRCA1 may be independent of its regulatory role on PLK1 activity, which is required to ensure the orientation of cell divisions and the subsequent display of luminal features.

Mammary glands isolated from *BRCA1* mutation carriers are characterized by a higher frequency of cells identified by their RNA-sequencing profiles as LPs[13]. This is not universally detected by scRNA sequencing[62] and was not obvious in the donor samples analyzed here (Figure S1a), but aligns with prior findings of a high frequency of LPs in premalignant tissues[10,12] and suggests non-cancer mammary tissues isolated from *BRCA1* mutation carriers show perturbed luminal differentiation. In addition, temporal single-cell RNA-sequencing analysis in BLG-Cre *Brca1^{f/f} Trp53^{+/−}* mice reveals a deficit in LP differentiation that precedes mammary tumorigenesis[14]. The LP fraction has been inferred to be the cell of origin for BRCA1-associated breast cancers[10,12]. Consistently, we find freshly-isolated progenitor-enriched cells from *BRCA1* mutation carriers are more highly mitotic and less likely to undergo cell death when grown ex vivo. In addition, these primary *BRCA1* mutant cells are deficient in their ability to orient the cell division axis, which is also observed in primary cells isolated from BLG-Cre *Brca1^{f/f} Trp53^{+/−}* mice; in the latter case, the deficit in control of cell division axis restricts the acquisition of features of mature luminal cells in organoid cultures. However, the *BRCA1* mutant/+ MCF10A cell lines, primary *Brca1*-mutant mouse mammary cells, and limited numbers of patient-derived *BRCA1*-mutant organoids utilized in this study may not capture the full spectrum of *BRCA1* mutations identified in human carriers. Mutations arising in specific domains, for example, may permit division axis control. Thus, future study with a larger collection of high-risk mammary tissue is warranted to better dissect the individual impact of diverse *BRCA1* mutations on the orientation of cell division and progeny cell function.

Overall, our data are consistent with a molecular axis wherein *BRCA1* mutations that disrupt E3 ubiquitin ligase activity promote hyperactivation of PLK1 on prometaphase spindle poles, potentially through the stabilization of HMMR[26] or other intermediates[63]. Mitotic hyperactivity of PLK1 results in a consequent loss of control of the cell division axis and reduced ability for progeny cells to acquire the features of mature luminal cells. Together, these data may mechanistically explain the premalignant phenotypic changes in the breast, and the type of tumors later produced, identified in the female *BRCA1* mutation carriers.

## Methods

### Ethics statement
Non-carrier breast tissues were obtained from consenting donors undergoing reduction mammoplasty surgery and used according to protocols approved by the University of British Columbia Research Ethics Board protocol H19-03798 (PI Eaves). *BRCA1*-mutated breast tissues and de-identified data were obtained from the Mayo Clinic's Living High-risk Breast Biobank. Study samples were approved by Mayo Clinic's Institutional Review Board (Mayo-IRB) protocol # 16-008725 (PI Kannan). Study samples were approved by Mayo-IRB under protocol # 17-010922. All animal experiments were carried out in the University of Barcelona-Bellvitge animal facility, under the Generalitat de Catalunya license authority (reference 9774) and approval of the IDIBELL University of Barcelona-Bellvitge Ethics Committee (PI Pujana).

### Cell culture
All cell lines were grown at 37 °C in a 5% (v/v) $CO_2$ incubator. Cells were serial passaged at 80% density, and re-seeded at 15–20% density. Parental MCF10A cells were directly obtained from Dr. J Brugge (Harvard University, Boston MA). All MCF10A sublines were cultured in Brugge media, which is DMEM/F12 (1:1) media supplemented with 5% horse serum, 20 ng/mL epidermal growth factor (EGF), 10 μg/mL insulin, 0.5 μg/mL hydrocortisone, cholera toxin 100 ng/mL and 10 μg/mL insulin.

MCF10A sublines encoding different heterozygous *BRCA1* mutations (*Exon 2_3 deletion, Exon 10 deletion, C61G, C64R, D67Y, R71G, L246V, S316G, Q356R, and I379M*) were provided by Dr. B. Park (The Johns Hopkins University, Baltimore, MD, USA). For these cells, gene targeting of the *BRCA1* gene was carried out using distinct recombinant Adeno-associated virus vectors for each of the respective BRCA1 exon as described[37]. MCF10A *BRCA1 185delAG/+* were purchased through Horizon Discovery (HD 101-018)[64]. MCF10A RFP-TUBA1B were purchased from Sigma-Aldrich (CLL1039).

For synchronization of cells in mitosis, cells were treated with 200 ng/mL nocodazole (Sigma) for 16 h. Cells were collected or fixed at designated times after synchronization in accordance with each experiment. The efficiency of cell synchronization was assessed based on the percentage of mitotic cells scored under light microscopy.

### Isolation and culture of primitive human mammary cells
Highly purified sub-populations of human mammary epithelial cells were isolated from non-carrier and *BRCA1*-mutated breast tissues as previously described[29,60]. Tissue was minced with scalpels and dissociated in DMEM/Ham's F12 media (1:1, STEMCELL Technologies) with 2% BSA (Gibco), 300 U/mL collagenase (Sigma) and 100U/Ml

hyaluronidase (Sigma) at 37 °C for 18 h. Then, the dissociated mammary organoids were centrifuged for 4 min at 80 × g. The dissociated mammary organoids were cryopreserved in liquid nitrogen in FBS supplemented with 6% DMSO prior to use. To isolate primitive mammary epithelia cells, the cryopreserved mammary organoids were thawed and washed with Hank's Balanced Salt Solution with 2% FBS (named HF). The organoids were dissociated in 2.5 mg/mL trypsin with 1 mM EDTA (STEMCELL Technologies) and 5 mg/mL dispase (STEMCELL Technologies) supplemented with 100 µg/mL DNase I (Sigma). The cells were washed with HF and passed through a 40 µm cell strainer to obtain a single-cell suspension. The resulting single-cell suspension of DAPI−CD45−CD31− cells were FACS-purified into 4 subpopulations based on their expression of EpCAM and CD49f using a FACSAria III cell sorter (BD Biosciences) (Figure S20). Basal cells (BCs) were purified by their EpCAM$^{low}$CD49f$^+$ phenotype. Luminal progenitors (LPs) were purified by their EpCAM$^{high}$CD49f$^+$ phenotype. Luminal cells (LCs) were purified by their EpCAM$^{high}$CD49f$^-$ phenotype and stromal cells (SCs) by their EpCAM$^-$CD49f$^-$ phenotype.

Isolated human mammary cells were cultured in 5% FBS modified SF-7 medium[60], consisting 1:1 DMEM/F12 (STEMCELL Technologies) supplemented with 0.5 µg/mL hydrocortisone (Sigma), 1 µg/mL insulin (Sigma), 10 ng/mL EGF (Sigma), and 10 ng/mL cholera toxin (Sigma).

**2D in vitro colony-forming cell (CFC) assay of primary mammary cells and MCF10A cells.** Primary mammary cells (500 cells) and irradiated NIH-3T3 fibroblasts (15,000 cells, obtained from the Eaves Lab) were co-cultured in individual wells in a 24-well plate for 8 days in SF-7 medium supplemented with 5% FBS as previously described[29]. Medium was refreshed every other day. MCF10A cells were seeded in a 6-well plate for 5 days in Brugge media as described[17].

**Irradiation of cells.** Cells were irradiated either in a plate or on coverslips at 50% to 70% confluency at room temperature using an X-ray irradiator, RS-2000 PRO Biological System from Radsource. Doses used with programs set at 160 kV (volts) and 25 mA (amp).

**Cytotoxicity assay for MCF10A cells.** The cell cytotoxicity assay kit was purchased from Abcam (ab 112118). This cytotoxicity assay uses a proprietary water-soluble dye, which has different absorption spectra upon cellular reduction. MCF10A cells were seeded in a 96-well plate at 10% confluency prior to inhibitor treatment. After 48 h incubation with inhibitors, the cells were cultured in medium supplemented with 1/5 volume of ratio assay solution for 4 h at 37 °C. Then, the absorbance of the cell cultures was measured at 570 and 605 nm wavelengths with an EnSpire™ multilabel reader (PerkinElmer). The background absorbance of the blank wells was corrected from the values for the wells containing the cells. The ratio of OD570 to OD605 was calculated to represent the cell viability in each well.

**Analysis of the cell division axis.** L-shaped micropatterns (CYTOO™ plate) were used to examine cell division angles for individual, isolated cells. Each well of a 96-well plate holds 4000 micropatterns, which are arrayed over the glass surface at a pitch of 80 µm. Wells were coated with a buffer containing fibronectin (isolated from bovine plasma, Sigma F4759-1MG), laminin (Sigma L2020), or collagen (Type I bovine collagen, STEMCELL Technologies 04902) at 20 µg/mL in PBS without Ca$^{2+}$, and Mg$^{2+}$ (Gibco, 14190-094). The coating buffer was added to each well and incubated at room temperature for two h. The adhesion molecules were rinsed by washing with PBS five times.

Culture medium was added to the wells and incubated for 1 h at 37 °C before the cells were seeded on the micropatterns. Next, the cells were seeded vertically to allow cells to sediment homogeneously in the bottom of the wells. Afterward, the culture plate was placed directly into the incubator for 8 h allowing the cells to spread out completely on the micropatterns. Live-cell imaging was initiated and continued for 12 h. 3000 cells were seeded in each well of a 96-well plate at a density of 15,000 cells/mL.

For the analysis of division angles on micropatterns, 50 mitotic cells per well were analyzed. Only cells that had spread out on the L-shape micropattern were measured during their subsequent mitosis. The cell division angle was determined in anaphase based on the position of the mitotic spindle (or the anaphase angle of the dividing cell) with respect to the micropattern.

**Definition of abnormal nuclei.** Abnormal nuclei included cells possessing either a micronucleus or nuclear budding[65]. Micronuclei were identified as the positive Hoechst staining separate from the nuclei (Figure S2a). Nuclear buds were defined as the Hoechst-positive bubbles that emerged from the nucleus (Figure S2a).

**Segmentation analysis of BRCA1 and γH2AX foci.** BRCA1 and γH2AX foci were segmented using "Find Maxima" (Fiji ImageJ) in combination with a DAPI mask, which was generated with default threshold followed by Analyze Particle with Outlines. ROI manager was employed to measure the nuclear BRCA1 and γH2AX in each cell with the DAPI mask.

**Classification of *BRCA1* variants.** *BRCA1* variants were classified based on BRCAshare guidelines[66]. Pathogenic variants were defined as variants associated with higher risk of acquiring breast cancers. Benign variants were defined as variants associated with lower risk of acquiring breast cancers.

**Generation of MCF10A cell lines with *BRCA1* variants using Cas9-initiated homology-directed repair**

*Guide RNA (gRNA) and single-stranded DNA (ssDNA) design.* gRNAs were designed using the University of California San Francisco Genome Browser and ordered from Integrated DNA Technologies (Table S2). When synthesizing oligonucleotides, CAACG was added to the 5′ end of the forward oligonucleotide, and AAAC was added to the 5′ end of the reverse oligonucleotide. A single C was added to the 3′ end. These sequences are overhangs for ligation to BbsI sites in PX458. gRNA sequences were listed in Table S2.

*pSpCas9 (BB)-2A-GFP (PX458) cloning.* PX458 plasmid expresses a chimeric gRNA plus EGFP and human codon-optimized Cas9[67] and was gifted from Dr. Lim (Addgene #48138). Each gRNA oligonucleotide pair (5 µM) was mixed and incubated at 95 °C for 10 min, and then the gRNA mix was cooled down to room temperature for 30 min. pX458 was digested with BbsI for 1.5 h at 37 °C. The digested pX458 was incubated with annealed oligonucleotides and T4 DNA ligase for 1.5 h at 37 °C. After ligation, X458 with the desired gRNA insertions was transformed into DH5α (Thermo Fisher Scientific).

*Transfection and single-cell sorting.* pX458 with the desired gRNA was transfected into cells through Lipofectamine 3000 (Thermo Fisher Scientific). GFP-positive transfected cells were sorted as single cells the following day to obtain the clones for screening for the desired genetic editing.

*Polymerase chain reaction (PCR).* Genomic DNA was extracted with the DNeasy extraction kit (Qiagen). Sample concentrations were measured with a NanoDrop (Thermo Fisher Scientific). PCR cycling conditions were: 95 °C for 5 min (initial holding stage), followed by 40 amplification cycles at 95 °C for 60 s (denaturing stage) and 60 °C for 30 s (annealing/extension stage), followed by 72 °C for 5 min (final extension stage), followed by the generation of the melting curve, and held at 4 °C. Primers were ordered from Integrated DNA Technologies and are listed in Table S2. The putative clone was confirmed with Sanger sequencer.

**Lentiviral production and transduction.** Sequences encoding short hairpins against BRCA1 (shBRCA1 #1, shBRCA1 #2) (Sigma-Aldrich) or non-hairpin (shNHP) sequences (Addgene #10879) were produced and used to transduce cells as previously described[27]. Briefly, HEK-293FT cells (R70007, Thermo Fisher) and were maintained in 10% fetal bovine serum (FBS) Dulbecco's modified eagle medium (DMEM) at 37 °C, in a 5% CO$_2$ incubator and split at 70–80% confluence. Lentivirus was produced by packaging the target shRNA plasmid with a packaging plasmid (psPAX2) and an envelope plasmid (pMD2.G), as described[27]. The targeting sequences are provided in Table S2.

To produce one 10 cm plate of lentivirus, HEK-293FT cells were seeded in 10 mL of 10% FBS/DMEM at the density of 4–4.5 million cells per plate. The cells were cultured in DMEM media supplemented with 10% FBS and were incubated overnight. Fresh media (6 mL) was added the next morning. Four hours after replacing the media, the cells were transfected at about 70% confluence. Transfection cocktail was prepared as the mixture of 10 µg vector, 7.5 µg psPAX2, 2.5 µg pMD2.G, and 87 µl calcium solution in a polypropylene tube. The transfection cocktail and the HEPES-buffered saline were added dropwise, with gentle mixing, and incubated for 5 to 15 min at room temperature. The mixture was added dropwise to a 10 cm plate containing HEK-293FT cells. The plate was incubated at 37 °C, 5% CO$_2$ incubator for 12 to 15 h within the designated lentiviral facility room. Media was changed in the morning (5 mL of fresh 10% FBS/DMEM) to remove the transfection reagent and cells were allowed to recover for 24 h.

Virus was harvested, transferred to a polypropylene storage tube, and stored at 4 °C. Five mL of fresh media was added to the dish for another harvest the following morning, and cells were incubated at 37 °C, 5% CO$_2$ incubator for another 24 h. Virus was harvested again and both harvests were centrifuged at 500 × g to pellet debris. Viral supernatant was filtered into ultracentrifuge tubes (Beckman #326823) through a 0.45 µm low protein binding Millipore filter to remove cells and debris. PBS was added to the tubes for ultracentrifugation, leaving no more than 0.5 cm of space at the top to prevent the tubes from collapsing at high speed. The tubes were balanced so that the weight difference was <0.005 g. Tubes were placed in a pre-cooled rotor (Sw32Ti) and spun at 175,000 × g for 1 h and 40 min at 4 °C, and the virus was retrieved in 70 µL PBS.

Lentiviral transduction was performed as previously described[27]. Non-hairpin shRNA (shNHP) served as a negative control. Cells were seeded in 6-well plates to achieve 80% confluence, and fresh media was changed 4 h prior to transduction. Cells were incubated with the viral supernatants overnight. In the morning, the media was changed. Protein expression was measured 72 h post transduction with western blot analysis or by immunofluorescence analysis. Cells were fixed for

analysis 72 h post transduction or live cell imaging was conducted between 72 and 96 h post transduction.

**Transfection and immunoblot analysis.** Cells were grown to 90% confluence in 6-well or 10 cm plates prior to transfection. Fresh media was changed 4 h prior to transfection. To transfect immortal cell-lines, each siRNA construct was incubated in 250 μl OPTI-MEM for 15 min at room temperature to make up a final concentration of 40 nM. Concurrently, 5 μl Lipofectamine 3000™ was added to 250 μl OPTI-MEM. For plasmid transfections, 2 ng of each plasmid was added. After the incubation period, the two mixtures were added together and incubated for 15 min at room temperature and then added dropwise to cells. After 16 h, cells were washed twice with PBS and re-seeded for analyses. Expression of mRNA or protein was measured 72 h post transfection. Cells were fixed for analysis 72 h post transfection or living cells were imaged between 72 and 96 h post transfection.

Western blot analyses were performed as previously described[68,69]. Once cultures reached 70–90% confluence, cells were collected after treatment with 0.25% trypsin and then were washed once with PBS. Cells were counted using a hemocytometer. $4 \times 10^6$ cells were pelleted and lysed in 200 μL of RIPA buffer with protease inhibitor (Roche 04-693-124-001). Cells were lysed for 30 min at 4 °C and then mechanically disrupted by passing the lysates through a 25G 5/8 syringe ten times. The lysate was spun at 4 °C for 15 min in the refrigerated bench top centrifuge at $16,000 \times g$, and supernatants were transferred to new tubes. Protein concentration was measured with BCA protein assay kit as per manufacturer's instructions (Thermo Fisher Scientific). Each reaction was run in triplicate.

Six bovine serum albumin (BSA) standards and one lysis buffer only negative control were used to obtain the protein concentration of the samples. A standard curve was plotted with absorbance reading at OD562 on the y-axis and standard concentration on the x-axis, and this equation was used to determine protein concentration.

Proteins were denatured by adding 3X loading buffer to the samples and boiled at 95 °C for 5 min. Subsequently, proteins were resolved by sodium dodecyl sulfate-polyacrylamide gel electrophoresis (SDS-PAGE) using a 4% stacking gel with a 6% separating gel for BRCA1 and an 8% separating gel for GAPDH. GAPDH levels were determined on the same blot as a control for equal loading.

Proteins were transferred onto polyvinylidene fluoride (PVDF) membrane using the Bio-Rad Semi dry transfer apparatus, after the membrane was soaked in methanol for 3 min. The transfer was performed at 4 °C overnight using 28 V. After washing once with TBST buffer, the membrane was blocked with 3% milk/TBST for 1 h at room temperature. The membrane was washed four times in TBST for 5 min each prior to incubation with primary antibodies (Table S3). Antibodies were diluted in 3% BSA/TBST and incubated overnight at 4 °C on a shaker. Blots were washed four times with TBST for 5 min each and then incubated with horseradish peroxidase (HRP)-conjugated secondary antibodies (Sigma) for 1 h at room temperature. A final wash was done with TBST (4 times for 5 min each). Proteins were detected using an ECL Kit (Thermo Fisher Scientific).

**Liquid chromatography with tandem mass spectrometry (LC-MS/MS): sample preparation.** Five biological replicas were prepared for non-hairpin shRNA and shRNA targeting BRCA1 conditions respectively, resulting in a total of 10 samples. Each sample consisted of 3 million cells harvested from cells at 80% confluency. Samples were prepped together in a single batch. In case of a technically underperforming sample, five replicas were used to allow for a sample removal if needed, while still providing statistical power to confidently identify differentially expressed proteins. For comprehensive analysis, we selected label-free data-independent acquisition (DIA) analyzed by directDIA™ in Spectronaut© software.

Samples were prepared as previously described[70]. Briefly, cells were lysed in buffer containing 1% SDS (Fisher BioReagents, Pittsburgh, Pennsylvania, USA), 1X Pierce protease inhibitor (Thermo Fisher Scientific, Waltham, Massachusetts, USA) in 50 mM HEPES (pH 8.5), followed by 5 min incubation at 95 °C and 5 min on ice. The samples were sonicated for 30 s two times, followed by treatment with benzonase (EMD Millipore/Novagen, Country) at 37 °C for 30 min to shear chromatin. Following benzonase treatment, each sample was reduced with 10 mM Dithiothreitol (DTT) (37 °C, 30 min) and alkylated with 40 mM Chloroacetamide (CAA) (30 min in the dark) and quenched in 40 mM DTT for 5 min at room temperature.

Lysates were cleaned using single-pot solid-phase-enhanced (SP3) bead technique using hydrophilic and hydrophobic Sera-Mag Speed Beads (GE Life Sciences, Issaquah, Washington, USA). Proteins were bound to paramagnetic beads with 100% ethanol (80% v/v), incubated for 18 min at room temperature, and washed twice with 90% ethanol using magnetic isolation. Beads were then resuspended in 30 μl 50 mM HEPES, pH 8.0, and incubated with sequencing-grade trypsin (Promega Madison, Wisconsin, USA) at 1:50 protein ratio for 16 h at 37 °C, and afterward acidified to pH 3–4 with formic acid. Peptide digests were washed on BioPure (Nest Group Inc.) C18 spin columns with 0.1% trifluoroacetic acid (TFA), eluted with 60% acetonitrile in 0.1% FA and dried in a Speed Vac. Dried samples were resuspended in 0.1% FA. As required for downstream analysis, iRT peptides (Biognosys, Schlieren, Switzerland) were added at a dilution of 1:30.

**LC-MS/MS acquisition.** Peptides were analyzed using a PharmaFluidics 50 cm uPAC™ (ESI Source Solutions, Woburn, MA, USA), column maintained at 50 °C on an Easy-nLC 1200 connected to a Q Exactive HF mass spectrometer (Thermo Scientific). The peptides were separated over a 3-h gradient consisting of Buffer A (0.1% FA in 2% acetonitrile) and 2 to 80% Buffer B (0.1% FA in 95% acetonitrile) at 300 nL/min. Samples were randomized across conditions and biological replica. The data-independent acquisition (DIA) method consisted of a MS1 scan from 300 to 1650 $m/z$ (AGC target of 3e6 or 60 ms injection time), and resolution of 120,000. DIA segment spectra was acquired with a twenty-four-variable window format (AGC target 3e6, resolution 30,000, auto for injection time), and NCE 25.5, 2, 30.

**LC-MS/MS analysis.** The raw files were analyzed by directDIA™ (no spectral library) using Spectronaut Pulsar X (Biognosys, Schlieren, Switzerland) with a human FASTA file from UniProt (downloaded 20200309). This FASTA file includes common contaminants. In addition, a FASTA file for iRT peptides, provided by Biognosys, was included in the search. Search was performed using the factory settings including specificity for Trypsin, Carbamidomethyl (C) as a fixed modification, and Acetyl (protein N-term) and Oxidation (M) as variable modifications. Precursor, peptide, and protein identifications threshold was set to 1%. Protein intensities were normalized in Spectronaut using the "global" setting, which normalizes by median protein intensity per sample. Missing values were eliminated, i.e., only proteins identified in all samples were retained for downstream analysis, thus providing the most robustly quantified proteins. Differential expression analysis was performed using student's t-test and significance was determined by a log2 fold change >1 and FDR < 1%. Gene ontology analysis was done in g:Profiler[71].

**Isolation and culture of primary mouse mammary epithelial cells and organoid cultures.** The $Trp53^{tm1Brd} Brca1^{tm1Aash/F22-24}$ Tg (BLG-cre) 74Acl/J mouse strain was purchased from The Jackson Laboratory (012620). All mice were maintained in a temperature-controlled room (21 °C) with a 12-h light-dark cycle with 40–60% humidity. Mammary glands from 6-week-old female mice were dissected out as organoids (around 1 $cm^3$) and were cryopreserved in 10% DMSO/FBS in liquid nitrogen for long-term storage. Upon thawing, cryopreserved mammary organoids isolated from 6-week old mice were cut into small pieces and dissociated into single-cell suspensions with the gentleMACS dissociator (Miltenyi Biotec). After single-cell suspension was obtained, cells were cultured with Epi-Cult Plus (STEMCELL Technologies) on collagen-coated plates (Corning) and the medium was refreshed every other day. An aliquot of 5500 mammary cells was mixed with 50 μl Matrigel (Invitrogen) and 40 μl of the mixture was dispensed in the center of a well of a warm 24-well plate. The cell and matrigel mixture solidified for 15 min at 37 °C. One ml of mouse mammary organoid growth medium (STEMCELL Technologies) was added to the well. Medium was refreshed 3 times a week.

**Microscopy.** For studies of living cells, cells were seeded in 96-well plates and incubated overnight then stained with Hoechst (1 μg/μl). Cells were imaged every 5, 10, or 15 min for up to 24 h at 37 °C in a 5% $CO_2$ environmental chamber using an ImageXpress Micro High Content Screening System (Molecular Devices Inc.) and analyzed with MetaXpress 5.0.2.0 software.

For confocal microscopy, fixed cells were imaged using a ×60 oil objective with a 1.2 numerical aperture on an Olympus Fluoview FV10i (Olympus) confocal microscope. Image z-stacks consisted of 3 to 25 optical sections with a spacing of 0.5 μm through the cell or 2 μm through organoid volume. Images were processed and analyzed using the Olympus Fluoview software. Antibodies are listed in Table S3.

For immunofluorescence analyses, cells were fixed, blocked, and antibodies were incubated and washed as previously described[17]. Coverslips were mounted with ProLong Gold Antifade Reagent containing DAPI (Invitrogen). Cells grown in matrigel were fixed in cold methanol for 30–60 min.

**OncoArray complete summary results.** CIMBA results are provided for SNPs genotyped on the OncoArray or the iCOGS array and after quality control, for SNPs imputed based on the 1000 Genomes Project Phase 3 reference panel. The risk analyses were as described[72].

**Statistics and reproducibility.** Data are expressed as mean ± standard error of the mean. Statistical analysis was performed using the unpaired two-tailed Student's t-test or an unpaired one-way ANOVA followed by Dunn's Multiple Comparison test with the following exceptions: paired two-tailed Student's t-test was used for comparison of primary cell data, including CFC, multipolar mitosis, and mitotic outcomes. The results were considered significant at $*P < 0.05$, $**P < 0.01$, $***P < 0.001$. All experiments, including western blotting, IF, IC50 measurements, and organoid cultures, were representative of at least two independent repeats to confirm reproducibility. Data are presented as the mean ± SD unless otherwise noted.

**Reporting summary**. Further information on research design is available in the Nature Research Reporting Summary linked to this article.

## Data availability

The mass spectrometry data have been deposited to the Proteome Consortium via the MassIVE partner repository and can be accessed at "PXD026626". All other relevant data supporting the key findings of this study are available within the article and its Supplementary Information files or from the corresponding author upon reasonable request. Source data are provided with this paper.

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

## Acknowledgements

We thank Dr. John Stingl for assistance with the mouse organoid cultures and helpful discussions. We thank Dr. Martin Hirst for assistance with the analysis of rs138974428 and PLK1 enhancer in mammary epithelial subpopulations. We thank the BCAC and CIMBA consortia for making the summary statistics of genome-wide association studies available to the public. This study was partially funded by the Canadian Institutes of Health Research (CIHR 169111 and CIHR CEEHRC Phase II EP2-120591), the Canadian Cancer Research Institute (CCSRI 21296), the Canadian Cancer Society (BC-RG-16), the Michael Cuccione Foundation for Childhood Cancer Research, and the BC Children's Hospital Research Institute. Z.H. and A.L. received University of British Columbia Four Year Doctoral Fellowships. N.K. was supported partly through Eagles Foundation Rochester Minnesota and Mayo Clinic Breast Cancer SPORE (CA116201-12CEP). S.T. received a CIHR Banting and Best Doctoral Studentship. The CIMBA data management and data analysis were supported by funders as listed[72,73].

## Author contributions

C.A.M. and Z.H. conceived and designed the study. Z.H. performed most experiments, Z.H., R.G., K.C., and J.B. performed data analysis, A.L. and P.L. performed proteomics analysis, M.A.P. performed bioinformatics analysis, S.T., S.M.M.A., M.A.P., N.K., and C.E. provided cells and reagents. C.A.M. oversaw the study, and C.E., C.A.M., and Z.H. wrote the manuscript. All authors approved the final manuscript.

## Competing interests

The authors declare no competing interests.
