## [Peer Review File · Nature Communications]

REVIEWER COMMENTS

Reviewer #1 (Remarks to the Author):

This manuscript aims to better understand how disease-linked BRCA1 mutations impact cell biology. In particular, the authors want to determine the source of the luminal differentiation abnormalities observed in patients and mice with BRCA1 mutations. Through analysis of both patient-derived cells and edited human cell lines, they find that disease-linked BRCA1 mutations cause defects in the orientation of the mitotic spindle. They propose that this is due to reduced BRCA1 expression, as BRCA1 mutants exhibit reduced BRCA1 foci formation, and spindle misorientation can be phenocopied by knocking down BRCA1 with shRNAs. Proteomics and western blot analysis of BRCA1 mutant cells reveals an increase in expression and activity of the mitotic kinase PLK1, a known regulator of spindle orientation. Inhibition of PLK1 activity in BRCA1 mutant cells not only rescues defects in spindle positioning, but also rescues the loss of luminal cell markers observed in BRCA1 mutant cell lines and organoids. Based on this data, the authors propose that reduced BRCA1 expression leads to PLK1 upregulation and activation, which then results in spindle misorientation and subsequent defects in luminal differentiation. Importantly, they propose that these phenotypes are independent of the role of BRCA1 in DNA damage repair.

Overall, this manuscript advances the field of BRCA1 biology; however, it is important to point out that several findings are not completely novel. A previous paper from the authors already demonstrated that BRCA1 depletion causes spindle orientation defects through HMMR stabilization, and previous work from other labs demonstrated that BRCA1 and HMMR regulate PLK1 activity, and that PLK1 is important for spindle positioning. The most novel aspects of this paper are 1) that the authors connect these observations together in the context of disease-linked BRCA1 mutations, and 2) that the spindle positioning and luminal differentiation functions of BRCA1 are both dependent on PLK1, but are independent of BRCA1's DNA damage repair role.

Before this paper is accepted to Nature Communication, several issues should be addressed to better clarify and support these findings.

Major points

- A main conclusion of the paper is that BRCA1's role in orienting the mitotic spindle is independent of its role in DNA damage repair. One way this was demonstrated is MCF10A cells were irradiated to induce DNA damage and subsequent imaging revealed no change in spindle orientation. However, imaging occurred only 24 hrs after irradiation. This does not allow much time for chromosome bridges or changes in ploidy to manifest and potentially impact spindle positioning, as suggested by the authors. Therefore, this experiment should be repeated, and spindle positioning should be assessed at longer time points after irradiation (48 and 72 hrs) to allow ample time for cells to go through mitosis and generate bridges and/or aneuploidies. Additionally, BRCA1 mutant cells should also be assessed +/-IR to determine if their existing spindle orientation defects are exacerbated by additional DNA damage.
- It is known that BRCA1 affects centrosome function, which could then cause some degree of spindle assembly checkpoint activation. If cells are delayed in mitosis, they will accumulate mitotic proteins such

as PLK1 and Aurora B, as shown here by mass spec. It would therefore be important to determine if overriding the SAC with an MPS1 inhibitor has the same effect as inhibition of PLK1.

- The authors use BRCA1 foci as a proxy for BRCA1 expression. However, a decrease in BRCA1 foci could mean a decrease in protein expression or, alternatively, a defect in protein localization. The authors must western blot for BRCA1 in the BRCA1 mutant cells to directly determine how the mutations affect BRCA1 expression.
- One point about the mechanism that remains unclear is how BRCA1 depletion regulates PLK1. Does BRCA1 depletion regulate PLK1 mostly by affecting its expression (as suggested by the mass spec), or by changing its localization and/or activity? (as suggested by the IF)? This has important implications for the mechanism, as previous data have suggested how BRCA1 may alter PLK1 activity (see minor comments below), but not expression level. In addition to the provided IF images of active PLK1, the authors should provide data on the localization (IF) and expression levels (western) of total PLK1 in BRCA1 mutant cells to clarify if PLK1 expression, activity, or both are altered. Expression levels of other factors identified in the proteomics (e.g. Aurora B) should also be assessed by western blot to validate the proteomics findings. Finally, PLK1 activity and expression levels should be assessed in the mouse mammary cells used for organoid experiments.
- The authors must assess if PLK1 inhibition can reverse spindle misorientation in the BRCA1 mutant MCF10A cell lines. This is assessed in the MCF10A BRCA1 shRNA line and in the patient-derived cells, but not in the engineered BRCA1 mutant MCF10A, which are the “cleanest” system for assessing the role of BRCA1 mutations on cellular phenotypes.

Minor points

- Previous work from the authors implicated the stabilization of HMMR in the spindle positioning defects caused by BRCA1 suppression (He et al. *Oncotarget* 2017). Another paper proposed that after replication stress, BRCA1 inhibits PLK1 activity by inhibiting its phosphorylation by Aurora A (Zou et al. *Cell Cycle* 2013). If BRCA1 loss is activating PLK1 through HMMR stabilization, one would predict that knocking down HMMR would prevent PLK1 activation—however, if BRCA1 loss is activating PLK1 through Aurora A activation, one would predict that inhibiting Aurora A would prevent PLK1 activation. Testing these possibilities would help to clarify if the mechanism of PLK1 activation is through HMMR stabilization or Aurora A activation (or both).
- This is perhaps outside of the scope of this paper—but are there separation of function mutants of BRCA1 that break either its PLK1 activation function or its DDR function? If such mutants exist, they would clearly demonstrate the independence of these two functions.
- Are there other ways to activate PLK1 activity (independent from BRCA1) to assess if this always leads to luminal differentiation defects? In addition to clearly implicating PLK1 in this process, this experiment would be another way to hammer the point home that this phenotype is independent of DNA damage.
- The authors identify a variant of PLK1 that is associated with increased breast cancer risk, and that

overlaps with a H3K27ac-marked PLK1 enhancer in multiple cell types. Does the H3K27ac enhancer peak exist in mammary gland tissue types?

- Figure 1B-D should be graphed in a way that better represents the distribution of the data. For these data, error bars should show standard deviation instead of standard error. If data are not normally distributed, a t-test is not the most appropriate statistical test.
- How do the authors explain the difference between LP and BC cells in DNA damage at 24 hrs after IR treatment (Fig. 1C)? What is the significance of this? Is damage better resolved in BRCA1 mutant BC cells compared to LP cells?
- On Line 170, the authors used the word “deceased” instead of “decreased”.
- “BC” cells should be introduced before using this abbreviation.
- In Fig. 1, correct axis label to be “abnormal”, not “anormal”
- In Fig. 1A, the immunofluorescence images are very difficult to see; BRCA1 and gH2AX foci are not visible. Use higher magnification images and/or zoomed in insets.
- In Fig. 1, the cell numbers listed in the figure (20 cells per experiment) are different than what is listed in the legend (30 cells per experiment).
- In Fig. 1D, how many cells were counted in total? Were the number of cells comparable for different fields of view?
- Overall, the discussion is confusing in how it presents the different BRCA1 phenotypes and their relationship to each other—the final model is unclear. My suggestions for altering the discussion section are below:
 - o Phenotypes of BRCA1 loss/mutation include: defects in DNA damage repair, change in mitotic rate, sensitivity to IR, decrease in colony size, misregulation of spindle orientation, increase in PLK1 activity/expression, defective luminal differentiation. How are all these connected? This might be most easily communicated in a simple model figure.
 - o What is the significance of looking LPs vs BCs? Clarify what is known about mature versus progenitor cells in the literature and how this work advances the current knowledge. If there is something significant about this comparison, it is lost in the text.
 - o In general, the discussion should present data in the context of the literature. I would like to know how this new data fits in the context of this lab’s previous work on BRCA1 and HMMR, as well as previous literature on PLK1, spindle orientation, and luminal differentiation. Include discussion of any new data or speculation on the mechanism of how BRCA1 regulates PLK1 (expression, localization, activity—through HMMR and/or Aurora A).

o The model for how BRCA1 mutant-associated DNA damage is linked to other BRCA1 phenotypes is very unclear from the discussion. Some points of confusion I have listed below:

♣ In the second paragraph of the discussion, it is stated that BRCA1 mutant had reduced clonogenic activity (growth) after irradiation, and this contrasts with the increased mitotic rate. However, increased mitotic rate was only assessed in the ABSENCE of IR. We don't know mitotic rate +IR, so we don't know if this is inconsistent.

♣ Also, it is misleading to say that reduced colony outputs were consistent across human BRCA1 mutant carriers and cell lines and mice. Human BRCA1 mutant carriers exhibited increased colony outputs compared to WT (Fig. 1E, sham-treated), while in MCF10A lines and mouse cells, BRCA-depleted cells showed reduced colony outputs (Fig. 5A, Fig. 6D).

♣ It is also misleading to say that the "inability to correctly orient cell division... contributed to the consequently reduced colony outputs seen", since, as you go on to say in the next sentence, PLK1 inhibition rescues spindle orientation but not cell outputs. Therefore, it would seem that these two phenotypes are independent.

♣ The authors should make it clearer that micronuclei do not directly result from the misoriented mitotic spindle, since micronuclei occur only +IR, and spindle misorientation occurs +IR and -IR.

Reviewer #2 (Remarks to the Author):

Please see attached report.

Reviewer #3 (Remarks to the Author):

This is an interesting manuscript that reports on the influence of BRCA1 variants on mitotic spindle orientation.

Major comment

Most results presented based on MCF10A cells. Can the most important results be replicated with other cell line(s)?

Comments regarding the proteomics specific aspects

It is not completely clear why the authors performed a proteomics experiment here. They hardly use the results and do not do anything else than GO analysis and differential expression analysis with the data.

A lot of details are missing in the workflow description that need to be clarified. Please include important information regarding how the proteomics experiment was performed and analyzed:

- o dataset identifier for proteomexchange.org or MassIVE repository
- o information on how cells were lysed and how protein was isolated

- o type of mass spectrometry, liquid chromatography instrumentation used and settings
- o mode of data acquisition
- o labelled, label-free experiment?
- o software used for raw data analysis and settings
- o experimental design and statistical rationale
- ♣ total number of samples analyzed and described, number of technical and biological replicates
- ♣ methods used for randomization
- ♣ rationale for choice of sample numbers, replicates etc.
- o database search parameters and acceptance criteria for identifications
- ♣ software and search engine used
- ♣ sequence database
- ♣ enzyme specificity
- ♣ fixed modifications, variable modifications
- ♣ known contaminants included, excluded
- ♣ false discovery rates at peptide and protein level
- ♣ proteins identified on basis of unique peptide
- o total protein identifications and overview of identifications per sample
- o information on missing value treatment
- o Which statistical test was used for differential expression analysis and is it suited for number of replicates?
- o type of GO analysis tool used
- o in addition to repository data provide user-friendly, annotated tables with identified proteins in supplement including differential expression analysis, fold-changes etc.

Other comments

MCF10A cell experiment with shRNAs => Possible to show reduction in BRCA1 expression with proteomics data, too?

Reviewer #4 (Remarks to the Author):

Summary:

In this manuscript, He et al. study the effect of BRCA1 deficiency on the orientation of the mitotic spindle in mammary epithelial cells, using a combination of primary human breast epithelial cells, CRISPR-engineered MCF10A cell lines, BRCA1 knockdown, and mouse mammary epithelial cells from conditional *Brca1*-flox/flox;p53-flox/flox mice.

Overall, the study is well controlled. For instance, the authors convincingly show that BRCA1 deficiency, but not DNA damage alone, is sufficient to disturb spindle positioning. This holds true across different model systems, including primary human cells, MCF10A cells and primary mouse epithelial cells. However, the direct evidence that (as promised by the title of the study) pathogenic BRCA1 variants disrupt both spindle orientation and luminal cell differentiation via PLK1 is less strong: First, the link to luminal differentiation is not clear to me from the data provided (see major point 5 below). Second, while the authors DO show that in MCF10A BRCA1 knockdown cells higher levels of pPLK1 are present at

the spindle AND that drug-mediated inhibition of PLK1 restores spindle orientation in BRCA1 mutation carriers, I think they provide correlation between rather than direct causal evidence for disrupted BRCA1 levels and PLK1 activity. Specifically, in the MCF10A CRISPRed mutant cell lines, no PLK inhibitor or PLK knockdown experiments are shown, which makes lines 315-316 of the discussion a bit of an overstatement. Perhaps the authors can rephrase this somewhat, or provide a bit more nuance and/or details about potential caveats of their study.

That being said, this study uses some interesting approaches that are of interest to others in the field of breast cancer research, as well as to researchers interested in the general mechanisms of cell division orientation. As such, this study should be of interest to a wide audience, such as the readership of Nature Communications. I've listed my main comments and questions below, with the hope that the authors can address them to improve and clarify some aspects of their work.

Major points:

1. Throughout: The sizes of the dots and font sizes of the figures are very small. In a PDF print out, the text was hardly readable and I had to zoom in considerably on my screen to see all the details.
2. In my opinion, critical technical details are missing in the materials and methods and/or the main text. The methods are quite condensed in general. At the very least, I think the authors should provide more details on
 - A) The segmentation/image analysis method used for quantification of BRCA and γ H2Ax foci and the segmentation/image analysis method and criteria used to define nuclear abnormality.
 - B) The generation of CRISPRed MCF10A cells: I know from experience that single cell sorting of MCF10A cells is not trivial, so additional information will help reproducibility and adoption of this method by others.
 - C) Cryopreservation of the mouse mammary gland: I am not familiar with this method, so please provide details on the procedure and/or a reference to a protocol.
3. Introduction, lines 63-73: It is not clear in how far the authors describe a well-known and general principle and in how far this mechanism is known play a role in the mammary gland (I think references 21-27 are not from breast tissue). I was not familiar with any of the players mentioned, so to make this study more appealing for a wide audience, I would propose that the authors include a schematic representing the model, including the knowledge gaps that this study addresses. As formulated now, it is difficult to put the information provided in a broader context.
4. Line 107: Can the authors explain the choice for an L shaped micropattern? It is not directly clear to me why an L shaped micropattern is a good representative for what normally happens in an epithelial monolayer (in a dish) or an epithelial bilayer (in vivo). Can the authors explain this or provide references?
5. Line 312 and lines 317/318: I do not believe the authors show a direct mechanistic connection between BRCA1 function and luminal differentiation. It is unclear which luminal features and/or experiments support this statement. This conclusion/interpretation either needs to be toned down or the authors should provide experimental evidence that really investigates luminal differentiation using markers or cell behavior etc. Related to this, it is also not entirely clear to me if the authors now suggest that BC or LC cells are most affected by the BRCA1 mutation, or both. And how does this relate to the cell of origin for BRCA1 like breast cancer, which based on the Blg-Cre mouse model data would be a

luminal progenitor? And how does it potentially relate to the in vivo situation and maintenance of the breast epithelium? Do all donors indeed show a change in BC/LC ratio as alluded to on lines 52-53 of the introduction?

6. Figure 1g: It took us quite some time to realize that the L-shaped micropattern is visible in the background of the rose plots in 1G, which we like. Please mention this in the figure legend.

Notwithstanding this fact, we would still like to see actual images in addition to the quantification to provide concrete examples of normal and aberrant mitotic spindle orientations. This is what the entire paper builds upon but very few actual images are presented.

7. Figure 2: Three of the BRCA1 mutant pathogenic variants don't do anything to spindle orientation. This does not appear to be linked to these mutations residing in a specific domain of the protein. Do the authors have an explanation or can they speculate on why three of the pathogenic variants do not show aberrant spindle orientation and what then is the cause for the pathogenicity of these BRCA1 mutants in this case?

8. Figure 4: Can the authors explain why the experiments in 4f-e were not performed with the MCF10A mutant cell lines? This is especially important since not all of these cell lines respond similarly in the orientation experiments in figure 2 (see point 6 above). For example, are the three pathogenic variants with normal spindle distribution also not upregulating pPLK1 and also not sensitive to PLK1 inhibition? What about wildtype donors? And why are only basal cells (BC) shown for most of the BRCA1-mutant donors?

9. Figure 5: No images of the cadherin-1 and ZO-1 staining pattern are shown. The authors also do not provide a representative staining of wildtype MCF10A on the L-shaped micropattern. Do CDH1 and/or ZO-1 localize differently than in a 3D MCF10A sphere or in a monolayer? And what does this pattern look like in the knockdown? Ideally, this would also be a way to characterize their CRISPRed mutant cell lines, since clonal selection might also occur independent from the engineered genetic mutation. How did the authors characterize their cell lines in this respect? Can they perform stainings for some of the players they mention in the introduction?

10. Related to point 9: In general, figure 5 summarizes some fancy analyses, but leaves some basic questions unanswered. Is cell shape different between mutant and wildtype MCF10A cells? Can this explain the difference in division angle? Have the authors done simple EDU incorporation assays and have they checked for cell death using caspase or a comparable assay?

Minor points:

11. Line 83: The acronym "BCs" was not mentioned in the introduction and should thus be introduced upon first mention.

12. Line 85: There is an inconsistency between the text and figure s1. WT donors are not age-matched as shown in the figure (in the text it says the mutant donor samples are age-matched, which seems to be correct).

13. Line 93: Please change "significant" to "statistically significant" and provide fold changes and/or effect sizes to highlight the biological difference in addition to the statistical difference.

14. Line 106 and 136: Can the authors explain why they use different coating for the primary breast epithelial cells and MCF10A cells and can this in any way affect their interpretation? Based on our experience, MCF10A cells also grow well on collagen.

15. Line 644: Change "described67": 67 needs to be formatted as a reference.

16. Line 673-684: Details on the animal strains are missing. If the animals were not to be maintained by the

authors and glands were reserved in a cryopreserved state, then this should be stated explicitly, but the strains etc. should still be provided.

17. Figure 1a: The contrast of the microscopy images is subpar. γ H2Ax foci cannot be discriminated at the contrast and resolution provided.

18. Figure 1d: We miss additional information on the quantification, where was the line between normal and abnormal? See also major point 2.

19. Figure 2b: We would like to see a more zoomed out view in addition to the example nucleus to judge how representative this magnification is.

20. Figure 2e: Please reconsider the color coding because green versus red is in 2c a discrimination between pathogenic and benign, but in 2e it is also a classification for mitotic spindle orientation. I would recommend to preserve the green and red for the pathogenic variants and to make all the rose plots the same color for a less subjective representation.

21. Figure 3d-e: I would like to see individual datapoints in the bar graph of figure 3d.

22. Figure 3g: Presumably the inset is a magnification of the cell depicted but this is not clear and it is also not informative. Please indicate from which part the magnification is derived and increase the size of the inset to make it a separate panel. Presumably the arrowheads are pointing to lagging chromosomes, but even at 600% magnification on my computer screen this is not clearly visible.

23. Figure 4c and 4e: We would like the authors to show individual data points of individual cells in the intensity quantification.

24. Figure 5: How many independent experiments were performed? And specifically for the cell tracking analysis, how many cell clusters were tracked? There can be large variability simply depending on local density.

25. Supplementary figure 1: Based on the FACS plots all samples show clear populations, with the exception of donor B1. Are the authors sure they are sorting the correct populations? Were additional markers used to verify BC and LC identity after the sort (see also major point 5)?

26. Supplementary figure 2: Can the authors include actual visual examples of the budding and micronuclei?

** See Nature Research's author and referees' website at www.nature.com/authors for information about policies, services and author benefits.

Reviewer #1 (Remarks to the Author):

This manuscript aims to better understand how disease-linked BRCA1 mutations impact cell biology. In particular, the authors want to determine the source of the luminal differentiation abnormalities observed in patients and mice with BRCA1 mutations. Through analysis of both patient-derived cells and edited human cell lines, they find that disease-linked BRCA1 mutations cause defects in the orientation of the mitotic spindle. They propose that this is due to reduced BRCA1 expression, as BRCA1 mutants exhibit reduced BRCA1 foci formation, and spindle misorientation can be phenocopied by knocking down BRCA1 with shRNAs. Proteomics and western blot analysis of BRCA1 mutant cells reveals an increase in expression and activity of the mitotic kinase PLK1, a known regulator of spindle orientation. Inhibition of PLK1 activity in BRCA1 mutant cells not only rescues defects in spindle positioning, but also rescues the loss of luminal cell markers observed in BRCA1 mutant cell lines and organoids. Based on this data, the authors propose that reduced BRCA1 expression leads to PLK1 upregulation and activation, which then results in spindle misorientation and subsequent defects in luminal differentiation. Importantly, they propose that these phenotypes are independent of the role of BRCA1 in DNA damage repair.

Overall, this manuscript advances the field of BRCA1 biology; however, it is important to point out that several findings are not completely novel. A previous paper from the authors already demonstrated that BRCA1 depletion causes spindle orientation defects through HMMR stabilization, and previous work from other labs demonstrated that BRCA1 and HMMR regulate PLK1 activity, and that PLK1 is important for spindle positioning. The most novel aspects of this paper are 1) that the authors connect these observations together in the context of disease-linked BRCA1 mutations, and 2) that the spindle positioning and luminal differentiation functions of BRCA1 are both dependent on PLK1, but are independent of BRCA1's DNA damage repair role.

Before this paper is accepted to Nature Communication, several issues should be addressed to better clarify and support these findings.

RESPONSE: We thank the reviewer for their insightful comments. We performed several additional experiments to address their comments and better clarify and support our findings. Figure panels that are new, or significantly altered, are marked with red font in the marked document.

Major points

- A main conclusion of the paper is that BRCA1's role in orienting the mitotic spindle is independent of its role in DNA damage repair. One way this was demonstrated is MCF10A cells were irradiated to induce DNA damage and subsequent imaging revealed no change in spindle orientation. However, imaging occurred only 24 hrs after irradiation. This does not**

allow much time for chromosome bridges or changes in ploidy to manifest and potentially impact spindle positioning, as suggested by the authors. Therefore, this experiment should be repeated, and spindle positioning should be assessed at longer time points after irradiation (48 and 72 hrs) to allow ample time for cells to go through mitosis and generate bridges and/or aneuploidies.

RESPONSE: We performed new experiments to measure cell viability, abnormal nuclei, and spindle orientation in MCF10A cells 48 hrs and 72 hrs after X-irradiation (new **Figure S6b-d**). MCF10A cell viability was reduced in a dose- dependent (sham to 1 Gy) and time-dependent (24 to 72 hours) manner (new **Figure S6b**). As well, the fraction of abnormal nuclei (micronucleus + budding nuclei) increased in a dose- and time-dependent manner (new **Figure S6c**). But, we did not observe changes to mitotic spindle orientation after irradiation even when measured after 48 hours or 72 hours (new **Figure S6d**). We added this new data to the manuscript (lines 180 – 183).

Additionally, BRCA1 mutant cells should also be assessed +/-IR to determine if their existing spindle orientation defects are exacerbated by additional DNA damage.

RESPONSE: In new experiments, we X-irradiated (1 Gy) MCF10A *BRCA1* mutant/+ cells and measured the cell division axis after 24 hours (new **Figure S6e**). The angles measured in the X-irradiated *BRCA1* mutant/+ cells were not different from those measured in non-irradiated cells (**Figure 3e**). That is, additional DNA damage did not exacerbate the defects observed in the cell division axis. We added this new data to the manuscript (lines 182 - 183).

• It is known that BRCA1 affects centrosome function, which could then cause some degree of spindle assembly checkpoint activation. If cells are delayed in mitosis, they will accumulate mitotic proteins such as PLK1 and Aurora B, as shown here by mass spec. It would therefore be important to determine if overriding the SAC with an MPS1 inhibitor has the same effect as inhibition of PLK1.

RESPONSE: To address the reviewer's excellent comment, we performed new experiments. We transduced MCF10A parental cells with either lentivirus encoding a nonhairpin shRNA (shNHP) or a shRNA targeting BRCA1 (shBRCA1). After 2 days, we stained cells with Hoechst and imaged their progression through mitosis to measure the time spent with aligned condensed chromosomes. We found that BRCA1-silenced MCF10A cells spent a significantly increased time with aligned chromosomes during metaphase (new **Figure S8a-c**). Further, we assessed whether incubation with a MPS1 inhibitor (100 nM NMS P715) affected metaphase duration and the cell division axis. We found that incubation with the MPS1 inhibitor significantly reduced the time BRCA1-silenced MCF10A cells spent with aligned chromosomes prior to anaphase (new **Figure S8b**). However, incubation with the MPS1 inhibitor did not normalize control of the cell division axis for BRCA1-silenced cells grown on L-shaped micropatterns (new **Figure**

S8d). Thus, reduction of time spent in metaphase is not sufficient to recover normal division angles for BRCA1-silenced cells. We added this new data to the manuscript (lines 209 - 214).

• **The authors use BRCA1 foci as a proxy for BRCA1 expression. However, a decrease in BRCA1 foci could mean a decrease in protein expression or, alternatively, a defect in protein localization. The authors must western blot for BRCA1 in the BRCA1 mutant cells to directly determine how the mutations affect BRCA1 expression.**

RESPONSE: We performed western blot analysis in nocodazole-synchronized whole cell lysates from MCF10A BRCA1 mutant/+ cells. For this analysis, we probed the levels of GAPDH, as a total protein loading control, and the levels of Aurora kinase B (AURKB), as a cell cycle-regulated protein that is expressed at an elevated level in mitosis. These controls were equivalently expressed in the mitosis-synchronized cell lysates (new **Figure S4c**). We did not observe consistent reduction in BRCA1 protein expression, or correlated changes to total PLK1 or phosphorylated PLK1 (pPLK1), across the cell lines (new **Figure S4c**). Therefore, total cellular levels of BRCA1, PLK1, or pPLK1 were not predictive for control of the cell division axis. We added these new data and interpretations to the revised manuscript (lines 140 – 142).

• **One point about the mechanism that remains unclear is how BRCA1 depletion regulates PLK1. Does BRCA1 depletion regulate PLK1 mostly by affecting its expression (as suggested by the mass spec), or by changing its localization and/or activity? (as suggested by the IF)? This has important implications for the mechanism, as previous data have suggested how BRCA1 may alter PLK1 activity (see minor comments below), but not expression level. In addition to the provided IF images of active PLK1, the authors should provide data on the localization (IF) and expression levels (western) of total PLK1 in BRCA1 mutant cells to clarify if PLK1 expression, activity, or both are altered. Expression levels of other factors identified in the proteomics (e.g. Aurora B) should also be assessed by western blot to validate the proteomics findings. Finally, PLK1 activity and expression levels should be assessed in the mouse mammary cells used for organoid experiments.**

RESPONSE: Our new experiments showed total levels of BRCA1, PLK1, and pPLK1 (T210) were not consistently altered in mitotic cell lysates from the 15 unique *BRCA1* mutant/+ MCF10A cells (new **Figure S4c**). So, we performed new experiments to examine spatiotemporal control of active PLK1 (new **Figure S9d**). We first immunolocalized BRCA1 to spindle poles at prometaphase and metaphase (new **Figure S9a,b**). BRCA1 intensity at metaphase spindle poles was significantly reduced in causal *BRCA1* mutant/+ MCF10A cells (new **Figure S9c**). These new experiments found pPLK1 was elevated on mitotic spindle poles in causal *BRCA1* mutant/+ MCF10A cells, with the exceptions of R71G/+, R1835X/+, W1873R/+ (new **Figure S9d**); indeed, the mean levels of pPLK1 on metaphase spindle poles correlated with the angle of cell division (new **Figure S9e**). Finally, we show that inhibition of active PLK1, through exposure of

these cells to 0.1 nM BI2536, is sufficient to normalize control of the intrinsic positioning pathway (new **Figure S9f**). These new findings are consistent with pathogenic *BRCA1* mutation altering PLK1 activity rather than overall PLK1 levels. We added these new data to the revised manuscript (lines 219 – 226, and line 234).

• **The authors must assess if PLK1 inhibition can reverse spindle misorientation in the BRCA1 mutant MCF10A cell lines. This is assessed in the MCF10A BRCA1 shRNA line and in the patient-derived cells, but not in the engineered BRCA1 mutant MCF10A, which are the “cleanest” system for assessing the role of BRCA1 mutations on cellular phenotypes.**

RESPONSE: Our new experiments indicate control of the cell division axis can be rescued when MCF10A cells encoding causal *BRCA1* mutations are exposed to 1 nM BI2536 (a specific PLK1 inhibitor) (new **Figure S9f**) (line 234).

Minor points

• **Previous work from the authors implicated the stabilization of HMMR in the spindle positioning defects caused by BRCA1 suppression (He et al. Oncotarget 2017). Another paper proposed that after replication stress, BRCA1 inhibits PLK1 activity by inhibiting its phosphorylation by Aurora A (Zou et al. Cell Cycle 2013). If BRCA1 loss is activating PLK1 through HMMR stabilization, one would predict that knocking down HMMR would prevent PLK1 activation—however, if BRCA1 loss is activating PLK1 through Aurora A activation, one would predict that inhibiting Aurora A would prevent PLK1 activation. Testing these possibilities would help to clarify if the mechanism of PLK1 activation is through HMMR stabilization or Aurora A activation (or both).**

RESPONSE: These are excellent points that we addressed with new experiments and revision of our discussion. Our original submission found that phosphorylated-Aurora kinase A (Thr288) is not altered in BRCA1 silenced cells (**Figure S11a**, lines 245 – 246) and small-molecule inhibition of AURKA is not able to recover control of the cell division axis (**Figure S11d**, lines 249 – 252); similarly, small-molecule inhibition of AURKB is not able to recover control of the cell division axis (**Figure S12e**, lines 249 – 252). Our new experiments indicate that elevated PLK1 activity is necessary for the loss of control of the cell division axis in BRCA1 mutant/+ cells as the inhibition of PLK1 activity recovers division axis control in these cells (new **Figure S9f**). Moreover, as detailed below, ectopic expression of GFP-PLK1 is sufficient to induce the loss of control of the cell division axis in the absence of *BRCA1* mutations (new **Figure S13a,b**, lines 252 – 256). Thus, our results are consistent with a model wherein the loss of BRCA1 serves to elevate PLK1 activity, potentially through the stabilization of HMMR, to disturb the intrinsic spindle positioning pathway. We comment on this potential mechanism in the discussion (lines 391 – 393, 427 – 431).

- **This is perhaps outside of the scope of this paper—but are there separation of function mutants of BRCA1 that break either its PLK1 activation function or its DDR function? If such mutants exist, they would clearly demonstrate the independence of these two functions.**

RESPONSE: We have not precisely localized the regions in *BRCA1* where mutations may be permissive to cell division axis control. Nor have we confirmed the putative detrimental effect to DDR function for mutations in this region. However, we now hypothesize that mutations in the extreme C-terminal exons of the *BRCA1* gene do not disturb control of mitotic spindle orientation. For example, we find that single causal mutations (n= 7) distributed between exon 2 (185delAG) and exon 20 (5283insG) cause loss of cell division axis control. But, single causal mutations (n= 2) in exon 24 do not disrupt the cell division axis. Also, the intensity of pPLK1 is not elevated on mitotic spindle poles for BRCA1 R1835/+ and W1827/+ MCF10A cells (new **Figure S9d**). We added a comment to the discussion (lines 383 – 398).

- **Are there other ways to activate PLK1 activity (independent from BRCA1) to assess if this always leads to luminal differentiation defects? In addition to clearly implicating PLK1 in this process, this would be another way to hammer the point home that this phenotype is independent of DNA damage.**

RESPONSE: In new experiments, we prepared lentivirus to express GFP-PLK1 in parental MCF10A and isolated three clones (EGFP-PLK1 clone 1 to 3) (new **Figure S13a**). Compared to parental cells and a clone that expresses GFP alone, we found that cell divisions for EGFP-PLK1 clones were misoriented on L-shaped micropatterns (new **Figure S13b**). We then clonally expanded EGFP-PLK1 clones and found they were prone to express basal-like structures rather than the luminal structures expressed in control cells (new **Figure S13c**). This new data is added to the revised manuscript (lines 252 – 256, and 273 – 275).

- **The authors identify a variant of PLK1 that is associated with increased breast cancer risk, and that overlaps with a H3K27ac-marked PLK1 enhancer in multiple cell types. Does the H3K27ac enhancer peak exist in mammary gland tissue types?**

RESPONSE: Yes, it does exist in human FACS-enriched mammary tissue types. We added the relevant data to a new **Figure S18c** (lines 340 – 342).

- 1.11• **How do the authors explain the difference between LP and BC cells in DNA damage at 24 hrs after IR treatment (Fig. 1C)? What is the significance of this? Is damage better resolved in BRCA1 mutant BC cells compared to LP cells?**

RESPONSE: This is a provocative observation. In non-carrier tissues, we find statistically significant increases in γ H2Ax foci 24 hours after irradiation in BCs versus LPs ($p < 0.01$). At

this time, we do not fully understand the significance but there are recent published and pre-published articles that may offer insights, and we discuss these observations (lines 401 – 404).

- **In Fig. 1A, the immunofluorescence images are very difficult to see; BRCA1 and gH2AX foci are not visible. Use higher magnification images and/or zoomed in insets.**
- **Figure 1B-D should be graphed in a way that better represents the distribution of the data. For these data, error bars should show standard deviation instead of standard error. If data are not normally distributed, a t-test is not the most appropriate statistical test.**

RESPONSE: We changed the presentation of the data in Figure 1. First, we split the Figure into two Figures, which allows us more space to better represent the heterogeneity of the individual patient sample data in Figure 1B-D. In Figure 1A, we now show higher magnification images. Low magnification images are also presented for human LPs and BCs (**Figure S1c,d**).

For Fig 1B-D, we previously presented 60 values per sample (n=20 cells per experiment, n=3 experiments per sample). We now present Mean \pm SD as well as the individual experimental values for the 3 experiments per sample.

- **On Line 170, the authors used the word “deceased” instead of “decreased”.**

RESPONSE: We made this change on line 79 in the revised manuscript.

- **“BC” cells should be introduced before using this abbreviation.**

RESPONSE: We made this change on line 85 in the revised manuscript.

- **In Fig. 1, correct axis label to be “abnormal”, not “anormal”**

RESPONSE: We made this change in the revised manuscript.

- **In Fig. 1, the cell numbers listed in the figure (20 cells per experiment) are different than what is listed in the legend (30 cells per experiment).**
- **In Fig. 1D, how many cells were counted in total? Were the number of cells comparable for different fields of view?**

RESPONSE: The cell numbers are correctly indicated in the legends.

- **Overall, the discussion is confusing in how it presents the different BRCA1 phenotypes and their relationship to each other—the final model is unclear. My suggestions for altering the discussion section are below:**

o Phenotypes of BRCA1 loss/mutation include: defects in DNA damage repair, change in mitotic rate, sensitivity to IR, decrease in colony size, misregulation of spindle orientation, increase in PLK1 activity/expression, defective luminal differentiation. How are all these connected? This might be most easily communicated in a simple model figure.

RESPONSE: We rewrote our discussion to address the reviewer's comments and link the increase in PLK1 activity to misoriented cell division, colony size and defective luminal differentiation. We also include a new graphical abstract (new **Figure S19**, lines 353 – 356).

o What is the significance of looking LPs vs BCs? Clarify what is known about mature versus progenitor cells in the literature and how this work advances the current knowledge. If there is something significant about this comparison, it is lost in the text.

RESPONSE: We examined LPs and BCs as they are the epithelial fractions that are known to proliferate ex vivo (now indicated and referenced on line 83). We comment in the final paragraph of the revised discussion on how this work advances the current knowledge.

In general, the discussion should present data in the context of the literature. I would like to know how this new data fits in the context of this lab's previous work on BRCA1 and HMMR, as well as previous literature on PLK1, spindle orientation, and luminal differentiation. Include discussion of any new data or speculation on the mechanism of how BRCA1 regulates PLK1 (expression, localization, activity—through HMMR and/or Aurora A).

RESPONSE: Thank you to each reviewer for their insightful comments on our discussion. In the revised manuscript, we now discuss the putative mechanisms by which BRCA1 regulates PLK1 activity (lines 391 – 393), including potential domains in BRCA1 that may be permissive to mutation without disruption of the cell division axis (lines 383 - 389) and other engineered R71G cell models that are permissive to function (lines 393 – 398). We discuss a potential mechanistic link between *BRCA1* mutation, misoriented cell division, and deficient or abnormal LP differentiation (dedifferentiation/transdifferentiation) (lines 377 – 379). We speculate on the reasons that may underlie the differential DNA damage response observed in LPs and BCs (lines 401 – 404) and why PLK1 inhibition selectively rescues oriented cell division but not colony output (lines 407 – 409). Reviewer 1 also highlighted misleading statements in the original discussion (below), and we removed or clarified those statements.

o The model for how BRCA1 mutant-associated DNA damage is linked to other BRCA1 phenotypes is very unclear from the discussion. Some points of confusion I have listed below:

♣ In the second paragraph of the discussion, it is stated that BRCA1 mutant had reduced

clonogenic activity (growth) after irradiation, and this contrasts with the increased mitotic rate. However, increased mitotic rate was only assessed in the ABSENCE of IR. We don't know mitotic rate +IR, so we don't know if this is inconsistent.

♣ Also, it is misleading to say that reduced colony outputs were consistent across human BRCA1 mutant carriers and cell lines and mice. Human BRCA1 mutant carriers exhibited increased colony outputs compared to WT (Fig. 1E, sham-treated), while in MCF10A lines and mouse cells, BRCA-depleted cells showed reduced colony outputs (Fig. 5A, Fig. 6D).

♣ It is also misleading to say that the “inability to correctly orient cell division... contributed to the consequently reduced colony outputs seen”, since, as you go on to say in the next sentence, PLK1 inhibition rescues spindle orientation but not cell outputs. Therefore, it would seem that these two phenotypes are independent.

RESPONSE: The reviewer is correct on each of these points. We removed the following lines from the discussion: “the reduced clonogenic cell numbers appear to contrast with our observation that the mitotic rate was significantly elevated. Reduced secondary colony outputs proved a feature not only of cells from human *BRCA1* mutation carriers but also of BRCA1-silenced MCF10A cells, as well as cells from Cre-activated *BLG-Cre;Brca1^{fl/fl};p53^{+/-}* mice. Live-cell imaging of the colony formation process indicated that the relative proximity of 2 progeny cells, as determined by the cell division axis, promoted subsequent proliferation suggesting that the inability to correctly orient cell division in BRCA1-silenced, *BRCA1*-mutant, or *Brca1*-mutant mammary cells contributed to the consequently reduced colony outputs seen.”

♣ The authors should make it clearer that micronuclei do not directly result from the misoriented mitotic spindle, since micronuclei occur only +IR, and spindle misorientation occurs +IR and -IR.

RESPONSE: We removed the following discussion of the genesis of micronuclei and the activation of pro-inflammatory signals: “The enhanced genesis of micronuclei displayed by primary human *BRCA1*-mutant LP- and BC-derived cells when stressed by X-irradiation suggests a pathway that might induce NF- κ B signaling and activation of pro-inflammatory signals indirectly³⁵ or directly⁴⁷⁻⁴⁹ and hence induce more extensive mutagenesis⁵⁰. Active NF- κ B signaling is known to occur in RANK-positive LPs due to DNA damage in *BRCA1*-mutant tissues⁵¹ and, in the mouse model, aberrant differentiation precedes induction of *Rankl* expression with potential cross-talk to tissue-resident macrophages resulting in a tumour-promoting environment¹⁴. In a TP53-deficient mammary cell, it is therefore inviting to speculate that a BRCA1 deficiency causing mitotic instability would cooperate with abnormal spindle orientation to promote tumorigenesis.”

Reviewer #2 (Remarks to the Author):

In this study, the authors wanted to understand whether the luminal differentiation abnormalities observed in female BRCA1-mutation carriers is linked to the loss of control of the mitotic spindle orientation they had previously observed in BRCA1-deficient human mammary cells.

They started by analyzing proliferation (growth and mitotic rate), cell division and response to DNA damage of Luminal progenitors (LPs) and Basal cells (BCs) from 3 cancer-free normal reduction mammoplasty and compared them with 3 age-matched BRCA1-mutation carriers breast samples. Overall, BRCA1-mutant LPs and BCs behave the same, including a reduction in colony-forming activity after irradiation despite an intrinsic increased mitotic rate and a defective cell division orientation control.

To further dissect the phenotype and mechanism, they decide to use engineered MCF10A cell lines each harboring at the heterozygous state, a distinct BRCA1 variant, either pathogenic or benign, or controls (positive: Dex2-3 or negative: Ex10). After confirming these clinically relevant mutant cells behaved as the breast samples obtained from the female carriers, they showed that although DNA damage levels were correlated with the level of expression of BRCA1, they show that DNA damage was not sufficient to induce the spindle positioning defects in these cells. Proteome analysis revealed an overexpression of proteins involved in cell cycle/mitosis and a deeper analysis revealed that PLK1 hyperactivity in BRCA1-mutant cells was responsible for the disruption of the normal cell division axis control, and consequently for a perturbation in the normal lineage distribution in a BRCA1-mutant breast tissue, characterize by a 'loss' of luminal cells.

This is an interesting study which reports a set of new findings obtained from the analyses of patient samples as well as dissected in vitro and confirmed in vivo using an appropriate mouse model. However, there are a few important controls missing and additional experiments should be performed in order to consolidate the study:

RESPONSE: We thank the reviewer for their positive comments. We performed several additional experiments to better consolidate our findings and address the reviewer's comments. Our description of new experimental data - and Figure panels that are new, or significantly altered - are marked with red font in the marked document.

Comments:

- **General comment:** we need to see BRCA1 WB for all the different samples studied (Human patient samples, MCF10A lines and mouse breast cells). The ability to form BRCA1 foci is not a sufficient surrogate for concluding about BRCA1 levels.

RESPONSE: To respond to the reviewer's comment, we performed western blot analysis for BRCA1 expression in nocodazole-synchronized whole cell lysates from MCF10A BRCA1

mutant/+ cells (new **Figure S4c**). The following controls were equivalently expressed: GAPDH, a total protein loading control, and Aurora kinase B (AURKB), a cell cycle regulated protein that is normally expressed at an elevated level in mitosis (new **Figure S4c**). In whole cell lysates from synchronized MCF10A BRCA1 mutant/+ cells, we did not observe consistent reduction in BRCA1 protein expression (new **Figure S4c**). We added these new data to the revised manuscript (lines 140 – 142).

Unfortunately, we do not have sufficient FACS-purified human patient samples to perform whole cell lysate and western blot analysis.

• Figure1: The pictures from panel 1a are not really convincing, because the signal is faint (maybe add some enlarged cells). We also need to know which patients these IF analyses correspond to as the mutations are different and knowing is relevant to the interpretation.

RESPONSE: We changed the presentation to split Figure 1 into two Figures. We now include high magnification images (**Figure 1a**) and low magnification images (**Figure S1c,d**). We have provided the patient details in the Figure legend.

• LP data are shown but it would be helpful to add the BCs in the Supplemental 1.

RESPONSE: We made this change and added the BC data to **Figure S1d**.

Also, in the text it says Cyclin B2 but everywhere is Cyclin B1.

RESPONSE: We corrected our mistake (line 93).

• Finally, given the CFC and the mitotic rate results, an analysis of cell death, and maybe the cell cycle checkpoints, would be further consolidating the conclusions.

RESPONSE: We returned to the brightfield movies for the freshly-isolated primary human mammary epithelial cells undergoing cell division on L-shaped patterns. From these movies, we measured the levels of cell death phenotypically defined by morphology¹ including cellular blebbing or budding, condensation, and fragmentation or condensation of the nucleus. We found that levels of cellular death were significantly reduced in primary cells isolated from *BRCA1* mutation carriers during their ex vivo growth on L-shaped patterns (new **Figure S3b**). We added this data to the manuscript (line 114).

• Figure2: better state in the text what the MCF10A controls are, with their genotype.

RESPONSE: We made this change in the text (lines 126 – 128).

Data are very nice. I would simply suggest here a better organization and color coding for the Fig2e, so the reader can easily see the results.

RESPONSE: We changed the organization of the panel and the color coding of the L-patterns in **Figure 3e**.

The authors should speculate about the 3 mutants (alterations located in the BRCT area) which behave differently than the rest of the mutants (BRCA1 levels, we need to see a WB, and cell division orientation).

RESPONSE: We now hypothesize that mutations in the extreme C-terminal exon 24 of the *BRCA1* gene do not disturb control of mitotic spindle orientation. While single causal mutations (n= 7) distributed between exon 2 (185delAG) and exon 20 (5283insG) cause the loss of control of the cell division axis, single causal mutations (n= 2) in exon 24 do not disrupt the cell division axis (**Figure 3e**) and do not alter the intensity of phosphorylated PLK1 at metaphase spindle poles (new **Figure S9d**). We added a comment on these observations to the discussion (lines 381 – 394).

We also added a WB for BRCA1 expression in these cells (new **Figure S4c**) and our confirmation of cell division angles in redundant clones (new **Figure S4e**).

• **Figure3: need to homogenize the data shown in Fig3a and S5, which report respectively 30min and 24h post IR (1Gy), especially because at 30min, no difference was observed in the human breast samples (Fig1c).**

RESPONSE: **Figure 4a** (formerly Figure 3a) correlates two responses across 15 MCF10A *BRCA1* mutant/+ cells. Experimentally, these responses are each induced by 1 Gy X-radiation but measured at different times post-irradiation. On the y-axis, we plot the levels of γ H2Ax foci that persist at 24 hours. We interpret these values to represent the inverse of DNA damage repair or resolution (includes repair, cell death, etc.) and we observe elevated persistent DNA damage in cells expressing causal *BRCA1* mutations (**Figure S5b**). BRCA1, CtIP, and MRE11 associate most strongly at DNA breaks during early repair². So, on the x-axis, we plot the levels of BRCA1 foci that are present 30 minutes after 1Gy X-radiation (**Figure 3c**).

Fig2b correlates BRCA1 foci after 30min 1Gy with CFC 5 days after irradiation at 0.25Gy. BRCA1 foci after 0.25Gy (most relevant time point) should be shown. And also, why the previous CFC assay performed in the human samples was done 8 days after irradiation at 1Gy? For consistency and a better comparison between conditions, it would be good to present the same dose and time points unless there is a valid reason, in which case it should be explained in the text. And legend of the plots in Fig3g should be more apparent (0 vs 1Gy).

RESPONSE: We revise the Methods and provide more details to the CFC and irradiation protocols (lines 734 – 743). Briefly, the CFC assays using immortal MCF10A cells (**Figure 3b**) measured colonies at day 5 according to our published protocols³. Experimentally, we evaluated the outcomes that followed graded X-ray doses (0.25, 0.5, 0.75, 1.0 Gy) and we present the outcomes that follow 0.25 Gy, which showed the most pronounced distinction between MCF10A *BRCA1* causal/+ versus MCF10A *BRCA1* benign/+ cells.

Briefly, we did not have sufficient FACS-enriched primary BCs or LPs to perform X-ray dose responses prior to colony assays. For this reason, we used a published X-radiation dose (1 Gy, ⁴), which is shown to induce DNA damage foci, and have near equivalent effects on CFC activity in primary BCs and LPs⁴.

We have changed the presentation to make the legend of the plots apparent in **Figure 4g**.

• **Figure4: Nice data with a few comments:**

o **It is important to show WB confirmation of the overexpression of PLK1 (as well as hyperactivity, pPLK1 T210) in the BRCA1-silenced cells as well as in the various engineered mutant MCF10A. and suggest ideas of the PLK1 overexpression.**

RESPONSE: We performed new experiments to examine total BRCA1, PLK1, and phosphorylated PLK1 (pPLK1) levels by western blot analysis in whole cell lysates from MCF10A *BRCA1* mutant/+ cells synchronized in mitosis through prior incubation with nocodazole (200 ng/ mL for 16 hours). The levels of the following control proteins did not vary between lysates: GAPDH, a total protein loading control, and AURKB, a cell cycle regulated protein that is expressed at an elevated level in mitosis (new **Figure S4c**). Moreover, total BRCA1, PLK1, and pPLK1 levels did not consistently differ in these cell lysates (new **Figure S4c**). These analyses are included in the revised manuscript (lines 140 – 142, 216 – 218).

We also analyzed pPLK1 intensity on metaphase spindle poles by immunofluorescence analysis. We found elevated levels of pPLK1 on spindle poles for cells encoding pathogenic *BRCA1* mutant/+ (new **Figure S9d**); moreover, exposure of these cells to 0.1 nM BI2536, to inhibit PLK1 activity, was sufficient to normalize control of the intrinsic positioning pathway (new **Figure S9f**). These data are included in the revised manuscript (lines 219 – 226, line 234).

o **A BRCA1 staining at the spindle added in supplementary would be helpful.**

RESPONSE: We include new analysis for BRCA1 immunofluorescence intensity through the stages of mitosis. This new analysis indicates spatiotemporally restricted localization of BRCA1 to prometaphase and metaphase mitotic spindles (new **Figure S9a,b**). These data are included in the revised manuscript (lines 219 – 221).

o For Fi4i, the authors should show all the human samples (as B1 and B2 LP are missing). One matched pair only could be shown in the main figure and the other 2 sets in supplementary S9.

RESPONSE: We performed PLK1 inhibition experiments on BCs and LPs that were FACS-enriched from *BRCA1* mutation carriers and cryopreserved. Unfortunately, upon thaw, the cryopreserved LPs from two *BRCA1* mutation carriers (B1 and B2) did not recover well or divide on L-patterns.

o the authors should re-organize the flow of this paragraph: AURKA/B experiments could be moved down, and S8 and S9 could be combined or follow each other. And we miss AURKB staining in BRCA1-silenced cells.

RESPONSE: We made this change (lines 243 – 256). We now also include images from immunofluorescence analysis for AURKB in *BRCA1*-silenced cells (new **Figure S12a**; lines 245 – 246).

o What about the under-represented proteins from the proteomic analysis?

RESPONSE: We extracted the under-expressed proteins and added the enriched GO terms (new **Figure S7c,d**). These data are included in the revised manuscript (lines 204 – 205).

o Line 225: the figures' references are: S8d and S10c.

RESPONSE: We made this change (lines 251 – 252).

• Figure5: interesting set of data. It would be useful to further articulate the rationale of the assay using the RFP/TUBA1B expression.

RESPONSE: We added to the text the rationale for using RFP/TUBA1B was to track mitotic spindle position and follow the cell division angles in living cells in real-time (lines 265 – 267).

Also, it is critical to mitigate the conclusions here. Indeed, the luminal population in reduced in number in the *BRCA1*-silenced cells, and expansion-cell division orientation-cell to cell contact properties are defective. However, one could say that the expansion for example is reduced AND delayed. And most importantly, the authors do not mention/comment at all the notion that *BRCA1*- null breast cells were shown to undergo de-differentiation/ transdifferentiation, which is a different concept than an inability differentiate into luminal cells.

RESPONSE: The reviewer's observations are correct. In response, we have mitigated the conclusions that we draw from our data. First, we altered the title for the manuscript. We also discuss the notion that BRCA1-null breast cells have been shown to deficient or abnormal luminal differentiation^{5, 6} (lines 377 – 380). We thank the reviewer for these important comments.

• Figure6: these data obtained from the mouse derived MECS are well done and convincing. Conclusions need however to also include the notion that the phenotype seems to indicate that PLK1 promotes acquisition of luminal features, but it could also be a result of a function in the maintenance of the luminal features. A luminal lineage reporter model should be used to firmly conclude.

RESPONSE: This is an excellent point. We conclude that oriented cell division – which is controlled by the PLK1-dependent intrinsic positioning pathway - is critical to the position of the resultant daughter cells and the acquisition of cell-cell contacts that promote luminal features.

In new experiments, we create MCF10A clones that overexpress GFP-PLK1. These new studies indicate PLK1 overexpression is sufficient to misorient cell divisions on L-shape patterns and alter the phenotype expressed by clonally expanded cells to decrease luminal and promote basal-like structures (new **Figure S13a-c**). This new data is added to the manuscript (lines 252 – 256; and 273 – 275).

In new experiments, we classify colony features generated from MCF10A cells (**Figure S14a-c**) and primary LPs or BCs (new **Figure S14d-i**) through multiple highly correlated parameters: cellular density; cell-cell contacts; and, expression of vimentin versus ZO-1 and CDH1 (lines 266 – 270). The reviewer also recommends the use of a luminal lineage reporter model for the mouse-derived organoid cultures. Rather than use a genetic model to track, for example, K14- or K8- expression in mouse-derived organoids, we measured the expression of these cytoskeletal markers by immunofluorescence analysis at defined timepoints (**Figure 7h**) (lines 321 – 325).

Also, the authors should propose an explanation for the results from Fig6d: PLK1 inhibition rescues phenotypical features of the BRCA1-null organoids as well as the cell division axis but not their ability to grow.

RESPONSE: In brief, we propose that the loss of BRCA1 disturbs control of the cell division axis due to a consequent increase in active PLK1 at prometaphase or metaphase spindle poles, which can be recovered by PLK1 inhibition. In addition, loss of BRCA1 reduces genomic stability. As PLK1 inhibition does not recover cellular growth induced by loss of BRCA1, we conclude supervision of genotoxic stress by BRCA1 may be independent of its regulatory role on PLK1 activity. We added this explanation to the discussion (lines 407 – 412).

- **A rescue experiment using an inducible shRNA would be important to test whether the phenotype is reversible and also it would test whether there is another path than PLK1 activity that is involved in the phenotype.**

RESPONSE: We performed new experiments using MCF12A sublines that were transduced with lentiviral particles containing Tet Repressor protein- (pLV-tTR-KRAB-IP) and shRNA-coding (pLV-THM) cassettes to create tet-on inducible expression of short hairpin RNAs (shRNA). We used (1) the MCF-12A naïve subline, (2) a subline that expresses a shRNA containing a scrambled sequence (shScr) with no target sequence in the human genome, and (3) a subline that expresses a shRNA targeting BRCA1³. Cells were treated with 2 µg/ml doxycycline for four days prior to analysis. Unfortunately, the characteristics of these engineered adherent cells were quite unique when grown on L-shape patterns. That is, these cells failed to adhere well to the L-patterns and required upwards of 3 hours to form an L-shape. Moreover, the divisions of the control cells (MCF-12A naïve) were not correctly oriented on L-patterns. Thus, we could not complete these requested studies.

Our new studies indicate that augmented PLK1 activity is both necessary for the loss of cell division axis control associated with *BRCA1* mutations (new **Figure S9f**; **Figure 5f, 5i**) and sufficient to disturb the cell division axis in the absence of *BRCA1* mutations (new **Figure S13b**).

- **The authors should include a comment about the role of the Notch pathway in this study.**
- **The authors should also mention what they think should be done next to embrace these findings and further promoter PLK1 as a valid candidate/target in BRCA1-driven breast cancers: a BRCA1 mouse model with the PLK1 variant in the enhancer region? A deep biochemistry study to understand how PLK1-mediated function of BRCA1 really works mechanistically? What about replication stress? Can they use the micropatterns as a readout of a screen to find associated proteins in this function?**

RESPONSE: These are a series of excellent suggestions. In response, we reference the Notch pathway in the introduction (line 67) and discussion (lines 368 – 373); and, we discuss the proposed mechanism in which BRCA1 indirectly activates PLK1 and controls mitotic spindle orientation (lines 389 - 393). The use of the L-pattern as a readout for a systemic screen will require some automation in the image analysis, which is an area of ongoing study.

Other comments:

- 2.20 • **The title may need to be changed**

RESPONSE: We changed the title and removed the reference to luminal differentiation.

- **In some places, the results should be better articulated for clarity.**

- **The pivotal assay for the cell division orientation need to be better explained in the manuscript and/or in the Methods to make it clear for a general audience.**

RESPONSE: Thank you for this suggestion. We better articulate the results in the revised manuscript. We also expand the explanation of the cell division axis assay in the Methods (lines 754 – 773) and Results (lines 108 – 109) sections.

- **I think it would be interesting to comment about the results from S1b-c.**

RESPONSE: Current **Figure S1a** displays the FACS results and gating strategy utilized by the Eaves group to enrich for BCs and LPs from premenopausal tissues of BRCA1 mutation carriers and non-carriers. The Eaves group helped to pioneer these enrichment procedures from the human breast⁷⁻¹⁴. In the discussion, we comment on the observation that the BRCA1 mutant donor samples analysed here do not show an obvious difference in LP ratio (lines 414 – 416).

- **S13c is not mentioned in the text**

RESPONSE: Current **Figure S17c** is now described on lines 311 – 317.

- **S14 legend should be more explanatory**

RESPONSE: Current **Figure S18** has been expanded and we clarified the acronyms to better explain the data.

Reviewer #3 (Remarks to the Author):

This is an interesting manuscript that reports on the influence of BRCA1 variants on mitotic spindle orientation.

3.1 Major comment

Most results presented based on MCF10A cells. Can the most important results be replicated with other cell line(s)?

RESPONSE: We agree that it is critical to replicate results in different experimental systems. For example, we report the loss of control of the cell division axis in primary human mammary epithelial cells freshly isolated from *BRCA1* mutation carriers (**Figure 2b,d**). We confirm this finding in primary mouse mammary cells (Cre-transduced BLG-Cre *Brca1* fl/fl Trp53 +/- cells) (**Figure 7a**) as well as in the various sublines of *BRCA1* mutant/+ MCF10A cell lines (**Figure 3e**). Mechanistically, we report an elevated level of phosphorylated PLK1 at metaphase spindle poles in *BRCA1*-silenced MCF10A cells (**Figure 5c**). We confirm this finding in primary human mammary epithelial cells freshly isolated from *BRCA1* mutation carriers (**Figure 5h**) and genome edited *BRCA1* mutant/+ MCF10A cells (new **Figure S9d**). Phenotypically, we report an altered colony output in *BRCA1*-silenced MCF10A cells (**Figure 6a**). We confirm this finding in organoid cultures generated from Cre-transduced primary mouse mammary epithelial cells (**Figure 7f-h**).

3.2 Comments regarding the proteomics specific aspects

It is not completely clear why the authors performed a proteomics experiment here. They hardly use the results and do not do anything else than GO analysis and differential expression analysis with the data.

A lot of details are missing in the workflow description that need to be clarified.

RESPONSE: We performed the proteomics experiment to discover putative mechanisms by which loss of *BRCA1* disturbs control of the cell division axis (lines 194 – 197). In response to the reviewer’s request, we provide the requested details in an expanded Methods section “Mass spectrometry analysis” (lines 892 – 944).

Please include important information regarding how the proteomics experiment was performed and analyzed:

o dataset identifier for proteomexchange.org or MassIVE repository; information on how cells were lysed and how protein was isolated; type of mass spectrometry, liquid chromatography instrumentation used and settings; mode of data acquisition; labelled, label-free experiment?; software used for raw data analysis and settings; experimental design and statistical rationale; total number of samples analyzed and described, number

of technical and biological replicates; methods used for randomization; rationale for choice of sample numbers, replicates etc; database search parameters and acceptance criteria for identifications; software and search engine used; sequence database; enzyme specificity; fixed modifications, variable modifications; known contaminants included, excluded; false discovery rates at peptide and protein level; proteins identified on basis of unique peptide; total protein identifications and overview of identifications per sample; information on missing value treatment; Which statistical test was used for differential expression analysis and is it suited for number of replicates?; type of GO analysis tool used; in addition to repository data provide user-friendly, annotated tables with identified proteins in supplement including differential expression analysis, fold-changes etc.

RESPONSE: We apologize for the truncated methods that we provided with our original submission. The requested details are now included in the Methods section of the revised manuscript (lines 892 – 944).

3.3 Other comments. MCF10A cell experiment with shRNAs => Possible to show reduction in BRCA1 expression with proteomics data, too?

RESPONSE: BRCA1 was not identified in the proteomics dataset from control transduced or shBRCA1-transduced MCF10A cells.

Reviewer #4 (Remarks to the Author):

In this manuscript, He et al. study the effect of BRCA1 deficiency on the orientation of the mitotic spindle in mammary epithelial cells, using a combination of primary human breast epithelial cells, CRISPR-engineered MCF10A cell lines, BRCA1 knockdown, and mouse mammary epithelial cells from conditional Brca1-flox/flox;p53-flox/flox mice. Overall, the study is well controlled. For instance, the authors convincingly show that BRCA1 deficiency, but not DNA damage alone, is sufficient to disturb spindle positioning. This holds true across different model systems, including primary human cells, MCF10A cells and primary mouse epithelial cells.

4.1 However, the direct evidence that (as promised by the title of the study) pathogenic BRCA1 variants disrupt both spindle orientation and luminal cell differentiation via PLK1 is less strong: First, the link to luminal differentiation is not clear to me from the data provided (see major point 5 below).

RESPONSE: We agree with the reviewer's assessment. We modified the Title of the study and removed the reference to luminal differentiation.

4.2 Second, while the authors DO show that in MCF10A BRCA1 knockdown cells higher levels of pPLK1 are present at the spindle AND that drug-mediated inhibition of PLK1 restores spindle orientation in BRCA1 mutation carriers, I think they provide correlation between rather than direct causal evidence for disrupted BRCA1 levels and PLK1 activity. Specifically, in the MCF10A CRISPRed mutant cell lines, no PLK inhibitor or PLK knockdown experiments are shown, which makes lines 315-316 of the discussion a bit of an overstatement. Perhaps the authors can rephrase this somewhat, or provide a bit more nuance and/or details about potential caveats of their study.

RESPONSE: Our new experiments discovered elevated phosphorylated PLK1 (pPLK1) intensity at metaphase spindle poles in MCF10A *BRCA1* mutant/+ cells (new **Figure S9d**, lines 224 – 226), which validates our results in human mammary epithelial cells isolated from *BRCA1* mutation carriers (**Figure 5h**, lines 235 – 237). The intensity of pPLK1 at metaphase spindle poles correlates with an inability to orient the cell division axis in MCF10A *BRCA1* mutant/+ cells (new **Figure S9e**, line 226). Moreover, incubation with a PLK1 inhibitor (0.1 nM BI2536) recovered control of the cell division axis for the panel of *BRCA1* mutant/+ MCF10A cells (new **Figure S9f**, line 234), which agrees with our findings in primary human *BRCA1* mutant progenitor cells (**Figure 5i**, line 237 - 239). Thus, hyperactive PLK1 at spindle poles is needed for the loss of control of the cell division axis in *BRCA1* mutant/+ cells.

In a new set of experiments, we generate 3 clonal MCF10A cell lines that overexpress GFP-PLK1 and one clonal MCF10A cell line that overexpresses GFP alone (lines 252 – 256).

This series of experiments demonstrates that overexpression of PLK1 alone (in the absence of *BRCA1* mutation) is sufficient to induce the loss of control of the cell division axis (new **Figure S13b**) and to alter the phenotypes expressed by clonally-expanded MCF10A cells (new **Figure S13c**). Therefore, we now conclude that hyperactive PLK1 is both necessary for the loss of control of the cell division axis in *BRCA1* mutant cells and is also sufficient to induce the loss of control of the cell division axis in the absence of *BRCA1* mutations.

That being said, this study uses some interesting approaches that are of interest to others in the field of breast cancer research, as well as to researchers interested in the general mechanisms of cell division orientation. As such, this study should be of interest to a wide audience, such as the readership of Nature Communications. I've listed my main comments and questions below, with the hope that the authors can address them to improve and clarify some aspects of their work.

RESPONSE: Thank you for your comments and questions. We performed several additional experiments to address the reviewer's comments and clarify some aspects of our work. Figure panels that are new, or significantly altered, are marked with red font in the marked document.

Major points:

2.3 1. Throughout: The sizes of the dots and font sizes of the figures are very small. In a PDF print out, the text was hardly readable and I had to zoom in considerably on my screen to see all the details.

RESPONSE: We divided the data presented in the original Figure 1 into two Figures (Figure 1 and Figure 2). We now present the mean values per experiment (n=3 per sample) rather than the individual cell datapoints (n= 60 cells per sample).

3.4 2. In my opinion, critical technical details are missing in the materials and methods and/or the main text. The methods are quite condensed in general. At the very least, I think the authors should provide more details on

A) The segmentation/image analysis method used for quantification of BRCA and γ H2Ax foci and the segmentation/image analysis method and criteria used to define nuclear abnormality.

B) The generation of CRISPRed MCF10A cells: I know from experience that single cell sorting of MCF10A cells is not trivial, so additional information will help reproducibility and adoption of this method by others.

C) Cryopreservation of the mouse mammary gland: I am not familiar with this method, so please provide details on the procedure and/or a reference to a protocol.

RESPONSE: We apologize for the reduced format of our Methods in our original submission. In response to the reviewer's request, we expanded the methods section and specifically added more details to the description of segmentation/image analysis for BRCA1 foci and γ H2Ax foci (lines 779 – 783), the criteria used to define abnormal nuclei (lines 774 – 778), the generation and clonal expansion of genome-edited MCF10A cells (lines 788 – 815), and the cryopreservation of the mouse mammary gland tissues (lines 947 – 949).

3. Introduction, lines 63-73: It is not clear in how far the authors describe a well-known and general principle and in how far this mechanism is known play a role in the mammary gland (I think references 21-27 are not from breast tissue). I was not familiar with any of the players mentioned, so to make this study more appealing for a wide audience, I would propose that the authors include a schematic representing the model, including the knowledge gaps that this study addresses. As formulated now, it is difficult to put the information provided in a broader context.

RESPONSE: In the revised introduction, we clarify the literature related to more general principles (lines 61 – 62) and the prior literature analysed in the mammary gland (lines 65 – 67). We also provide a graphical abstract to summarize our findings (new **Figure S19**).

4. Line 107: Can the authors explain the choice for an L shaped micropattern? It is not directly clear to me why an L shaped micropattern is a good representative for what normally happens in an epithelial monolayer (in a dish) or an epithelial bilayer (in vivo). Can the authors explain this or provide references?

RESPONSE: When the L-pattern is introduced in the results section (lines 108 – 109), we detail their utility in assessing intrinsic spindle positioning in single cells and we cite the seminal publication in this area¹⁵. Briefly, the intrinsic spindle positioning pathway in the mitotic cell is overseen by extrinsic cues. These extrinsic cues include cell-cell contacts and cell-matrix adhesions. Experimentally, cell-cell interactions can be removed through the study of single cell divisions and the location of cell-adhesion contacts can be controlled through the use of L-shaped patterns.

Prior to mitosis, the mitotic cell rounds up but maintains cell-matrix adhesions. These adhesions are termed retraction fibers, which serve to educate the intrinsic spindle position. For this reason, cells grown on tissue culture plastic tend to orient their cell divisions along their long axis prior to division (as that is the location of the retraction fibers)¹⁶. On L-patterns, the retraction fibers are oriented to the arms of the L-shape, which educates the cell division axis along the hypotenuse of the L-pattern.

5. Line 312 and lines 317/318: I do not believe the authors show a direct mechanistic connection between BRCA1 function and luminal differentiation. It is unclear which

luminal features and/or experiments support this statement. This conclusion/interpretation either needs to be toned down or the authors should provide experimental evidence that really investigates luminal differentiation using markers or cell behavior etc.

RESPONSE: We agree with the reviewer's comment. In response, we altered the title for the manuscript. We also provide new evidence that strongly correlates cellular density and cell-cell contacts in CFC assays generated with primary human mammary epithelial cells (MECs) with the expression of luminal differentiation markers, CDH1 and ZO-1, and the absence of expression of vimentin (new **Figure S14d-i**). This data complements our analysis of these same markers in organoids generated from primary mouse MECs (**Figure 7h**). In addition, we discuss the published work that shows BRCA1-null breast cells undergo phenotypes consistent with deficient or abnormal luminal differentiation^{5, 6, 17, 18} (line 379).

Related to this, it is also not entirely clear to me if the authors now suggest that BC or LC cells are most affected by the BRCA1 mutation, or both. And how does this relate to the cell of origin for BRCA1 like breast cancer, which based on the Blg-Cre mouse model data would be a luminal progenitor? And how does it potentially relate to the in vivo situation and maintenance of the breast epithelium? Do all donors indeed show a change in BC/LC ratio as alluded to on lines 52-53 of the introduction?

RESPONSE: In our revised manuscript, we discuss the significance of our new insights, and other recent work, on the important issues of BC/LP ratios, and the cell-of-origin for BRCA1-like breast cancer (line 413 – 433).

6. Figure 1g: It took us quite some time to realize that the L-shaped micropattern is visible in the background of the rose plots in 1G, which we like. Please mention this in the figure legend. Notwithstanding this fact, we would still like to see actual images in addition to the quantification to provide concrete examples of normal and aberrant mitotic spindle orientations. This is what the entire paper builds upon but very few actual images are presented.

RESPONSE: We now mention the L-pattern in the Figure Legends and provide representative images for primary mammary cells on micropatterns as well as their angles of cell division (**Figure 2b**). Thank you for suggesting these improvements.

7. Figure 2: Three of the BRCA1 mutant pathogenic variants don't do anything to spindle orientation. This does not appear to be linked to these mutations residing in a specific domain of the protein. Do the authors have an explanation or can they speculate on why three of the pathogenic variants do not show aberrant spindle orientation and what then is the cause for the pathogenicity of these BRCA1 mutants in this case?

RESPONSE: While additional experimental work is needed, we now hypothesize that mutations in (at least) the very C-terminal exon 24 of the *BRCA1* gene do not disturb control of mitotic spindle orientation. Both *BRCA1* R1835/+ and W1827/+ lie in exon 24. We draw further support for this hypothesis by the observation that active pPLK1 intensity is not elevated on mitotic spindle poles for R1835/+ and W1827/+ cells (new **Figure S9d**). However, we have not precisely localized the region in *BRCA1* where mutations are permissive to the maintenance of cell division axis control. Nor have we confirmed the putative detrimental effect of mutations in this region to DNA damage response.

R71G lies in a region of the *BRCA1* gene where mutations are disruptive to cell division axis control. Moreover, R71G is predicted to be causal¹⁹. However, MCF10A cells engineered with *BRCA1* R71G/+ gave a neutral result in our study (**Figure 3e**). Mechanistically, R71G is predicted to cause aberrant splicing of the transcript and result in a truncated protein¹⁹. But, MCF10A *BRCA1* R71G/+ cells did not differ in BRCA1 foci formation or total BRCA1 levels in whole cell lysates. Thus, our engineered cells may not reconstitute the pathogenic attributes of this mutation. Importantly, the following R71G/+ engineered cells also fail to reconstitute the pathogenic attributes of this mutation:

1. Human breast cancer MCF7 cells were engineered to express one of 14 distinct point mutations in the BRCA1 RING domain, including R71G or 13 other mutations, and their affect on centrosome amplification was measured. R71G was one of only two mutations (also T37R) that did not alter centrosome control²⁰.
2. BRCA1 was silenced in human cervical cancer HeLa cells and a series of synthetic BRCA1 deletion mutants were assayed for their function in a HDR assay, including R71G and 15 other mutants in the N-terminal RING domain. Ten of 16 mutations resulted in loss of HDR activity. R71G was the only mutation that augmented HDR activity relative to WT. R71G was also able to purify cellular BARD1 protein²¹.
3. *BRCA1* mutations - in the first 100 N-terminal amino acids - were generated from a randomly mutagenized BRCA1 library and selected for loss of E2- (UbcH5a) binding in yeast. This analysis identified 22 missense variants in 19 residues that lose interaction with E2-binding and E3 ubiquitin ligase activity. However, R71G did not inhibit the E2 interaction²².

We added a comment to the discussion about R71G engineered models that, like ours, fail to reconstitute the pathogenic attributes of this mutation (lines 394 – 398).

8. Figure 4: Can the authors explain why the experiments in 4f-e were not performed with the MCF10A mutant cell lines? This is especially important since not all of these cell lines respond similarly in the orientation experiments in figure 2 (see point 6 above). For example, are the three pathogenic variants with normal spindle distribution also not upregulating pPLK1 and also not sensitive to PLK1 inhibition?

RESPONSE: The reviewer has raised excellent points. We ran new experiments to study spatiotemporal control of PLK1 activity measured by phospho-Thr210 PLK1 (pPLK1) intensity at mitotic spindle poles in MCF10A BRCA1 mutant/+ cells and to measure their responses to PLK1 inhibition. For these experiments, we analyzed pPLK1 intensity at metaphase spindle poles in MCF10A BRCA1 mutant/+ cells expressing causal variants, including three variants (R71G/+, R1835X/+, W1873R/+) that do not misorient the cell division axis. We focussed our analysis on metaphase spindle poles because BRCA1 intensity on spindle poles is highest at metaphase (new **Figure S9a,b**). These new experiments found that pPLK1 levels were significantly elevated on mitotic spindle poles for MCF10A cells expressing causal BRCA1 mutations, with the exceptions of R71G/+, R1835X/+, W1873R/+ (new **Figure S9d**); indeed, the mean levels of pPLK1 strongly correlated with the ability to orient the cell division axis (new **Figure S9e**). Finally, we show that inhibition of PLK1 activity, through exposure of these cells to 0.1 nM BI2536, is sufficient to normalize control of the intrinsic positioning pathway (new **Figure S9f**). We added these new data and interpretations to the revised manuscript (lines 219 – 226, line 234).

What about wildtype donors? And why are only basal cells (BC) shown for most of the BRCA1-mutant donors?

RESPONSE: We performed PLK1 inhibition experiments on cryopreserved BCs and LPs that had been previously FACS-enriched from BRCA1 mutation carriers. Upon thaw, unfortunately, the cryopreserved LPs from two BRCA1 mutation carriers (B1 and B2) did not recover well or divide on L-patterns. For this reason, we can not determine the cell division axis in these primary donor cells on L-patterns grown in the presence of PLK1 inhibition.

9. Figure 5: No images of the cadherin-1 and ZO-1 staining pattern are shown. The authors also do not provide a representative staining of wildtype MCF10A on the L-shaped micropattern. Do CDH1 and/or ZO-1 localize differently than in a 3D MCF10A sphere or in a monolayer? And what does this pattern look like in the knockdown?

RESPONSE: We evaluated MCF10A and primary human mammary cells grown at clonal density. In the generated MCF10A colonies, we measured cell numbers and cell contacts per unit area to derive three phenotypes (high density, mixed density, low density) (**Figure S14a-c**). Our prior published work demonstrates that high density MCF10A colonies are ZO-1^{high}CD49f^{low} ³. We also performed new experiments to classify colonies generated from primary human mammary cells by cell density and expression of vimentin, CDH1 or ZO-1. In these primary human mammary colonies, the intensity of staining per unit area was correlated with cellular density (new **Figure S14d-f**). In this way, we confirmed that low density colonies tended to express high levels of vimentin while high density colonies expressed elevated levels of CDH1 and ZO-1

(new **Figure S14g-i**). We include these new data in the revised manuscript, including images for vimentin, ZO-1 and CDH1.

Ideally, this would also be a way to characterize their CRISPRed mutant cell lines, since clonal selection might also occur independent from the engineered genetic mutation. How did the authors characterize their cell lines in this respect? Can they perform stainings for some of the players they mention in the introduction?

RESPONSE: Thank you for raising this important issue about the potential for clonal selection to affect the cellular phenotypes expressed. We possess at least two distinct clones for 14 of 15 MCF10A *BRCA1* mutant/+ variants; the exception is MCF10A *BRCA1* 185delAG/+ cells, which we purchased from Horizon Discovery. For each clone, we sequenced the edited site (**Figure S4a**) and performed immunoblot analysis for BRCA1, PLK1, pPLK1 (new **Figure S4c**). We also selected the following three MCF10A *BRCA1* mutant/+ cell-lines of high interest and confirmed cell division axis phenotypes in two distinct clones (new **Figure S4e**)(line 147 – 149).

1. R71G/+ clones were obtained from the Park lab – R71G/+ cells gave results that were discordant with other mutations in the RING domain. Therefore, we confirmed the results were similar in two distinct clones (**Figure S4e**).
2. A1708E/+ clones were created in the Maxwell lab – A1708E/+ cells misorient cell division. We observed mixed effects on the cell division axis for C-terminal *BRCA1* mutations. Therefore, we confirmed equivalent results for two distinct clones (**Figure S4e**).
3. R1835X/+ clones were created in the Maxwell lab – R1835/+ cells orient cell division. We observed mixed effects on the cell division axis for C-terminal *BRCA1* mutations. Therefore, we confirmed equivalent results for two distinct R1835/+ clones (**Figure S4e**).

We did not address the cortical retention of NuMA-LGN or dynein heavy chain at the mitotic cell cortex in the *BRCA1* mutant/+ cells but we have previously done so in *BRCA1*-silenced MCF10A. This published analysis demonstrates abnormal cortical retention of these complexes³.

10. Related to point 9: In general, figure 5 summarizes some fancy analyses, but leaves some basic questions unanswered. Is cell shape different between mutant and wildtype MCF10A cells? Can this explain the difference in division angle? Have the authors done simple EDU incorporation assays and have they checked for cell death using caspase or a comparable assay?

RESPONSE: We re-analysed the movies generated from the colony forming assays contained in the data presented in Figure 6 and measured cell shape, cell cycle time and apoptotic rate (Lines 295 – 297). Briefly, we observed no differences in cell shape as measured by cell shape roundness (new **Figure S16e**), but *BRCA1*-silenced MCF10A cells required a significantly longer time measured between cell divisions (cell division 1 to daughter cell division 1) (new **Figure S16f**), and showed an elevated rate of apoptosis (new **Figure S16g**). Recall that these

movies were generated from 2 – 4 cell clusters identified at 36 hours post plating. Thus, these movies are selected for those cells that were division-proficient in the first 1.5 days after plating. Among those colony forming cells, the apoptotic rate was extremely low for shNHP-transduced control cells and was elevated in BRCA1-silenced MCF10A cells (new **Figure S16g**).

Minor points:

11. Line 83: The acronym “BCs” was not mentioned in the introduction and should thus be introduced upon first mention.

RESPONSE: We now introduce basal cells before using the “BC” acronym (line 85).

12. Line 85: There is an inconsistency between the text and figure s1. WT donors are not age-matched as shown in the figure (in the text it says the mutant donor samples are age-matched, which seems to be correct).

RESPONSE: We now describe the primary donor samples as premenopausal rather than age-matched.

13. Line 93: Please change “significant” to “statistically significant” and provide fold changes and/or effect sizes to highlight the biological difference in addition to the statistical difference.

RESPONSE: We made the suggested change (line 95) and we added the fold change in the Legend.

14. Line 106 and 136: Can the authors explain why they use different coating for the primary breast epithelial cells and MCF10A cells and can this in any way affect their interpretation? Based on our experience, MCF10A cells also grow well on collagen.

RESPONSE: We clarified these distinctions in our revised Methods section under “Analysis of cell division axis” (line 754 – 773). In addition, we performed new experiments to address changes in the cell division axis that may occur when MCF10A cells are grown on different coating. We found MCF10A cell division angles were equivalent on fibronectin- or collagen-coated L patterns.

15. Line 644: Change “described67”: 67 needs to formatted as a reference.

RESPONSE: We have made this change (line 816, under “Lentiviral production and transduction”).

16. Line 673-684: Details on the animal strains are missing. If the animals were not be maintained by the authors and glands were reserved in a cryopreserved state, then this should be stated explicitly, but the strains etc. should still be provided.

RESPONSE: We have added these details (lines 945 - 957, under “Isolation and culture of primary mouse mammary epithelial cells and organoid cultures”).

17. Figure 1a: The contrast of the microscopy images is subpar. γ H2Ax foci cannot be discriminated at the contrast and resolution provided.

RESPONSE: We modified Figure 1 and now provide images at a higher magnification to enable the better discrimination of γ H2Ax- and BRCA1 foci.

18. Figure 1d: We miss additional information on the quantification, where was the line between normal and abnormal? See also major point 2.

RESPONSE: We did not clearly state that “abnormal” nuclei captured the sum of two measurements of abnormality: 1. the presence of micronuclei; or, 2. the presence of nuclear budding. We clarified the legends and we now include representative images (**Figure S2a**).

19. Figure 2b: We would like to see a more zoomed out view in addition to the example nucleus to judge how representative this magnification is.

RESPONSE: We include lower magnification images to demonstrate that the presented images are representative (**Figure S4b**).

20. Figure 2e: Please reconsider the color coding because green versus red is in 2c a discrimination between pathogenic and benign, but in 2e it is also a classification for mitotic spindle orientation. I would recommend to preserve the green and red for the pathogenic variants and to make all the rose plots the same color for a less subjective representation.

RESPONSE: We followed the reviewer’s advice and changed the rose plots to be black rather than colored red or green (**Figure 3e**).

21. Figure 3d-e: I would like to see individual datapoints in the bar graph of figure 3d.

RESPONSE: We followed the reviewer’s advice and provide the individual experimental data as well as the mean and standard deviation (**Figure 4d**).

22. Figure 3g: Presumably the inset is a magnification of the cell depicted but this is not clear and it is also not informative. Please indicate from which part the magnification is derived and increase the size of the inset to make it a separate panel. Presumably the arrowheads are pointing to lagging chromosomes, but even at 600% magnification on my computer screen this is not clearly visible.

RESPONSE: We made this change in revised **Figure 4g**.

23. Figure 4c and 4e: We would like the authors to show individual data points of individual cells in the intensity quantification.

RESPONSE: We made this change in revised **Figure 5c** and **Figure 5e**.

24. Figure 5: How many independent experiments were performed? And specifically for the cell tracking analysis, how many cell clusters were tracked? There can be large variability simply depending on local density.

RESPONSE: We include these details in the legend for revised **Figure 6**. These experiments were performed in six well dishes with replicate wells per treatment. For each well, three cell clusters were tracked per experiment. Triplicate experiments were performed. In total, the formation of 18 colonies were tracked per treatment.

25. Supplementary figure 1: Based on the FACS plots all samples show clear populations, with the exception of donor B1. Are the authors sure they are sorting the correct populations? Were additional markers used to verify BC and LC identity after the sort (see also major point 5)?

RESPONSE: FACS-based enrichment of BCs and LPs was performed by the Eaves' lab (BC Cancer Research Center, Canada), which pioneered these enrichment procedures from the human breast⁷⁻¹⁴.

26. Supplementary figure 2: Can the authors include actual visual examples of the budding and micronuclei?

RESPONSE: We include representative images in the revised **Figure S2a**.

References for response to reviewers

1. Ziegler, U. & Groscurth, P. Morphological features of cell death. *News Physiol Sci* **19**, 124-128 (2004).
2. Whelan, D.R. & Rothenberg, E. Super-resolution mapping of cellular double-strand break resection complexes during homologous recombination. *Proc Natl Acad Sci U S A* **118** (2021).
3. He, Z. *et al.* BRCA1 controls the cell division axis and governs ploidy and phenotype in human mammary cells. *Oncotarget* **8**, 32461-32475 (2017).
4. Balani, S. in *Interdisciplinary Oncology*, Vol. Doctor of Philosophy-PhD 153 (University of British Columbia, Vancouver, BC, Canada; 2018).
<https://open.library.ubc.ca/soa/cIRcle/collections/ubctheses/24/items/1.0365711>
5. Bai, F. *et al.* BRCA1 suppresses epithelial-to-mesenchymal transition and stem cell dedifferentiation during mammary and tumor development. *Cancer Res* **74**, 6161-6172 (2014).
6. Wang, H. *et al.* Inadequate DNA Damage Repair Promotes Mammary Transdifferentiation, Leading to BRCA1 Breast Cancer. *Cell* **178**, 135-151.e119 (2019).
7. Stingl, J. *et al.* Purification and unique properties of mammary epithelial stem cells. *Nature* **439**, 993-997 (2006).
8. Eirew, P. *et al.* A method for quantifying normal human mammary epithelial stem cells with in vivo regenerative ability. *Nat Med* **14**, 1384-1389 (2008).
9. Eirew, P. *et al.* Aldehyde dehydrogenase activity is a biomarker of primitive normal human mammary luminal cells. *Stem Cells* **30**, 344-348 (2012).
10. Kannan, N. *et al.* Glutathione-dependent and -independent oxidative stress-control mechanisms distinguish normal human mammary epithelial cell subsets. *Proc Natl Acad Sci U S A* **111**, 7789-7794 (2014).
11. Raouf, A. *et al.* Transcriptome analysis of the normal human mammary cell commitment and differentiation process. *Cell Stem Cell* **3**, 109-118 (2008).
12. Knapp, D.J.H.F., Kannan, N., Pellacani, D. & Eaves, C.J. Mass Cytometric Analysis Reveals Viable Activated Caspase-3. *Cell Rep* **21**, 1116-1126 (2017).
13. Pellacani, D. *et al.* Analysis of Normal Human Mammary Epigenomes Reveals Cell-Specific Active Enhancer States and Associated Transcription Factor Networks. *Cell Rep* **17**, 2060-2074 (2016).
14. Nguyen, L.V. *et al.* Barcoding reveals complex clonal dynamics of de novo transformed human mammary cells. *Nature* **528**, 267-271 (2015).
15. Thery, M. *et al.* The extracellular matrix guides the orientation of the cell division axis. *Nat Cell Biol* **7**, 947-953 (2005).
16. Théry, M., Jiménez-Dalmaroni, A., Racine, V., Bornens, M. & Jülicher, F. Experimental and theoretical study of mitotic spindle orientation. *Nature* **447**, 493-496 (2007).
17. Bach, K. *et al.* Time-resolved single-cell analysis of Brca1 associated mammary tumorigenesis reveals aberrant differentiation of luminal progenitors. *Nat Commun* **12**, 1502 (2021).
18. Lim, E. *et al.* Aberrant luminal progenitors as the candidate target population for basal tumor development in BRCA1 mutation carriers. *Nat Med* **15**, 907-913 (2009).
19. Vega, A. *et al.* The R71G BRCA1 is a founder Spanish mutation and leads to aberrant splicing of the transcript. *Hum Mutat* **17**, 520-521 (2001).
20. Kais, Z., Chiba, N., Ishioka, C. & Parvin, J.D. Functional differences among BRCA1 missense mutations in the control of centrosome duplication. *Oncogene* **31**, 799-804 (2012).
21. Ransburgh, D.J., Chiba, N., Ishioka, C., Toland, A.E. & Parvin, J.D. Identification of breast tumor mutations in BRCA1 that abolish its function in homologous DNA recombination. *Cancer Res* **70**, 988-995 (2010).

22. Morris, J.R. *et al.* Genetic analysis of BRCA1 ubiquitin ligase activity and its relationship to breast cancer susceptibility. *Hum Mol Genet* **15**, 599-606 (2006).

REVIEWER COMMENTS

Reviewer #1 (Remarks to the Author):

We thank the authors for their thorough response to our review. The major points we raised have been addressed to our satisfaction—therefore, we recommend this paper for publication in Nature Communications. Below are some additional minor points that should be addressed before final publication.

Minor points:

- Line 31 of the abstract still says “BRCA1 +/-mutant cells exhibit low BRCA1 levels”, although the authors have now demonstrated that mutant BRCA1 cells do not exhibit expression defects but localization defects. This should read “BRCA1 +/-mutant cells exhibit defective BRCA1 localization”.
- In Fig. S6d, the first plot on the top left indicates that “6%” of the cell division axes were oriented, but I think this label should indicate something closer to “60%”.
- When comparing Fig. 3E and Fig. S6E, it would be useful to know the statistical significance of the differences in cell division axes orientation +/-IR to confirm there is really no effect of irradiation.
- In Fig. S8c, “Prolonged metaphase” is labeled in peach in the legend but appears nowhere in the actual graphs.
- Lines 141-142 state that “total levels of BRCA1 measure in whole mitotic lysates were not consistently altered in the 15 unique BRCA1 mutant/+ MCF10A cell lines”; however, it should be noted that the 5382insG BRCA1 mutant does exhibit decreased expression.
- In the figures demonstrating cell division angles, it is unclear what the gray percentages labeling the inner and outer circles refer to.
- In Fig. S9d, it is unclear if the error bars represent SEM of the 3 triplicate experiments or the SEM of the total 60 cells (if representing individual cells, should be SD).

Reviewer #2 (Remarks to the Author):

Revised Manuscript

Pathogenic BRCA1 variants disrupt PLK1-1 regulation of mitotic spindle orientation

Zhengcheng et al

1.4.22

In this study, the authors have provided new insights about the role BRCA1 in the control of cell division orientation which appears to be independent of its functions in DNA damage, and also mediated via PLK1. They also suggest that this function may be participating in luminal differentiation, which could explain the basal features of most BRCA1 cancer. However, since this is still not completely

demonstrated in this study, as we also know from literature that BRCA1-driven breast cancers may actually de-differentiate from luminal to basal, it is good the authors mitigated the conclusions.

The revised manuscript has been largely improved and the authors did address most of my comments by either re-analyzing some of their existing data or by performing new experiments. I would suggest adding a few comments in the discussion about some of the results and an emphasis on the novel aspects of their study, but the take home message of the study is novel and advances the field of BRCA1 biology.

A few comments remain:

- The new data of BRCA1 protein expression levels that were asked are shown for only some of the MCF10A lines (due to sample limitation) and are indicating no major differences in expression although some mutations are leading to protein truncations. A complementary RNA analysis of BRCA1 transcript would have been great, especially in the primary human samples.
- The pPLK1/PLK1 levels should be coming from the same gels and normalized to total PLK1 to really give the best conclusion. Also, it is unclear how BRCA1 loss induces increased pPLK1 or PLK1 hyper-activation. Is BRCA1 modulating PLK1 phosphorylation? Is BRCA1 rather mediating PLK1 recruitment to the spindles? Or both? This maybe require a comment in the discussion. Indeed, overexpression PKL1 is not quite the same as specifically using a phosphomimetic PKL1 which would then directly address this question, and then the phenotype should not be 'rescuable' with the PLK1 inhibitor.
- What about total PLK1 levels at spindles?
- FigS4c may need to be relocated somewhere else in the manuscript.
- The authors need to add back the information about where the mutations are located in the human primary samples, it's very important.
- The link between PLK1 and BRCA1 breast cancer risk is still a little weak but interesting. Just be cautious on the conclusions.
- The notch pathway has been discussed and given the literature involving also NUMB in the transdifferentiation process, and the fact that NUMB also participates in asymmetric division control, a comment in the discussion would be interesting. Numb and PLK1 were shown to bind and regulate each other, including Numb phosphorylation by PLK1 (one reference: *oncogene* 2018, doi: 10.1038/onc.2017.379).

A couple suggestions for the final version:

Line 86: 'prophylactic mastectomies' should be used instead of 'samples of breast tissues obtained from...'

Line 88: isn't it 8 days??

Line 104: reorganize the text so figure 2a is mentioned in the next paragraph for clarity.

Line 156: rephrase

Line 209 and 248: rather use 'restored' instead of 'normalized'

Line 211: rephrase for clarity

Reviewer #3 (Remarks to the Author):

Our comments were addressed. Ad 3.1 We meant to ask whether the main proteomics findings could be replicated in another cell line.

Reviewer #4 (Remarks to the Author):

I have read the revised manuscript and the extensive rebuttal the authors provided and I think that all my questions have been addressed. I have no further comments at this time.

** See Nature Research's author and referees' website at www.nature.com/authors for information about policies, services and author benefits.

REVIEWER COMMENTS

Reviewer #1 (Remarks to the Author):

We thank the authors for their thorough response to our review. The major points we raised have been addressed to our satisfaction—therefore, we recommend this paper for publication in Nature Communications.

RESPONSE: We are grateful to the reviewer for their prior and current comments, which have clarified and strengthened our findings and the manuscript.

Below are some additional minor points that should be addressed before final publication.

Minor points:

- Line 31 of the abstract still says “BRCA1 +/-mutant cells exhibit low BRCA1 levels”, although the authors have now demonstrated that mutant BRCA1 cells do not exhibit expression defects but localization defects. This should read “BRCA1 +/-mutant cells exhibit defective BRCA1 localization”.

RESPONSE: We made this change on line 31 in the revised manuscript.

- In Fig. S6d, the first plot on the top left indicates that “6%” of the cell division axes were oriented, but I think this label should indicate something closer to “60%”.

RESPONSE: We corrected this typo in Figure S6d in the revised manuscript.

- When comparing Fig. 3E and Fig. S6E, it would be useful to know the statistical significance of the differences in cell division axes orientation +/-IR to confirm there is really no effect of irradiation.

RESPONSE: We performed these statistical comparisons (Mann-Whitney test). We include the p values in Figure 4f, Figure S6d, and Figure S6e, and we describe the statistical test in the legends in the revised manuscript (lines 564, 1106, 1111).

- In Fig. S8c, “Prolonged metaphase” is labeled in peach in the legend but appears nowhere in the actual graphs.

RESPONSE: Thank you for pointing out this error. We removed “Prolonged metaphase” from the legend in Figure S8c in the revised manuscript.

- Lines 141-142 state that “total levels of BRCA1 measure in whole mitotic lysates were not consistently altered in the 15 unique BRCA1 mutant/+ MCF10A cell lines”; however, it should be noted that the 5382insG BRCA1 mutant does exhibit decreased expression.

RESPONSE: Yes, our original phrasing was inaccurate. We rephrased this sentence in the revised manuscript. Lines 141 – 144 now state: “While mitotic cell lysates prepared from individual BRCA1 mutant/+ MCF10A cell lines showed alterations in total levels of BRCA1 relative to parental cells, a consistent change in the total levels of BRCA1 was not observed across the 15 unique BRCA1 mutant/+ MCF10A cell lines”.

- **In the figures demonstrating cell division angles, it is unclear what the gray percentages labeling the inner and outer circles refer to.**

RESPONSE: We clarified that these gray percentages indicate “percent of total mitotic cells examined” in the Figure legends in the revised manuscript (lines 513, 535, 564, 602, 606, 614, 654, 1040, 1061, 1106, 1111, 1150, 1177, 1207, 1230, 1240).

- **In Fig. S9d, it is unclear if the error bars represent SEM of the 3 triplicate experiments or the SEM of the total 60 cells (if representing individual cells, should be SD).**

RESPONSE: The error bars represent SD of 60 individual cells. This is noted in the legends for Figure S9e in the revised manuscript (line 1168).

Reviewer #2 (Remarks to the Author):

Revised Manuscript

Pathogenic BRCA1 variants disrupt PLK1-1 regulation of mitotic spindle orientation

Zhengcheng et al

1.4.22

In this study, the authors have provided new insights about the role BRCA1 in the control of cell division orientation which appears to be independent of its functions in DNA damage, and also mediated via PLK1. They also suggest that this function may be participating in luminal differentiation, which could explain the basal features of most BRCA1 cancer. However, since this is still not completely demonstrated in this study, as we also know from literature that BRCA1-driven breast cancers may actually de-differentiate from luminal to basal, it is good the authors mitigated the conclusions.

The revised manuscript has been largely improved and the authors did address most of my comments by either re-analyzing some of their existing data or by performing new experiments. I would suggest adding a few comments in the discussion about some of the results and an emphasis on the novel aspects of their study, but the take home message of the study is novel and advances the field of BRCA1 biology.

RESPONSE: We thank the reviewer for their prior and current comments.

A few comments remain:

• The new data of BRCA1 protein expression levels that were asked are shown for only some of the MCF10A lines (due to sample limitation) and are indicating no major differences in expression although some mutations are leading to protein truncations. A complementary RNA analysis of BRCA1 transcript would have been great, especially in the primary human samples.

RESPONSE: We provide new RNA analysis of *BRCA1* transcript expression in primary human samples and MCF10A *BRCA1* mutant/+ cell lines. For the analysis of *BRCA1* expression in MCF10A cells, we isolated RNA from unsynchronized cultures. For the analysis of *BRCA1* expression in primary cells, we isolated RNA from FACS-enriched subpopulations. Because cells were not synchronized prior to the RNA preparation, cell cycle-dependent changes in *BRCA1* expression affect this analysis.

Real-time RT-PCR was performed in triplicate on the RNA preps using two separate sets of primers: primers recognizing N-terminal sequence or primers recognizing C-terminal sequence (listed in Supplemental Table 2). For preps isolated from MCF10A *BRCA1* mutant/+ cells, we performed statistical comparison of normalized values obtained from individual *BRCA1* mutant/+ cells versus parental MCF10A cells. For preps isolated from primary *BRCA1* mutation carriers, we performed statistical comparison of normalized values obtained from individual *BRCA1* carriers versus mean values from pre-menopausal non-carriers.

In MCF10A cells and primary LPs, *BRCA1* 185delAG (or 185delA) mutations associated with a significant increased *BRCA1* expression but other variants did not show consistent effects. With the material available for this analysis, however, it is difficult to distinguish the effects on expression that results from an increase in the relative cellular proliferation in the sample versus the effect on expression that results from the specific *BRCA1* variant. For this reason, we include the data here rather than in the main text of the paper.

• The pPLK1/PLK1 levels should be coming from the same gels and normalized to total PLK1 to really give the best conclusion.

RESPONSE: As outlined in the Methods, the antibodies detecting pPLK1 (T210) or total PLK1 were both generated in rabbits, which excludes their use on the same gels. Analysis on the same gels using alternate antibodies that recognize pPLK1 (T210) or total PLK1 would be further complicated by the inability to visually separate the respective bands. Although pPLK1(T210) and total PLK1 were measured on different gels, we measured their respective abundance using the same amount of protein loaded (20 µg) from the same cellular lysates in duplicate gels. For these reasons, we are confident in the conclusions generated from these western blot analyses.

Also, it is unclear how BRCA1 loss induces increased pPLK1 or PLK1 hyper-activation. Is BRCA1 modulating PLK1 phosphorylation? Is BRCA1 rather mediating PLK1 recruitment to the spindles? Or both? This may require a comment in the discussion.

RESPONSE: The reviewer is correct. We do not currently know how BRCA1 loss induces an increase in pPLK1 at the spindle pole. As recommended by the reviewer, we have added the following comment to the discussion on line 373 – 376 of the revised manuscript: “We also do not currently know how BRCA1 loss induces an increase in pPLK1 at the spindle pole, and future research into mechanisms that may modulate PLK1 phosphorylation or recruitment to the poles is warranted.”

- The link between PLK1 and BRCA1 breast cancer risk is still a little weak but interesting. Just be cautious on the conclusions.

RESPONSE: We agree with the reviewer’s comment. We have altered the wording in our discussion (starting on line 354) as follows: “PLK1 is now revealed to be a critical target of BRCA1 that is aberrantly activated when BRCA1 function is compromised, which then causes a (associates with) loss of control of spindle positioning and deficient acquisition of luminal features by the progeny generated (Figure S19).”

Indeed, overexpression PKL1 is not quite the same as specifically using a phosphomimetic PKL1 which would then directly address this question, and then the phenotype should not be ‘rescuable’ with the PLK1 inhibitor.

RESPONSE: We extensively discussed the point raised by the reviewer. In considering the experiment, we presumed the reviewer was recommending us to knockin PLK1 T210D in BRCA1 mutant/+ or parental MCF10A cells. T210D knockin lines would be expected to show phenotypes consistent with activated PLK1 but would not express endogenous WT PLK1 and would resist the PLK1 inhibitor. But, we note Paschal et al [PMID: 2256210] performed this experiment in tert-immortalized human retinal pigment epithelial cells and HCT116 carcinoma cells to examine cell cycle regulation. Their findings “caution against relying on PLK1 T210D as an in vivo surrogate for the natively activated kinase.” Of course, other publications that overexpress PLK1 T210D find that it approximates some of the functions of the mitotically activated PLK1 [PMIDs: 10980711; 12524548; 18615013]. However, Paschal et al “found that the T210D substitution severely diminishes Plk1’s ability to support BubR1 phosphorylation, K-fiber stability, chromosome congression, and cell proliferation.” Thus, while potentially interesting, we feel the generation and study of phosphomimetic PLK1 T210D knockin cells lies outside the scope of the current manuscript.

- What about total PLK1 levels at spindles?

RESPONSE: We examined total PLK1 levels at the spindles and have added the new data to Figure S9d. In brief, total PLK1 levels at the spindle were not consistently elevated in BRCA1 mutant lines that misorient cell divisions (Figure S9d).

Here, we plot the levels of total PLK1 versus pPLK1 measured in metaphase cells (but in separate preparations of metaphase cells). This plot shows little variation in normalized total PLK1 levels (x-axis) at the spindle but marked variation in normalized pPLK1 levels (y-axis). Elevated pPLK1 levels were observed in BRCA1 mutant/+ cells that misorient the cell division axis.

- **FigS4c may need to be relocated somewhere else in the manuscript.**

RESPONSE: We agree with the reviewer's suggestion. We have moved the prior Figure S4c (western blot analysis of total BRCA1, PLK1, pPLK1 levels) from the supplemental Figures to Figure 3c. To accomplish this relocation, we removed the prior Figure 3a (schematic of *BRCA1* mutations in MCF10A cells). We now place the schematic of *BRCA1* mutations in MCF10A cells in the supplemental Figure S4a. We have modified the Figure Legends (lines 525 – 528; lines 1045 – 1050) to match these changes in the revised manuscript.

- **The authors need to add back the information about where the mutations are located in the human primary samples, it's very important.**

RESPONSE: We added the mutation information for *BRCA1* mutant donors in Figure 1a and the detailed donor information is included in Supplementary Table 1.

- **The notch pathway has been discussed and given the literature involving also NUMB in the transdifferentiation process, and the fact that NUMB also participates in asymmetric division control, a comment in the discussion would be interesting. Numb and PLK1 were shown to bind and regulate each other, including Numb phosphorylation by PLK1 (one reference: oncogene 2018, doi: 10.1038/onc.2017.379).**

RESPONSE: We have modified the discussion slightly to add this reference to the revised manuscript (lines 369 – 370).

A couple suggestions for the final version:

Line 86: 'prophylactic mastectomies' should be used instead of 'samples of breast tissues obtained from...'

RESPONSE: We made this change (line 86) in the revised manuscript.

Line 88: isn't it 8 days??

RESPONSE: We corrected this typo (line 88) in the revised manuscript.

Line 104: reorganize the text so figure 2a is mentioned in the next paragraph for clarity.

RESPONSE: We made this change (Line 105-107). In the revised manuscript, we start the paragraph with "Consistent with the expression of a mitotic instability phenotype, *BRCA1*-mutant samples displayed a corresponding ~50% reduction in their content of cells with colony-forming activity²⁹ at day 8 post 1Gy X-radiation (**Figure 2a**)".

Line 156: rephrase

RESPONSE: We have rephrased this sentence in the revised manuscript to state "To further evaluate the impact of *BRCA1* mutations on protein function, we measured other characteristics, including the levels of DNA damage and sensitivity to PARP inhibitors, in these *BRCA1* mutant/+ cell lines" (lines 159-161).

Line 209 and 248: rather use 'restored' instead of 'normalized'

RESPONSE: We made this change on line 211 and line 251 in the revised manuscript.

Line 211: rephrase for clarity

RESPONSE: Yes, our original phrasing was not clear. We rephrased this sentence in the revised manuscript to state “Thus, the restoration of normal metaphase kinetics through MPS1 inhibition is not sufficient to correctly orient the cell division axis in BRCA1-silenced cells” (lines 213-215).

Reviewer #3 (Remarks to the Author):

Our comments were addressed. Ad 3.1 We meant to ask whether the main proteomics findings could be replicated in another cell line.

RESPONSE: Thank you to the reviewer for their prior criticisms and comments, which prompted us to clarify aspects of our work and has improved the manuscript.

Reviewer #4 (Remarks to the Author):

I have read the revised manuscript and the extensive rebuttal the authors provided and I think that all my questions have been addressed. I have no further comments at this time.

RESPONSE: We are grateful to the reviewer for their prior and current comments, which have improved and clarified several aspects of our work.

REVIEWERS' COMMENTS

Reviewer #2 (Remarks to the Author):

2.18.22

We thank the authors for their thorough response to our review.

All the points have now been addressed to our satisfaction, including their comments regarding data that they won't include in the manuscript, but which were provided for our review.

Therefore, we recommend this paper for publication in Nature Communications.

** See Nature Portfolio's author and referees' website at www.nature.com/authors for information about policies, services and author benefits

He Z, et al. Pathogenic *BRCA1* variants disrupt PLK1-regulation of mitotic spindle orientation. (NCOMMS-21-23127B)

REVIEWER COMMENTS

Reviewer #2 (Remarks to the Author):

2.18.22

We thank the authors for their thorough response to our review. All the points have now been addressed to our satisfaction, including their comments regarding data that they won't include in the manuscript, but which were provided for our review.

Therefore, we recommend this paper for publication in Nature Communications.

***RESPONSE:* We are grateful to the reviewer for their prior and current comments, which have clarified and strengthened our findings and the manuscript.**